# The E3 ubiquitin-protein ligase Trim31 alleviates non-alcoholic fatty liver disease by targeting Rhbdf2 in mouse hepatocytes

Minxuan Xu [1,2,3,7], Jun Tan [1,2,7✉], Wei Dong[4,7], Benkui Zou[4,7], Xuepeng Teng[4,7], Liancai Zhu[3,7], Chenxu Ge[1,2,3,7], Xianling Dai[1,2,3], Qin Kuang[1,2,3], Shaoyu Zhong[1,2], Lili Lai[1,2], Chao Yi[1,2], Tingting Tang[1,3], Junjie Zhao[1,2], Longyan Wang[1,2], Jin Liu[1,2], Hao Wei[1,2], Yan Sun[1,2,3], Qiufeng Yang[1,2], Qiang Li[1,2], Deshuai Lou[1,2], Linfeng Hu[1,2,3], Xi Liu[1,2], Gang Kuang[1,2], Jing Luo[5], Mingxin Xiong[1,2], Jing Feng[1,6], Chufeng Zhang[4✉] & Bochu Wang[3✉]

Systemic metabolic syndrome significantly increases the risk of morbidity and mortality in patients with non-alcoholic fatty liver disease (NAFLD) and non-alcoholic steatohepatitis (NASH). However, no effective therapeutic strategies are available, practically because our understanding of its complicated pathogenesis is poor. Here we identify the tripartite motif-containing protein 31 (Trim31) as an endogenous inhibitor of rhomboid 5 homolog 2 (Rhbdf2), and we further determine that Trim31 directly binds to Rhbdf2 and facilitates its proteasomal degradation. Hepatocyte-specific Trim31 ablation facilitates NAFLD-associated phenotypes in mice. Inversely, transgenic or ex vivo gene therapy-mediated Trim31 gain-of-function in mice with NAFLD phenotypes virtually alleviates severe deterioration and progression of steatohepatitis. The current findings suggest that Trim31 is an endogenous inhibitor of Rhbdf2 and downstream cascades in the pathogenic process of steatohepatitis and that it may serve as a feasible therapeutical target for the treatment of NAFLD/NASH and associated metabolic disorders.

[1] Chongqing Key Laboratory of Medicinal Resources in the Three Gorges Reservoir Region, School of Biological and Chemical Engineering, Chongqing University of Education, 400067 Chongqing, PR China. [2] Research Center of Brain Intellectual Promotion and Development for Children Aged 0-6 Years, Chongqing University of Education, 400067 Chongqing, PR China. [3] Key Laboratory of Biorheological Science and Technology (Chongqing University), Ministry of Education, College of Bioengineering, Chongqing University, 400030 Chongqing, PR China. [4] Shandong Cancer Hospital and Institute, Shandong First Medical University & Shandong Academy of Medical Sciences, 250117 Jinan, PR China. [5] Department of Experimental Center, School of Biological and Chemical Engineering, Chongqing University of Education, 400067 Chongqing, PR China. [6] The Laboratory of Cell Biochemistry and Topogenetic Regulation, College of Bioengineering and Faculty of Sciences, Chongqing University, 400067 Chongqing, PR China. [7] These authors contributed equally: Minxuan Xu, Jun Tan, Wei Dong, Benkui Zou, Xuepeng Teng, Liancai Zhu, Chenxu Ge. ✉email: tanjun@cque.edu.cn; zcf18866126280@163.com; wangbc2000@126.com

The latest epidemiological studies have found that obesity and its metabolic complications have become one of the most serious public health crises[1]. With the increase in high-risk obesity populations, continuous and prolonged high-energy diet intake increases the risk of systemic metabolic syndrome, including obesity, hyperlipidemia, and severe cardiovascular and cerebrovascular diseases[2–4]. In fact, the prevalence of obesity-related comorbidities such as non-alcoholic fatty liver disease (NAFLD), has increased in parallel, and NAFLD is already the most common chronic liver disease[1,3,5]. The severity of NAFLD ranges from simple steatosis to hepatocyte injury with malignant transformation and necrotizing inflammatory alterations characterized as nonalcoholic steatohepatitis (NASH), which renders patients more susceptible to liver fibrosis and hepatocellular carcinoma[6,7]. Importantly, the formed NASH, in turn, will further promote its complications, e.g., insulin resistance, hypertension, chronic kidney disease (CKD), and type 2 diabetes (T2D)[8,9]. Thus far, unfortunately, there are no approved effective therapeutic strategies for NASH in the world, and methods to alleviate the related complications induced by this disease do not fully meet expectations. The development of effective drugs and therapeutic options for NAFLD/NASH mainly depends on targeting pivotal signaling regulators or regulatory processes closely related to pathogenic mechanisms. Given that the pathological process of NAFLD and NASH is a tangled progression associated with metabolic syndrome and systemic inflammatory response syndrome (SIRS)[9–11], the common targets should be identified to manipulate the pathogenic signaling pathways.

Rhomboid 5 homolog 2 (Rhbdf2), also known as iRhom2, is an inactive member of the rhomboid intramembrane proteinase family that has been determined to be a pivotal pathogenic regulator of inflammation-related diseases, e.g., obesity, arthritis, nephritis, atherosclerosis, and fibrosis[12–15]. Our previous study also confirmed that Rhbdf2 recruits MAP3K7 to significantly increase its phosphorylation levels and activate downstream inflammatory signaling[15]. The activation of Rhbdf2-MAP3K7 signaling contributes to the occurrence of NAFLD, which predisposes pathological phenotypes to hepatic fibrosis. Meanwhile, Rhbdf2 has the ability to be regulated by ubiquitination, another important regulatory modification[16]. A previous study speculated that the half-life of Rhbdf2 is significantly prolonged in the presence of the proteasomal inhibitor MG-132, further indicating that there may be K48-linked ubiquitination sites in the cytoplasmic domain, and therefore proteasomal degradation[17]. However, it is still unknown whether there are certain regulatory factors that control the ubiquitination of Rhbdf2 and Rhbdf2-mediated pathogenesis of NAFLD/NASH. Accordingly, it is essential to confirm the regulatory signaling pathway resulting in Rhbdf2-MAP3K7 suppression or activation in NAFLD/NASH.

E3 ubiquitin ligase-tripartite motif-containing protein 31 (Trim31) has been identified as a "Janus-faced" regulator of innate immune responses by facilitating the targeted substrate degradation or signal transduction via ubiquitin modification[18–20]. Animals with dysfunctional Trim31 exhibit severe intestinal inflammation and imbalance of intestinal flora, which induces an endotoxemia phenotype and elevates NLRP3 inflammasome activation[21,22]. By contrast, Trim31 markedly promotes myocardial dysfunction by facilitating apoptosis and NF-κB signaling in sepsis[18]. Trim31-mediated invasion and metastasis in colorectal cancer are enhanced by regulating of NF-κB signaling-associated chronic inflammation[23]. These seemingly contradictory findings revealed that Trim31 virtually has completely distinct functions in different pathological processes. Moreover, the role of Trim31, especially in the pathology of NAFLD/NASH, remains unknown. These studies have prompted us to explore the molecular mechanism of Trim31 in depth.

In the current report, we revealed that Trim31 mitigates genetically and high-energy diet-triggered insulin resistance, liver steatosis, inflammation, and hepatic fibrosis by promoting degradation of Rhbdf2 by K48-linked polyubiquitination, which results in suppression of Rhbdf2-MAP3K7 signaling and downstream events. Our results indicate that Trim31 is a key suppressor of NAFLD/NASH and metabolic disorders and may serve as a molecular target for the treatment of these diseases.

## Results

**Trim31 expression is downregulated in livers with hepatic steatosis.** To determine whether Trim31 is involved in hepatic steatosis and metabolism, we first investigated its expression levels in liver tissues isolated from both dietary and obese mice models with steatohepatitis. We found that Trim31 protein expression was significantly lower in liver samples of mice fed a high-fat diet (HFD) for 16 weeks and *ob/ob* mice than in mice fed a standard normal chow diet (NCD) and lean controls, accompanied by increased Rhbdf2 expression levels (Fig. 1a, b and Supplementary Fig. S1a). Moreover, the dynamic expression levels from 0 to 16 weeks after HFD treatment indicated that Trim31 expression was gradually suppressed in the fatty liver (Fig. 1c and Supplementary Fig. S1b, c). Meanwhile, in HFD-fed mice model, liver Trim31 mRNA expression levels were negatively correlated with the liver Rhbdf2, serum ALT and AST contents, liver TG, TC and NEFA levels, and NAS score in liver samples by Pearson correlation analyses and corresponding obesity and fatty liver-related indicators analysis (Supplementary Fig. S2a–i). In addition, the additional multiple linear regression and Pearson multiple correlation analyses further confirmed the negative correlation of liver Trim31 expression with fatty liver severity (Supplementary Fig. S2j–l). Next, we examined its expression profile in liver samples of NAFLD and NASH patients. In patients with NAFLD or NASH, we found that Trim31 levels were drastically reduced, but hepatic Rhbdf2 levels were increased, compared to the levels in non-steatosis samples. Of note, significantly lower expression levels of TRIM31 were observed in the livers from NASH patients than in those from patients with only simple steatosis (Fig. 1d and Supplementary Fig. S1d). In addition, to explore the mechanism underlying the decrease of Trim31 expression, we further investigated its expression profile in vitro. Previous studies indicated that increased release of free fatty acids from adipose cells results in triacylglycerol (TG) accumulation in hepatocytes, which may progress into steatosis and steatohepatitis[13,15]. It was also have shown that saturated fats, including palmitic acid (PA) and stearic acid, are more toxic than other fatty acids and promote hepatocyte toxicity in steatohepatitis models. Significant deposition of TG in the liver was caused by a disturbance of the lipid metabolism equilibrium. The metabolic disturbance not only promoted steatosis and hepatocyte injury, but it also markedly suppressed insulin signaling and facilitated insulin resistance[13,15]. Thus, PA was used in the in vitro experiments. In cultured primary mouse hepatocytes, we observed that administration with PA and TNF-α markedly reduced the Trim31 protein expression levels, accompanied by elevated Rhbdf2 expression levels (Fig. 1e and Supplementary Fig. S1e). Also, the levels of reactive oxygen species (ROS), an efficient pro-steatotic inducer, have been confirmed to increase in several established animal models. Thus, the ROS scavenger N-Acetyl-L-cysteine (NAC) and the antioxidant catalase (CAT) were used in these in vitro experiments. Indeed, suppression of ROS levels significantly restored the

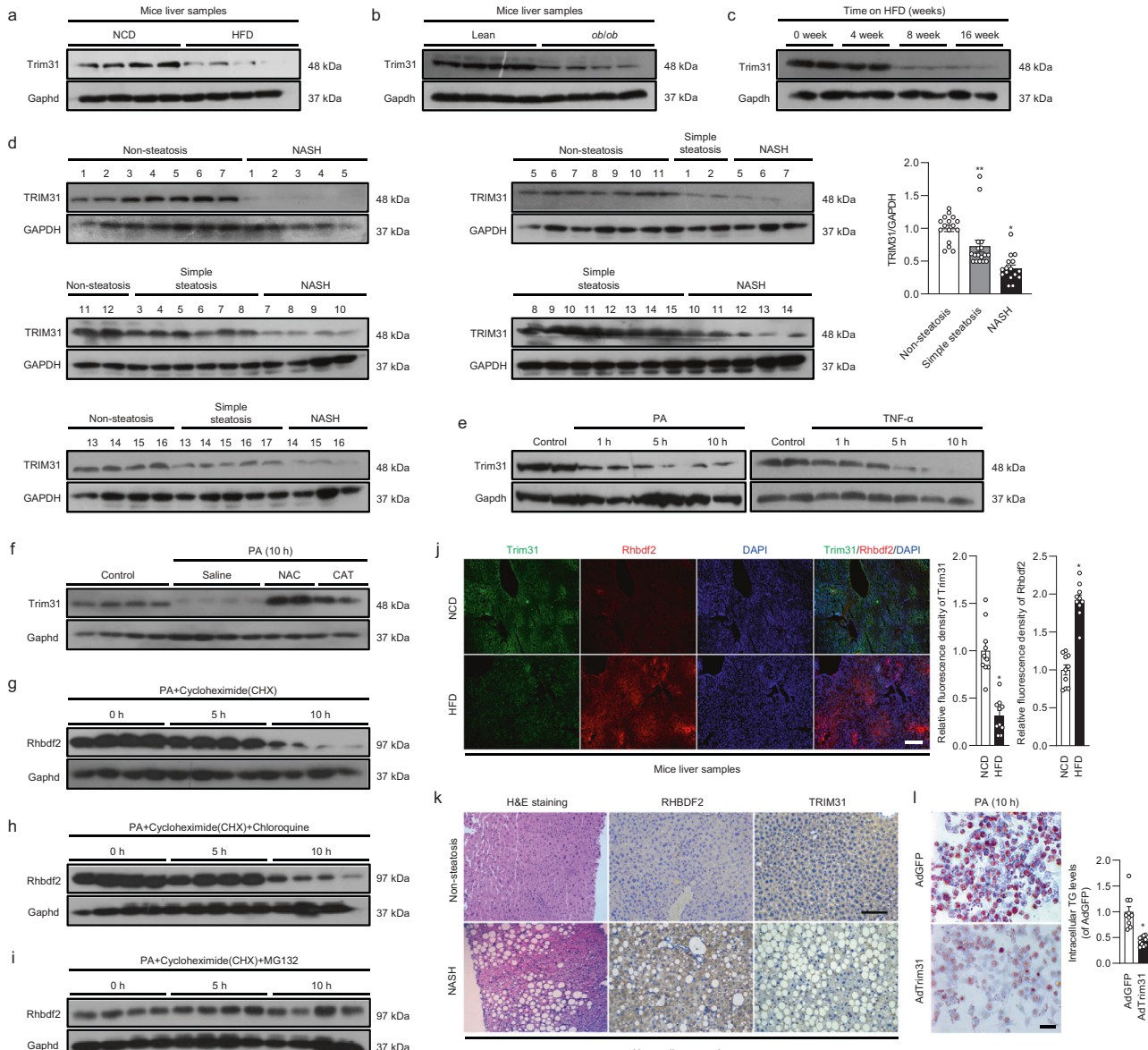

**Fig. 1 Trim31 expression is restrained in the livers with hepatic steatosis. a, b** Representative immunoblotting bands of Trim31 expression in the liver samples isolated from C57BL/6 N mice that were treated with a NCD or HFD for uninterrupted 16 weeks (**a**) or from *ob/ob* or lean mice (**b**) (*n* = 4 per experiment). **c** Representative immunoblotting bands of Trim31 expression in the liver samples isolated from C57BL/6N mice that were treated with a HFD over time (*n* = 4 per experiment). **d** Representative immunoblotting bands and relative expression levels of TRIM31 in the liver samples of donors with non-steatosis (*n* = 16 samples), simple steatosis (*n* = 17 samples) or NASH (*n* = 16 samples) phenotype (**P < 0.01 vs. non-steatosis groups, *P < 0.05 vs. simple steatosis groups). **e–i** Representative immunoblotting bands of Trim31 or Rhbdf2 expression in cultured primary hepatocytes that were incubated with 400 μM palmitate (PA) or 100 ng/ml TNF-α (**e**), respectively; with PA in combination with 10 μM *N*-acetyl-L-cysteine (NAC) or 400 U/ml catalase (CAT) (**f**); or with PA and protein synthesis inhibitor cycloheximide (CHX) (**g**) in combination with 20 μM chloroquine (**h**) or 10 μM MG132 (**i**). The BSA or saline was treated as controls (*n* = 4 per experiment). (**j**) Representative immunofluorescence images of Trim31 and Rhbdf2 co-expression in mice liver tissue isolated from C57BL/6 N mice that were treated with a NCD or HFD for 16 weeks (magnification, ×40; *n* = 10 images per group for each staining) (*P < 0.01 vs. NCD group). **k** Representative images of hematoxylin–eosin (H&E)-stained pathological section, and immunohistochemical staining of TRIM31 or RHBDF2 expression in patients with NASH (magnification, ×100; *n* = 12 images per group for each staining). **l** Intracellular triglyceride (TG) analysis with the Oil-red O staining of adenovirus-loading full-length Trim31 sequences (AdTrim31)-transfected L02 cells under PA treatment for 10 h. The adenovirus-containing GFP vector (AdGFP) was used as controls (magnification, ×100; *n* = 10 images per group for each staining) (*P < 0.01 vs. AdGFP group). Data are expressed as mean ± SEM. The relevant experiments presented in this part were performed independently at least three times. Significance determined by one-way analysis of variance (ANOVA) followed by Dunnett's multiple comparisons test (**d**) and Student's two-tailed *t* test analysis (**j, l**).

decrease of PA-triggered Trim31 expression (Fig. 1f), suggesting an antagonistic role of ROS in Trim31 function. Importantly, our previous study also indicated that strong inactivation of Trim31 was observed in inflammation-related diseases, accompanied by a significant increase of Rhbdf2 activity[15]. Also, Rhbdf2 has been shown to be regulated by ubiquitination modification[16]. Consistent with these studies, we confirmed that Rhbdf2 levels were significantly increased in livers of human patients with NASH and simple steatosis phenotypes, as compared to the levels in non-steatosis samples. Meanwhile, higher expression levels of

Rhbdf2 were further observed in the livers of NASH patients than in the livers of patients with only simple steatosis (Supplementary Fig. S1d). Additionally, cycloheximide (CHX), which functions as a protein synthesis inhibitor in eukaryotes, was used to determine the inhibition of Rhbdf2 protein expression upon PA treatment. Indeed, compared to the control group (0 h), the increase of Rhbdf2 at 5 h was inhibited, and then reduced at 10 h. Rhbdf2 degradation was dramatically restrained during co-treatment with the proteasome inhibitor MG132, whereas incubation with chloroquine (a kind of lysosome inhibitor) had an inappreciable effect (Fig. 1g–i). These findings indicated that Rhbdf2 degradation occurred by the ubiquitin-proteasome pathway. Virtually all Rhbdf2 was ubiquitinated in PA-treated primary hepatocytes, and the effect was counteracted after ROS was obliterated (Supplementary Fig. S1f). The changes in Rhbdf2 (RHBDF2) and Trim31 (TRIM31) expression in fatty liver were further supported by immunofluorescence analysis of mice livers (Fig. 1j), immunohistochemical analysis of human liver sections (Fig. 1k), and analysis of intracellular TG levels in adenovirus-mediated Trim31 expression (AdTrim31)-transfected L02 cells (Fig. 1l). Based on the results, we sought to explore the molecular mechanism by which Rhbdf2 and Trim31 play associated roles in fatty liver. More in-depth results were obtained in the following experiments.

**Trim31 ameliorates insulin resistance and abnormal glucose metabolism.** Given the tight correlation of Trim31 with fatty liver, we constructed a series of mice models to explore the role of Trim31 in the regulation of the major hallmarks of insulin resistance and glycometabolic disorder. We established hepatocyte-specific Trim31 knockout (THKO) mice (Supplementary Fig. S3a, b) and hepatocyte-specific Trim31 overexpression (THTG) mice (Supplementary Fig. S4a, b) to examine the protective effects of Trim31 on HFD-induced insulin resistance and glucose metabolism disorder, which are common complications and vital predisposing factors of fatty liver. As expected, Trim31 dysfunction markedly increased the prolonged HFD-triggered elevations in their body weight, fasting blood glucose levels, fasting insulin levels, and HOMA-IR index (Fig. 2a–d), but these were alleviated by Trim31 overexpression (Supplementary Fig. S5a–d). Of note, there was no marked difference in food intake between the NCD-treated and HFD-treated THKO (Supplementary Fig. S3c) and THTG mice (Supplementary Fig. S4c). THKO mice had significantly higher blood glucose levels than their Trim31$^{flox/flox}$ littermate controls, as determined by record from the glucose tolerance test (GTT) and insulin tolerance test (ITT) (Fig. 2e, f), whereas the glucose levels recorded in all the tests were remarkably reduced in THTG mice (Supplementary Fig. S5e, f). Accordingly, by using the glycogen detection kit, we found that the decrease of liver glycogen reserve was significantly accelerated by Trim31 deletion (Fig. 2g) but restrained by Trim31 overexpression (Supplementary Fig. S5g). Moreover, liver Trim31 upregulation markedly reduced the mRNA and protein expression profiles associated with gluconeogenesis-related indicators including glucose-6-phosphatase (G6Pase) and phosphoenolpyruvate carboxykinase (PEPCK), but increased phosphorylated AKT (p-AKT), glycogen synthase kinase 3β (p-GSK3β), and forkhead box O1 (p-FOXO1) levels. Also, dysregulation of liver insulin signaling caused by HFD administration was promoted in the THKO mice, but was dramatically restored in the THTG mice, as compared to those in the controls (Fig. 2h and Supplementary Fig. S5h). The obtained results revealed that liver Trim31 is a crucial suppressor of HFD-triggered insulin resistance and abnormal glucose metabolism.

**Trim31 protects against liver steatosis and inflammation.** Based on the effective and consistent protective effect of Trim31 on insulin resistance and glycometabolic disorder[21], we next investigated the role of Trim31 in improving the main features of liver steatosis and inflammation. Unsurprisingly, long-term HFD-fed THKO mice exhibited remarkable increases in liver weight and the ratio of liver weight to body weight (LW/BW), as compared to those in HFD-fed corresponding littermate controls (Fig. 3a). Meanwhile, the deposition of liver lipids, as determined by liver observations, transmission electron microscopy (TEM) analysis, hematoxylin and eosin (H&E) staining, Oil red O staining, and measurements of the levels of TG, total cholesterol (TC), non-esterified fatty acids (NEFA), and the liver function-related indicators serum alanine transaminase (ALT), aspartate aminotransferase (AST), and alkaline phosphatase (AKP), was visibly enhanced in HFD-treated THKO mice (Fig. 3b–d). Functional loss of Trim31 in the liver was also involved in markedly upregulated expression of fatty acid uptake and synthesis-related genes and with dramatically downregulated expression of fatty acid β-oxidation-related genes in HFD-fed THKO mice (Fig. 3e). In THTG mice, the opposite observations were made. HFD-fed THTG mice displayed decreases in liver weight and in the LW/BW ratio, compared to HFD-fed non-transgenic (NTG) controls (Fig. 3f). Then, by a similar approach, we found that hepatic lipid accumulation and hepatic function-associated parameters including TG, TC, NEFA, ALT, AST, and AKP were significantly reduced in HFD-fed THTG mice (Fig. 3g–i). Likewise, the expression of fatty acid uptake- and synthesis-related genes was decreased and the expression of fatty acid β-oxidation-related genes was accordingly increased in THTG mice compared to the NTG HFD group (Fig. 3j). The upregulated effect of Trim31 dysfunction on lipid deposition was also supported by results from palmitate (PA)-treated primary hepatocytes (Fig. 3k and Supplementary Fig. S4d). Of note, fat accumulation in the viscera and subcutaneous fat tissues is always tightly involved in and links to metabolic syndrome[24]. Indeed, the viscera and subcutaneous fat tissues were elevated in the HFD-fed THKO mice, but this elevation was significantly restrained in THTG mice with a marked change in the ratio of visceral fat weight to body weight and subcutaneous fat weight to body weight (Supplementary Fig. S6a). Meanwhile, the size of adipose cells was greatly increased in HFD-fed mice compared with Flox mice, but no significant difference was observed between the HFD-fed THKO and THTG groups (Supplementary Fig. S6b). This result further suggested that adipocyte proliferation might be linked to the higher visceral fat weight in HFD-fed THKO mice than those in the HFD-fed Flox mice.

To further examine whether Trim31 participates in the regulation of liver inflammation, we investigated the activity of inflammatory signal pathways and corresponding alterations in the expression profiles of inflammation-associated cytokines and chemokines. Indeed, HFD-triggered overexcitation of inflammation-related signaling was dramatically enhanced in THKO mice, but this effect was reversed in THTG groups, as indicated by the changes in protein expression of phosphorylated IKKβ, IκBα and NF-κB, and changes in mRNA expression of inflammation-associated indicators (Supplementary Fig. S7a, b). In addition, the contents of inflammatory mediators including TNF-α, IL-1β, IL-6, and CCL-2 were greatly higher in serum of THKO mice but reduced in the serum of THTG mice compared with controls (Supplementary Fig. S7c). Meanwhile, similar findings were further observed in the PA-treated cultured primary hepatocytes, as revealed by activation of NF-κB-mediated inflammatory signaling (Supplementary Fig. S7d). The above results illustrated that liver Trim31 is an important inhibitor of HFD-triggered inflammation and liver lipid deposition in mice.

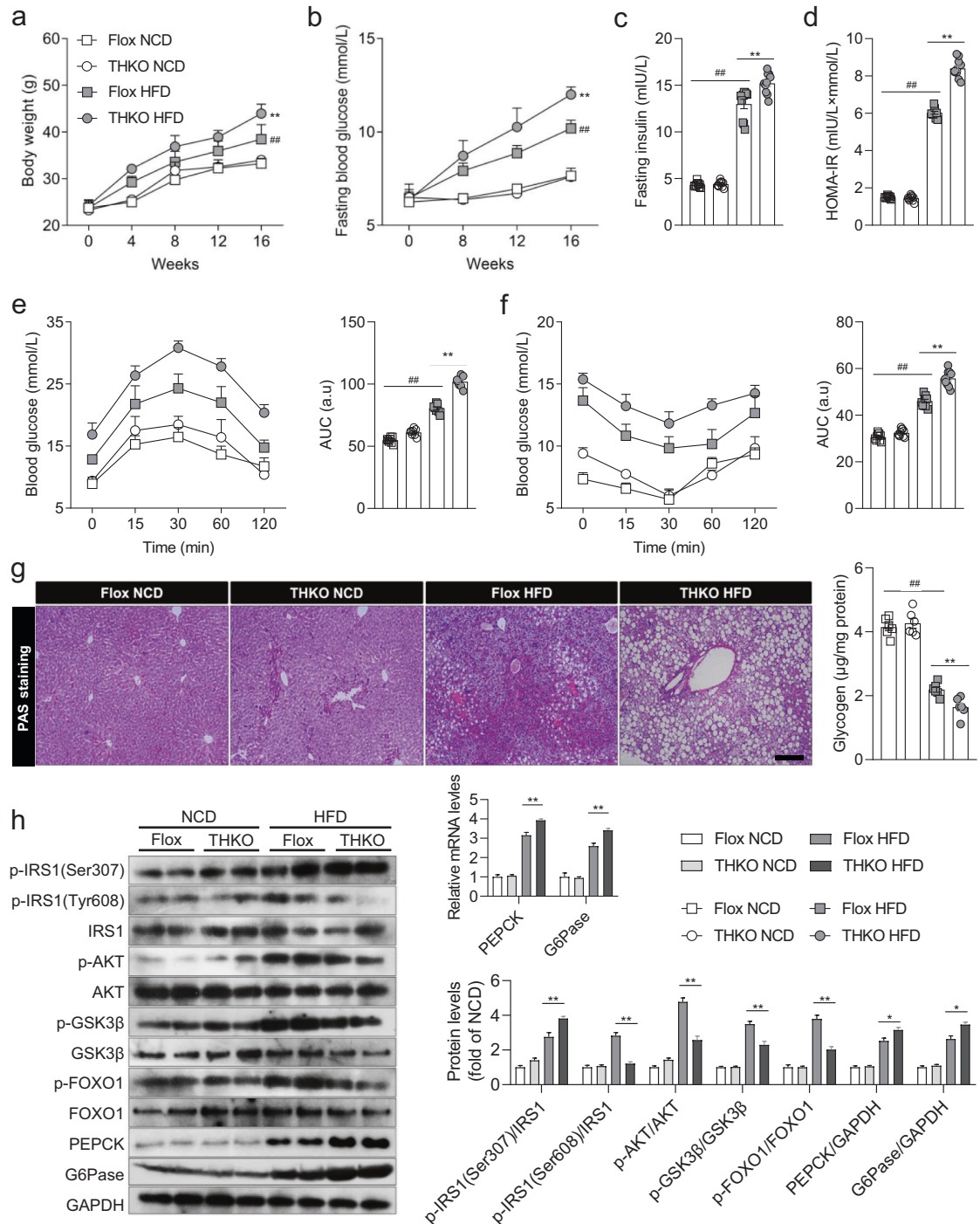

**Fig. 2 Hepatocyte-specific Trim31 deletion aggravates HFD-triggered pathological phenotypes of insulin resistance. a**, **b** Body weight records (**a**) and fasting blood glucose levels (**b**) of Trim31 flox (Flox) and hepatocyte-specific Trim31 deletion (THKO) mice during 16 weeks of NCD or HFD administration ($n = 15$ mice) (**P < 0.01 vs. Flox HFD groups, ##P < 0.01 vs. Flox NCD groups). **c**, **d** Fasting insulin levels (**c**) and corresponding HOMA-IR index (**d**) of Flox and THKO mice at the last week of NCD or HFD feeding ($n = 10$ mice) (**P < 0.01 vs. Flox HFD groups, ##P < 0.01 vs. Flox NCD groups). **e**, **f** Records for the glucose tolerance test (GTT) (**e**) and insulin tolerance test (ITT) (**f**) in the Flox and THKO mice at the last week of NCD or HFD ingestion; a.u, arbitrary unit ($n = 10$ mice) (**P < 0.01 vs. Flox HFD groups, ##P < 0.01 vs· Flox NCD groups). **g** Representative images of Periodic Acid-Schiff stain (PAS)-stained pathological section for glycogen storage changes in the liver samples from the Flox and THKO mice (magnification, ×100; $n = 6$ images per group for each staining) (**P < 0.01 vs. Flox HFD groups, ##P < 0.01 vs. Flox NCD groups). **h** Representative immunoblotting bands for expression alterations of total amounts and phosphorylated forms of critical indicators involved in insulin signaling, including IRS1, AKT, GSK3β, FOXO1, PEPCK, and G6Pase, in the liver of Flox and THKO mice after HFD feeding for 16 weeks ($n = 4$ per experiment) (*P < 0.05 and **P < 0.01 vs. Flox HFD groups, ##P < 0.01 vs. Flox NCD groups). The GAPDH was used as a loading control. Data are expressed as mean ± SEM. The relevant experiments presented in this part were performed independently at least three times. Significance determined by two-way analysis of variance with general linear model procedures using a univariate approach.

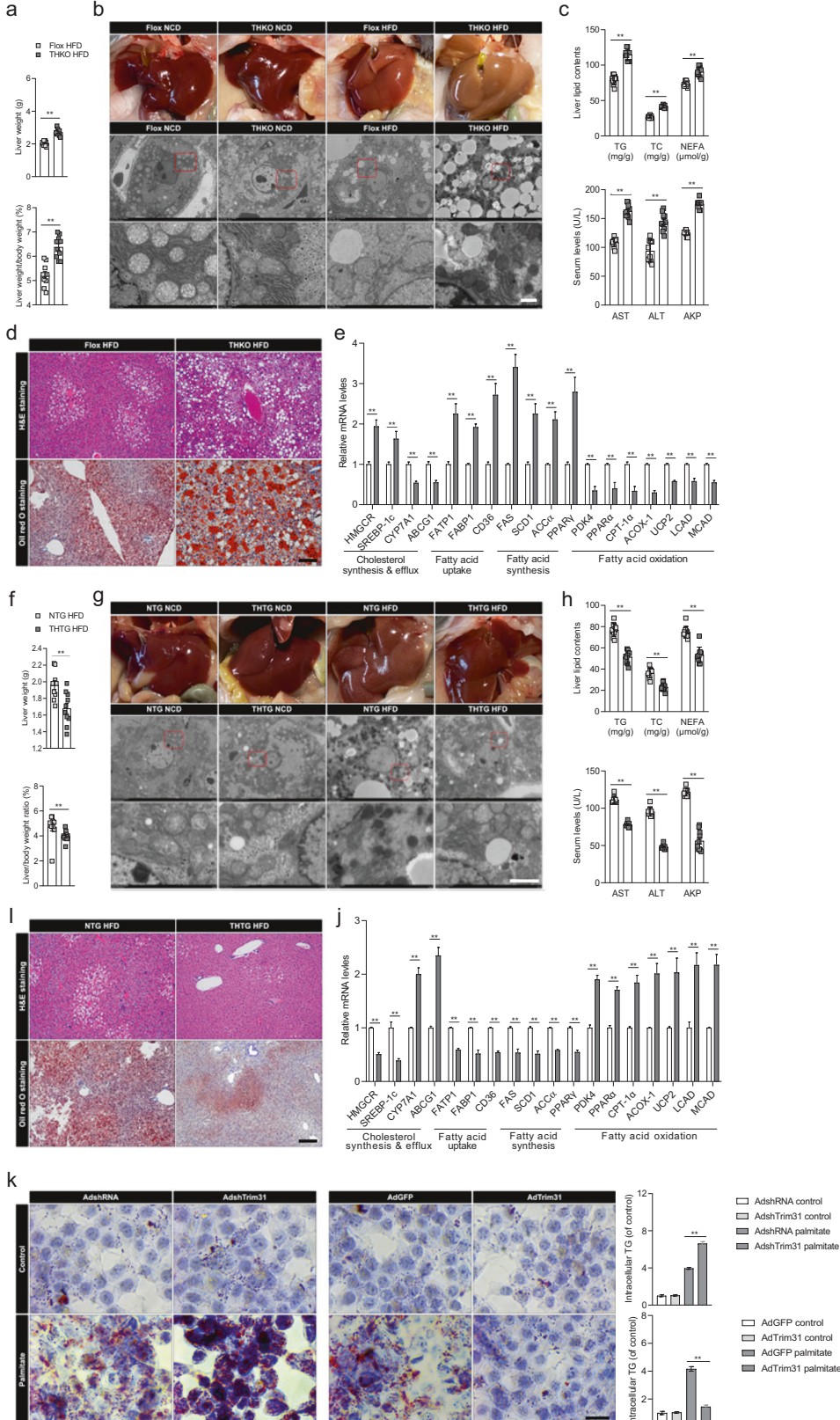

**Trim31-mediated ex vivo gene therapy facilitates the mitigation of fatty liver**. To better confirm the inhibitory effect of Trim31 on steatohepatitis, the ex vivo gene therapy intervention approach by lentivirus loaded with mouse full-length Trim31 sequences (LV-Trim31) or human full-length Trim31 sequences (LV-hTRIM31) was employed to further evaluate the role of Trim31 on mitigation of

liver steatosis, insulin resistance and inflammation (Fig. 4a and Supplementary Fig. S8a). Unsurprisingly, using ex vivo cultured and transduced hepatocytes, the mice with hepatocyte-specific Trim31 gain-of-function (THKO)(LV+) exhibited a remarkable decrease in HFD-induced liver weight, body weight, LW/BW ratio, AST and ALT levels, and hepatic lipid deposition, compared to the

**Fig. 3 Functional loss of hepatic Trim31 aggravates HFD-induced hepatic steatosis. a** Records for the liver weight (upper) and the ratio of liver weight/ body weight (lower) (%) of the Flox and THKO mice at the last week of HFD treatment ($n = 10$ mice per group) (**$P < 0.01$ vs. Flox HFD groups). **b** Representative pictures for liver appearance and transmission electron microscope (TEM)-indicated histological changes of the liver in Flox and THKO mice after NCD or HFD feeding for 16 weeks (Scale bar, 10 μm for upper image, 2 μm for the lower image; $n = 10$ images per group for each group) (**$P < 0.01$ vs. Flox HFD groups). (**c**) Liver lipid contents including triglyceride (TG), total cholesterol (TC) and non-esterified fatty acids (NEFA) (upper), and serum alanine transaminase (ALT), aspartate aminotransferase (AST), and alkaline phosphatase (AKP) levels (lower) of the Flox and THKO mice after HFD treatment for 16 weeks ($n = 10$ mice per group) (**$P < 0.01$ vs. Flox HFD groups). **d** Representative pictures of H&E-stained (upper) and Oil-red O-stained (lower) pathological section of the liver from the 16-week HFD-fed Flox and THKO mice (magnification, ×100; $n = 10$ images per group for each staining) (**$P < 0.01$ vs. Flox HFD groups). **e** qPCR analysis of the relative mRNA expression of genes associated with fatty acid uptake, synthesis, and β-oxidation in Flox and THKO mice after 16-week HFD feeding ($n = 10$ liver samples per group) (**$P < 0.01$ vs. Flox HFD groups). **f** Records for the liver weight (upper) and the ratio of liver weight/body weight (lower) (%) of the non-transgenic (NTG) mice and hepatocyte Trim31 transgenic (THTG) mice at the last week of HFD treatment ($n = 10$ mice per group) (**$P < 0.01$ vs. NTG HFD groups). **g** Representative pictures for liver appearance and TEM-indicated histological changes of the liver in NTG and THTG mice after 16-week NCD or HFD feeding (Scale bar, 10 μm for upper image, 2 μm for the lower image; $n = 10$ images per group for each group) (**$P < 0.01$ vs. NTG HFD groups). **h** Liver lipid contents including TG, TC, and NEFA (upper), and serum ALT, AST, and AKP levels (lower) of the NTG and THTG mice after HFD treatment for 16 weeks ($n = 10$ mice per group) (**$P < 0.01$ vs. NTG HFD groups). **i** Representative pictures of H&E-stained (upper) and Oil-red O-stained (lower) pathological section of the liver from the 16-week HFD-fed NTG and THTG mice (magnification, ×100; $n = 10$ images per group for each staining) (**$P < 0.01$ vs. NTG HFD groups). **j** qPCR analysis of the relative mRNA expression of genes associated with fatty acid uptake, synthesis, and β-oxidation in NTG and THTG mice after 16-week HFD feeding ($n = 10$ liver samples per group) (**$P < 0.01$ vs. NTG HFD groups). **k** Representative pictures of Oil red O staining of primary hepatocytes that were transfected with AdTrim31 or AdshTrim31 and/or treated with corresponding controls or PA for 10 h (magnification, ×200; $n = 10$ images per group for each staining) (**$P < 0.01$ vs. AdshRNA palmitate groups (upper) and AdGFP palmitate groups (lower)). Data are expressed as mean ± SEM. The relevant experiments presented in this part were performed independently at least three times. Significance determined by Student's two-tailed $t$ test analysis.

corresponding controls (THKO)(LV−) (Fig. 4b–d). Furthermore, the THKO(LV+) mice also had lower blood glucose levels than those of THKO(LV−) controls, as confirmed by results from the GTT and ITT test (Fig. 4e, f). Furthermore, the high TG, TC, and NEFA contents in the liver were greatly decreased in THKO(LV+) mice (Fig. 4g). Meanwhile, restoration of Trim31 expression in hepatocytes significantly reduced the expression of genes involved in fatty acid uptake and synthesis and promoted the expression of genes involved in fatty acid β-oxidation in HFD-fed THKO(LV+) mice (Fig. 4h). Also, the impaired insulin signaling and activated inflammation-associated signaling were markedly altered in THKO(LV+) mice compared to controls, as determined by immunoblotting, mRNA expression analysis, and measurements of the concentrations of pro-inflammatory mediators (Fig. 4i,j and Supplementary Fig. S8b, c). Consistent with the above data, the ex vivo gene therapy by LV-hTRIM31 further demonstrated that hepatocytes with human TRIM31 restoration significantly alleviated HFD-induced insulin resistance, liver steatosis and inflammation (Supplementary Fig. S9a–k). These results might be partly and possibly extrapolated to human pathophysiology, or provide some evidence for pathological research.

**Inactivation of the Rhbdf2–MAP3K7 axis by Trim31 suppresses NAFLD progression.** Given the consistent and remarkable prohibitive function of Trim31 on fatty liver and its associated pathological phenotypes, the above results prompted us to study the molecular mechanism of Trim31 and its intrinsic function. Because Rhbdf2 is ubiquitinated, and it plays a key role in promoting the development of NAFLD and NASH phenotype, we then investigated the influence of Trim31 on Rhbdf2 and its downstream events components. As expected, significant activation of Rhbdf2–MAP3K7 axis induced by HFD treatment was boosted in the THKO mice, but was greatly repressed in the THTG mice, as indicated in immunofluorescence analysis of tissue sections (Fig. 5a). Likewise, Rhbdf2-MAP3K7 signaling members including ADAM17, TNFR1/2, MKK7, and c-Jun, were also determined to be involved in fatty liver progression. The levels of ADAM17, TNFR1/2, and p-MAP3K7 and its downstream component MKK7 were upregulated by Trim31 ablation but markedly inhibited by Trim31 overexpression in vivo and in vitro (Fig. 5b, c).

To further examine the regulatory effect of Trim31 on the function of Rhbdf2, we next constructed an impaired insulin signaling model in PA-induced isolated primary hepatocytes. The adenovirus-packed full-length Rhbdf2 sequence (AdRhbdf2) and shRNA targeting Rhbdf2 (AdshRhbdf2) were used to overexpress and inactivate Rhbdf2, respectively, in Trim31-specific deletion or Trim31-transgenic hepatocytes. Indeed, impairment of insulin signaling stimulated by PA was significantly deteriorated in AdRhbdf2-transfected THKO-hepatocytes but was virtually assuaged in AdshRhbdf2-transfected THKO-hepatocytes, as compared to controls. In contrast to this, however, the THTG-hepatocytes transfected with AdshRhbdf2 showed a decreasing trend in PA-induced impaired insulin signaling (i.e., alteration of p-AKT, p-GSK3β, and p-FOXO1 levels) compared with AdRhbdf2-transfected THTG-hepatocytes (Fig. 5d and Supplementary Fig. S10a). The above data further demonstrated that dysfunction of the Rhbdf2-MAP3K7 signaling pathway is essential for the protective effect of Trim31 against hepatic steatosis.

**Trim31 directly interacts with Rhbdf2.** The observed effects of Trim31 on Rhbdf2 signaling-associated hepatic steatosis, inflammation, and insulin resistance prompted us to examine whether Trim31 directly interacts with Rhbdf2 during the development of NAFLD. To answer this question, an immuno-precipitation assay was employed. In vitro interaction tests suggested that exogenically expressed TRIM31 could directly bind to RHBDF2 and vice versa (Fig. 6a). Subsequently, a glutathione S-transferase (GST)-tagged RHBDF2 significantly pulled down TRIM31, and GST-tagged TRIM31 also pulled down RHBDF2 (Fig. 6b). These data strongly suggest that TRIM31 directly interacts with RHBDF2. Importantly, as a key member of the E3 ubiquitin ligase family, TRIM31 is mainly composed of three parts: an N-terminal RING-finger domain, a B-Box domain, and a C-terminal coiled-coil (CC) domain (Fig. 6c). To confirm which domain of TRIM31 is responsible for the interaction with RHBDF2, a series of vectors encoding Flag-tagged truncated TRIM31 mutants, including wild-type (TRIM31-Flag WT), a RING-finger domain ablation mutant (TRIM31-Flag RING$^\Delta$), a B-Box domain deletion mutant (TRIM31-Flag Box$^\Delta$), and a CC domain deletion mutant (TRIM31-Flag CC$^\Delta$), were generated for the following binding experiments. The co-immunoprecipitation

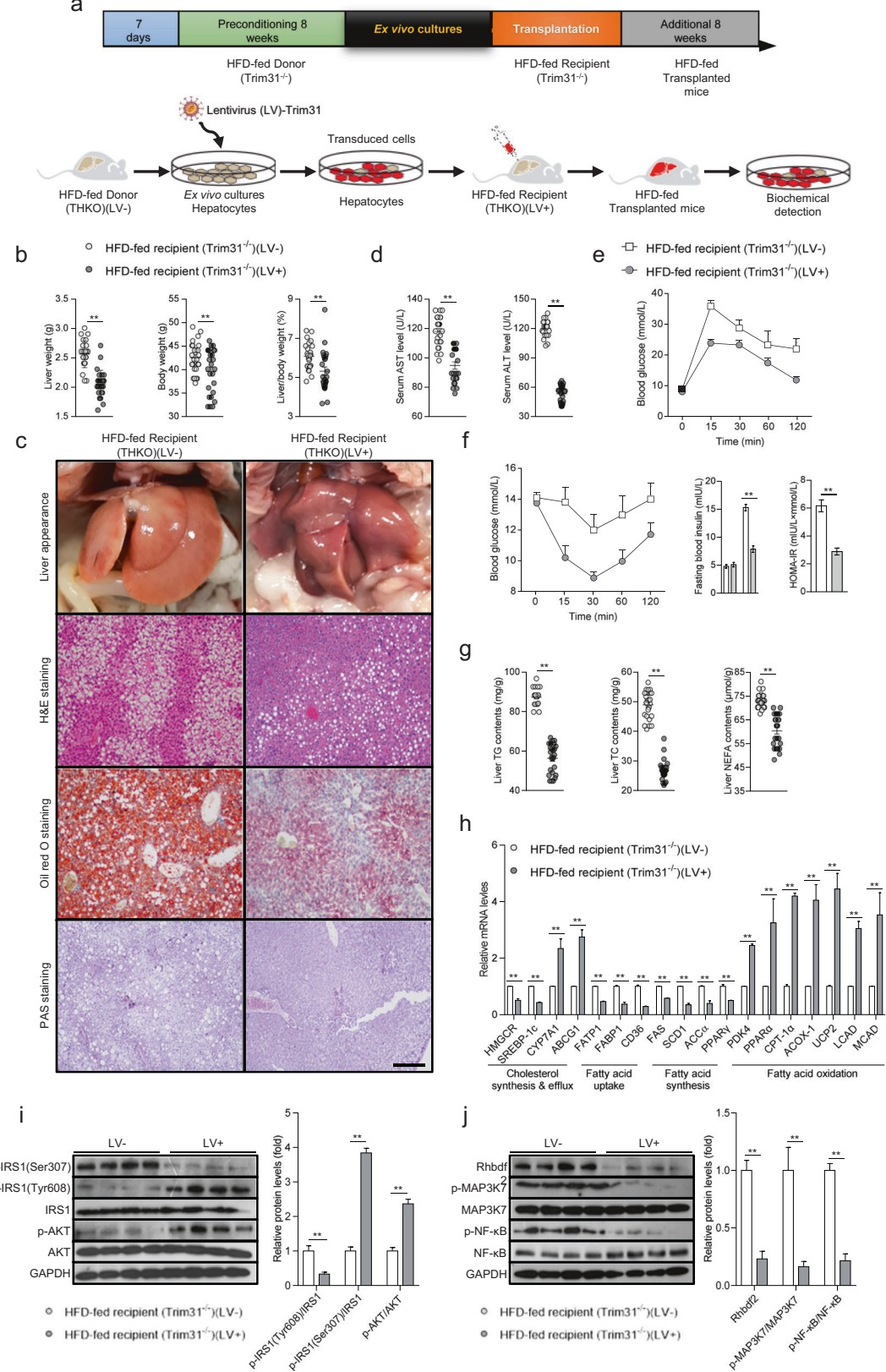

assays demonstrated that RHBDF2 co-precipitated with TRIM31 WT, TRIM31 RING$^\Delta$, and TRIM31 Box$^\Delta$. These results indicated that the CC domain contributes to binding to RHBDF2. Next, three vectors encoding HA-tagged truncated RHBDF2 mutants, including wild-type (RHBDF2-HA WT), a

transmembrane domain (TMD) deletion mutant (RHBDF2-HA TMD$^\Delta$), an inactive rhomboid homology domain (IRHD) deletion mutant (RHBDF2-HA IRHD$^\Delta$), and an N-terminal cytoplasmic tail (Tail) domain deletion mutant (RHBDF2-HA Tail$^\Delta$), were also generated. Co-immunoprecipitation detection with

**Fig. 4 Allogeneic hepatocyte transplantation using lentivirus-mediated Trim31 expression alleviates HFD-triggered hepatic steatosis, insulin resistance, and inflammation.** Preconditioned liver-specific Trim31 deletion (THKO) mice with an 8-week HFD treatment as donors were transduced (THKO)(LV−). The hepatocytes isolated from the (THKO)(LV−) group mice were transduced with a lentivirus-loaded full-length Trim31 sequence. The corresponding blank vector was transduced as controls. Then the additional HFD-fed THKO mice as recipient were injected with transduced hepatocytes via the portal vein. The HFD-fed transplanted (THKO)(LV+) mice were harvested for further experimental detection. **a** The strategy diagram for ex vivo gene therapy used in the current study. **b** Records for the liver weight, body weight, and the ratio of liver weight/body weight (%) of the HFD-fed recipient (THKO)(LV−) and (THKO)(LV+) mice at the last week of HFD treatment ($n = 25$ mice per group) (**$P < 0.01$ vs. HFD-fed recipient (THKO)(LV−) groups). **c** Representative pictures for liver appearance and Oil red O staining, H&E staining, and PAS staining-indicated liver histopathologic changes of the HFD-fed recipient (THKO)(LV−) and (THKO)(LV+) mice after ex vivo experiment (magnification, ×100; $n = 10$ images per group for each staining). **d** Liver function markers ALT and AST levels were detected in transduced recipient mice ($n = 25$ mice per group) (**$P < 0.01$ vs. HFD-fed recipient (THKO)(LV-) groups). **e, f** Records for the glucose tolerance test (GTT) (**e**) and insulin tolerance test (ITT) (**f**), and the corresponding fasting insulin levels and HOMA-IR index in the HFD-fed recipient (THKO)(LV−) and (THKO)(LV+) mice after transplantation ($n = 25$ mice per group) (**$P < 0.01$ vs. HFD-fed recipient (THKO)(LV−) groups). **g** Liver lipid contents including TG, TC, and NEFA of the transduced mice ($n = 25$ mice per group) (**$P < 0.01$ vs. HFD-fed recipient (THKO)(LV−) groups). **h** qPCR analysis of the relative mRNA expression of genes associated with fatty acid uptake, synthesis, and β-oxidation in the HFD-fed recipient (THKO)(LV−) and (THKO)(LV+) mice after transplantation ($n = 10$ liver samples per group) (**$P < 0.01$ vs. HFD-fed recipient (THKO)(LV-) groups). **i, j** Representative immunoblotting bands for expression alterations of total amounts and phosphorylated forms of critical indicators involved in insulin signaling (**i**), including IRS1, p-IRS1(Ser307), p-IRS1(Tyr608), AKT, and p-AKT, and Rhbdf2−MAP3K7 axis and p-NF-κB pathway (**j**) in the liver of HFD-fed recipient (THKO)(LV−) and (THKO)(LV+) mice ($n = 4$ per experiment) (**$P < 0.01$ vs. HFD-fed recipient (THKO)(LV−) groups). The GAPDH was used as a loading control. Data are expressed as mean ± SEM. The relevant experiments presented in this part were performed independently at least three times. Significance determined by Student's two-tailed $t$ test analysis.

these RHBDF2 mutants revealed that RHBDF2-HA Tail$^\Delta$ has no ability to bind to Trim31. Thus, the N-terminal cytoplasmic tail (Tail) domain of RHBDF2 is essential for the interaction with TRIM31 (Fig. 6c and Supplementary Fig. S10b). In addition, because the CC domain of TRIM31 and the N-terminal (Tail) domain of RHBDF2 in humans are essential for their interaction, we examined whether they have a similar biological function in rodents. Thus, the binding domains of mouse Trim31 and Rhbdf2 were also determined using Trim31-Flag CC$^\Delta$, Rhbdf2-HA Tail$^\Delta$, and their corresponding wild-type vectors. Consistent with the protein binding results, in transfected mice hepatocytes, Trim31 with CC domain deletion did not bind to Rhbdf2 (Supplementary Fig. S10c).

**Trim31 facilitates degradation of Rhbdf2 via K48-linked polyubiquitination.** Previous studies have speculated that the cytoplasmic domain of Rhbdf2 may contain sites for K48-poly-ubiquitination, and therefore proteasomal degradation[17]. We have confirmed Trim31 is a Rhbdf2-related protein (Fig. 6c). Thus, we wondered whether Rhbdf2 could be regulated by Trim31 via its E3 ubiquitin ligase activity. Indeed, a significant increase in Rhbdf2 ubiquitination levels was observed in vitro and in vivo (Fig. 6d and Supplementary Fig. S1f). Also, Rhbdf2 was co-transfected with Myc-ubiquitin and Flag-tagged WT Trim31 into L02 cells. Rhbdf2 ubiquitination levels were greatly enhanced in the presence of a Trim31 expression vector. Of note, the Trim31 expression vector with RING-finger domain ablation mutant fails to catalyze the ubiquitination of Rhbdf2, suggesting the RING-finger domain is required for Trim31 function in the regulation of Rhbdf2 ubiquitination (Fig. 6e). Besides, a previous study has indicated that Rhbdf2 could be modified via K63-linked ubiquitination[16]. Consistent with these results, the endogenous Rhbdf2 was shown to be virtually ubiquitinated with K48 and K63 linkage in PA-challenged L02 cells. As expected, only K48-linked polyubiquitination of Rhbdf2 was dramatically suppressed in Trim31-deficient L02 cells (THKO-L02), but the effect on K63-linked ubiquitination of Rhbdf2 was negligible (Fig. 6f). Furthermore, to accurately investigate the manner of Trim31-regulated Rhbdf2 polyubiquitination, the ubiquitin mutation plasmids K48, K63, K33, K6, K29, and K27 were used for in vitro transfection experiments. The tail label of "O" in K48O, K63O, K33O, K6O, K29O, and K27O reveal ubiquitin in which all lysine residues except themselves were completely mutated. Rather,

Trim31-mediated increase in polyubiquitination of Rhbdf2 could be greatly observed in the presence of K48-loading vector, but not with other vectors (Fig. 6g). It is generally considered that the RING-finger domain mutant of Trim31 has virtually lost its E3 ubiquitin ligase activity. Given the essential role of the RING-finger domain in Trim31 function, indeed, deletion of RING-finger domain of Trim31 not only abolished the polyubiquitina-tion of Rhbdf2, but also impeded the ability of Trim31 to restrain the activation of the Rhbdf2-MAP3K7 pathway and its down-stream signaling cascades (e.g., ADAM17, p-NF-κB, p-IκBα, and p-JNK1/2) (Fig. 6g). The adverse effect of RING-finger domain deficiency on Trim31 function was further supported by the changes in intracellular TG concentrations in PA-stimulated cells in a series of vector-transfected Trim31-deficient L02 cells (THKO-L02) (Fig. 6h). The above data indicated that Trim31-induced degradation of Rhbdf2 by K48-linked polyubiquitination depends on its E3 ubiquitin ligase activity.

**Trim31–Rhbdf2 interaction positively contributes to Trim31-regulated hepatic steatosis and inflammation.** Given the strong correlation of Rhbdf2 activity with Trim31 E3 ubiquitin ligase activity, to thoroughly explore whether the Trim31–Rhbdf2 inter-action significantly and positively contributes to the protective function of Trim31 on hepatic steatosis, a lentivirus loaded Trim31 with a RING-finger domain ablation (LV-Trim31 RING$^\Delta$) was generated and used to further investigate the influence of Trim31 RING$^\Delta$ on HFD-induced liver steatosis, insulin resistance, and inflammation (Supplementary Fig. S11a and Supplementary Fig. S12a), as described in Fig. 4. As expected, in the ex vivo experiment, in HFD-fed mice, Trim31 RING$^\Delta$ hepatocyte trans-plantation (THKO)(LV+) failed to alter liver weight, body weight, the LW/BW ratio, AST and ALT levels, and liver lipid accumula-tion, compared to controls (THKO)(LV−) (Supplementary Fig. S11b–d). Additionally, analysis of blood glucose levels and contents of hepatic TG, TC and NEFA indicated no significant differences (Supplementary Fig. S11e–g). Meanwhile, hepatocytes expressing Trim31 with RING-finger domain deletion were unable to regulate the expression of genes involved in fatty acid uptake and synthesis or the expression of genes involved in fatty acid β-oxidation in HFD-fed THKO (LV+) mice (Supplementary Fig. S11h). Furthermore, no significant changes in insulin signaling activity or in the Rhbdf2-MAP3K7 axis and downstream events were observed compared to controls, as

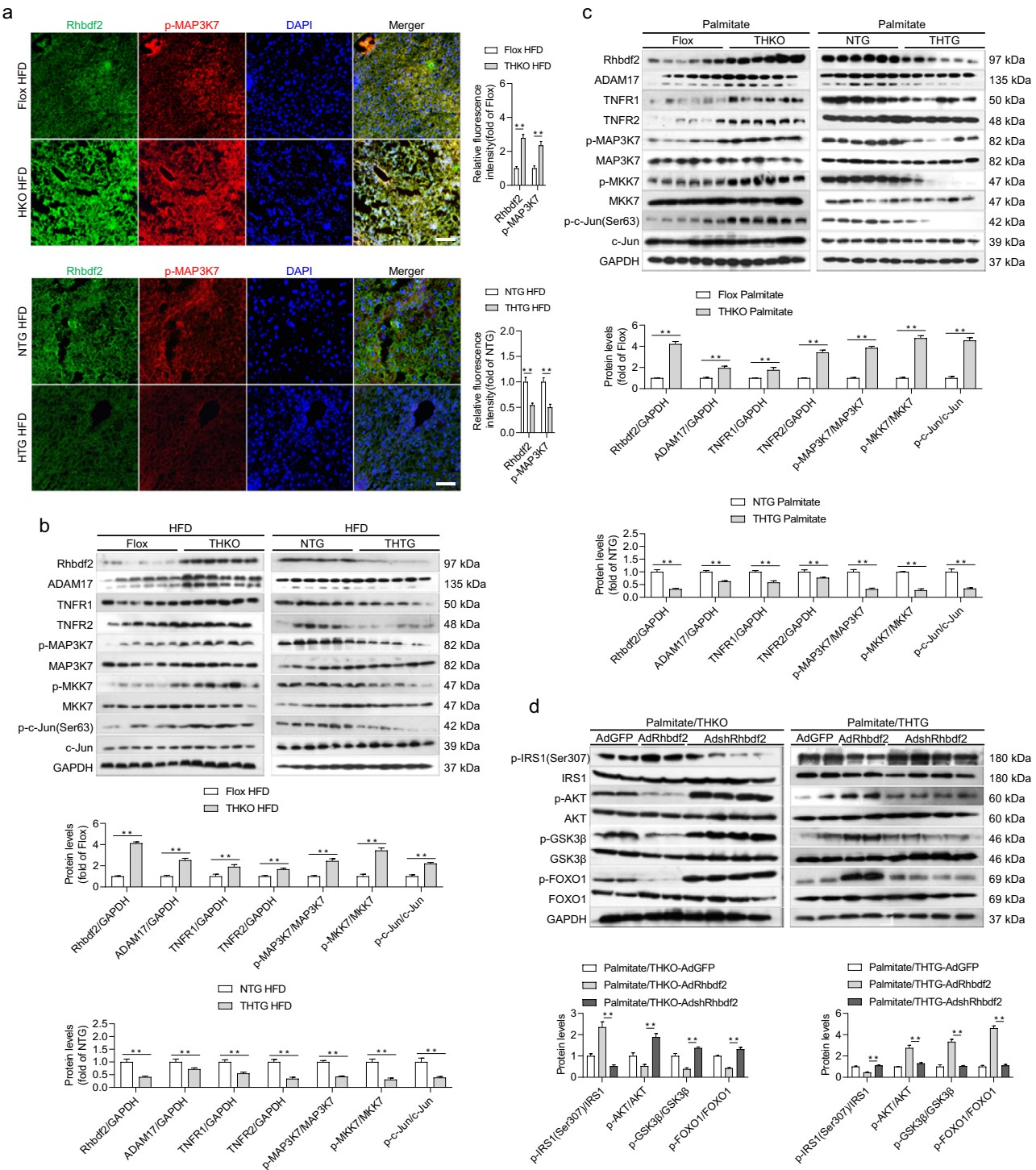

revealed by immunoblotting, mRNA quantification, and measurements of pro-inflammatory mediators (Supplementary Fig. S11i, j and Supplementary Fig. S12b, c). Consistent with these findings, the ex vivo gene therapy by LV-hTRIM31 further demonstrated that mice transplanted with human TRIM31 RING$^\Delta$-hepatocytes transplantation also did not markedly alleviate HFD-induced insulin resistance, liver steatosis, and inflammation (Supplementary Fig. S13a–g). These data further indicated that the RING domain of Trim31 in mice and TRIM31 in humans is essential for the protective function of E3 ubiquitin-protein ligase. In addition, an adeno-associated virus serotype 8 (AAV8)-thyroxine-binding

globulin (TBG) harboring Trim31 with a RING-finger domain mutant (AAV-Trim31 RING$^\Delta$) was accordingly generated and injected into the 6-week HFD-fed preconditioned WT mice in parallel with AAV-Trim31. The mice injected with AAV-GFP were treated as controls (Supplementary Fig. S14a). Undoubtedly, AAV-Trim31 RING$^\Delta$ lost the ability to affect body weight, blood glucose levels, blood insulin levels, liver weight, LW/BW ratio, AST, ALT, and AKP levels, pathological phenotype, and hepatic TG, TC, and NEFA levels, while AAV-Trim31 did not (Supplementary Fig. S14b–f). Meanwhile, consistent with the ex vivo data as indicated in Supplementary Fig. S11, mice with AAV-Trim31 RING$^\Delta$

**Fig. 5 Inactivation of Rhbdf2–MAP3K7 axis is essential for Trim31 function. a** Representative immunofluorescence images of Rhbdf2 and p-MAP3K7 co-expression in mice liver sections isolated from Flox, THKO- NTG, and THTG mice that were treated with a HFD for 16 weeks (magnification, ×40; $n = 10$ images per group for each staining) (**$P < 0.01$ vs. Flox HFD groups or NTG groups). **b** Representative immunoblotting bands for expression alterations of total amounts or phosphorylated forms of critical indicators associating with the Rhbdf2-MAP3K7 axis and its downstream events cascades including ADAM17, TNFR1/2, MKK7, p-MKK7, c-Jun, and p-c-Jun (Ser63) in the liver of 16-week HFD-fed Flox, THKO- NTG and THTG mice ($n = 6$ per experiment) (**$P < 0.01$ vs. Flox HFD groups or NTG groups). The GAPDH was used as a loading control. **c** Representative immunoblotting bands for expression changes of total amounts or phosphorylated forms of critical indicators associating with the Rhbdf2–MAP3K7 axis and its downstream events cascades including ADAM17, TNFR1/2, MKK7, p-MKK7, c-Jun, and p-c-Jun (Ser63) in primary hepatocytes isolated from the Flox, THKO- NTG, and THTG mice that were incubated with 400 μM PA for 10 h ($n = 6$ per experiment) (**$P < 0.01$ vs. Flox palmitate groups or NTG palmitate groups). The GAPDH was used as a loading control. **d** Representative immunoblotting bands for expression changes of total amounts or phosphorylated forms of critical indicators associating with the insulin signaling, including IRS1, p-IRS1(Ser307), AKT, p-AKT, GSK3β, p-GSK3β, FOXO1, and p-FOXO1, in the adenovirus-packed full-length Rhbdf2 sequences (AdRhbdf2) or shRNA targeting Rhbdf2 (AdshRhbdf2)-transfected THKO or THTG primary hepatocytes that were treated with 400 μM PA for 10 h ($n = 4$ per experiment) (**$P < 0.01$ vs. Palmitate/THKO-AdRhbdf2 or Palmitate/THTG-AdRhbdf2). The corresponding AdGFP was used as controls. The GAPDH was used as a loading control. Data are expressed as mean ± SEM. The relevant experiments presented in this part were performed independently at least three times. Significance determined by one-way analysis of variance (ANOVA) followed by Dunnett's multiple comparisons test (**a–c**) and Student's two-tailed $t$ test analysis (**d**).

administration did not show any significant improvement in insulin signaling, expression of genes involved in fatty acid uptake/synthesis and fatty acid β-oxidation, and the Rhbdf2-associated downstream inflammatory signaling cascade (Supplementary Fig. S14g–i). Collectively, the protective effects of Trim31 on the regulation of hepatic steatosis and inflammation can be primarily attributed to the Trim31-Rhbdf2 interaction and the RING-finger domain of Trim31.

**Targeting Rhbdf2 is required for the protective effects of Trim31 against hepatic steatosis.** To further confirm that the inhibition of Rhbdf2 signal mediates the protective function of Trim31 against fatty liver, the mice with hepatocyte-specific Trim31 deficiency (THKO) were mated with Rhbdf2$^{flox/flox}$ mice (Alb-Cre; Rhbdf2$^{flox/flox}$, hereafter referred to as RHKO), as indicated in the Methods section, to generate hepatocyte-specific Trim31 and Rhbdf2 double deletion mice (Alb-Cre; Rhbdf2$^{flox/flox}$, Trim31$^{flox/flox}$, hereafter referred to as DHKO). Both of Trim31 and Rhbdf2 deficiency in liver samples was determined by western blotting analysis (Fig. 7a). The deficiency of Rhbdf2-triggered mitigation of prolonged HFD-induced steatohepatitis has been intrinsically confirmed by our previous study. As expected, indeed,Rhbdf2 deletion blocked the effects of Trim31 ablation on the HFD-stimulated upregulation of the Rhbdf2-MAP3K7 axis and downstream signaling cascades, body weight, liver weight, LW/BW ratio, insulin signaling, blood glucose levels, liver lipid deposition, and elevated expression of pro-inflammatory genes (Fig. 7a–f and Supplementary Fig. S15a–e). In addition, PA-induced changes in intracellular TG levels in vitro also showed the same trend (Fig. 7g and Supplementary Fig. S16). These results indicated that the Rhbdf2 pathway is required for the protective effects of Trim31 against NAFLD.

**Evaluation of therapeutic feasibility and effect of targeting Trim31-Rhbdf2 administered signaling in nonalcoholic steatohepatitis (NASH).** Based on the protective effect of Trim31 in NAFLD, we accordingly investigated the therapeutic feasibility and effect of targeting the Trim31-Rhbdf2-MAP3K7 pathway in the development and progression of NASH. The THTG mice were then fed with an HFHF diet containing 14% protein, 42% fat, 44% carbohydrates, and 0.2% cholesterol and with a total of 42 g/L of carbohydrate mixed in drinking water at a ratio of 55% fructose and 45% sucrose by weight, for 16 weeks to generate mice with a NASH phenotype (THTG-HFHF)[25]. NTG mice with ad libitum access to this diet for 16 weeks were used as controls (NTG-HFHF). Unsurprisingly, in the HFHF-stimulated NASH

model, the liver weight, LW/BW ratio, and concentrations of hepatic TG, TC, and NEFA levels were markedly lower in the THTG group than in the NTG controls after 16 weeks on an HFHF diet, accompanied by no significant difference in body weight between the THTG-HFHF and NTG-HFHF mice (Fig. 8a–c). Also, compared to NTG mice, remarkable decreases in liver lipid deposition, expression of fatty acid synthesis genes (e.g., CD36, FASN, and ACACa), expression of inflammation-related genes (e.g., TNF-α, IL-1β, IL-6, and CCL-2), liver fibrosis (i.e., collagen deposition), and expression of collagen synthesis-related genes (e.g., TIMP1, CTGF, COL1A1, and COL3A1) were observed in THTG mice after HFHF diet treatment (Fig. 8d–i), as determined by H&E staining, Oil red O staining, F4/80 immunohistochemical assay, Sirius red staining, Masson staining, and measurements of the expression levels of inflammation- and collagen-associated genes. Moreover, Trim31 overexpression also decidedly downregulated mice serum AST, ALT, and AKP levels, pro-inflammatory cytokines TNF-α, IL-6, and IL-1β concentrations, and the Trim31-regulated Rhbdf2-MAP3K7 axis and downstream signaling cascades after 16 weeks of HFHF challenge (Fig. 8j and Supplementary Fig. S17a, b). Besides, a gene therapy approach using AAV vectors was also used to further evaluate the therapeutic potential of Trim31 targeting. Consistent with the data in Fig. 8, AAV8-mediated Trim31 gene therapy significantly protects against HFHF-induced nonalcoholic steatohepatitis (NASH) in vivo (Supplementary Fig. 18a–j). Collectively, our data revealed the positive effects of Trim31 on mitigation of steatohepatitis and associated metabolic syndrome in mice.

## Discussion

Nowadays, changing lifestyle, healthy diets, moderate exercise, and maintaining a good mental state are still the only option for clinical management of NAFLD[26–28]. Considering the complex pathogenesis of NAFLD, currently, there are no clinically effective therapeutic drugs for NAFLD. Therefore, a comprehensive and detailed understanding of the pathogenic mechanism underlying the NAFLD process and the search for pivotal signaling regulators have become urgent issues for the development of NASH drugs. In the present report, using hepatocyte-specific Trim31 deficiency (THKO)/transgenic mice (THTG) and lentivirus-mediated ex vivo gene therapy (LV-Trim31) mice, we identified liver Trim31 as a significant protective regulator against HFD/HFHF-stimulated or genetically induced inflammation, insulin resistance, liver steatosis, and NASH. A study on the underlying molecular mechanism uncovered that after prolonged exposure to a high-calorie diet, reduced Trim31 expression, which results in activation of the Rhbdf2-MAP3K7

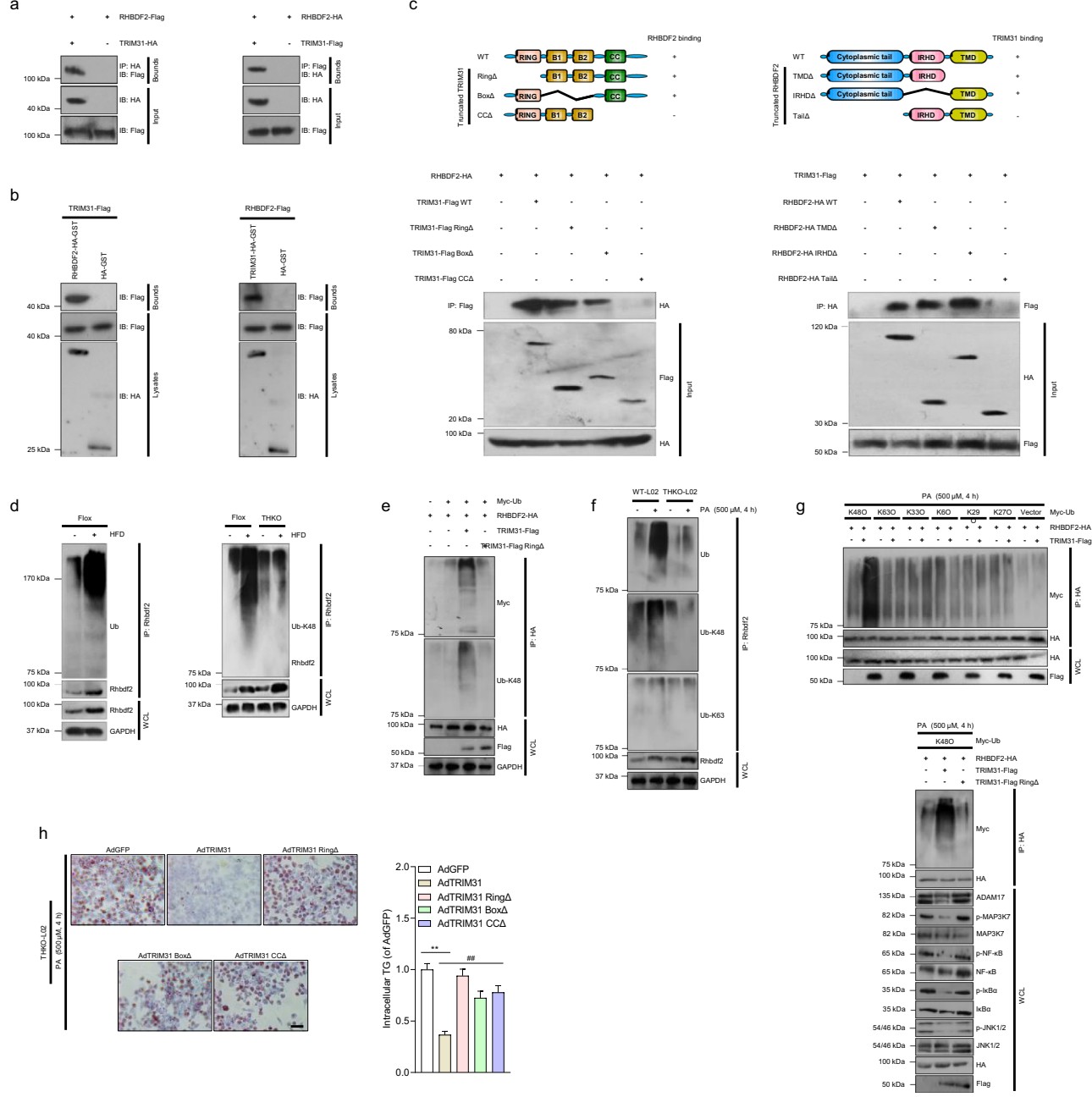

axis, was observed in in vivo and in vitro experiments. Acutely activated Rhbdf2–MAP3K7 signaling with dysfunctional Trim31 further increases downstream events, including JNK-IRS1-related insulin signaling, the IκBα-NF-κB-c-Jun pathway, suppression of AKT-GSK3β-FOXO1 phosphorylation levels, and CTGF-TIMP1 cascades, promoting the occurrence of impaired insulin signaling (insulin resistance), chronic inflammation, abnormal glucose metabolic disorder, and collagen deposition in the liver, accordingly aggravating abnormal lipid metabolism and the consequential liver steatosis and NASH phenotype (Fig. 8k).

Insulin resistance, liver steatosis, prolonged inflammation, and pro-collagenic progress corporately facilitate the pathological formation of metabolic syndrome and NASH[1–4,29]. Of note, long-term high-calorie diet treatment is capable of stimulating severe insulin resistance in the liver, where the normal function of insulin to regulate glucose metabolism is inhibited, however,

the effect of insulin to promote adipogenesis is fully preserved. Thus, increasing insulin resistance leads to abnormal accumulation of excessive lipids in the liver[29,30]. In addition, elevated inflammatory action also can be triggered by the too high energy intake, such as HFD/HFHF, and synergistically promotes the development of insulin resistance[31]. Our previous studies have also indicated that upregulated Rhbdf2 triggered by HFD significantly drives progression of NAFLD/NASH pathological phenotype in vivo, revealing a potential treatment prospect of Rhbdf2 in liver steatosis-associated metabolic disorders[15,32]. Fortunately, studying Rhbdf2, we found that Trim31, an important member of the E3 ubiquitin ligase, is a potential and effective inhibitor of Rhbdf2 expression. Notably, in-depth investigations have further found that the protective function of hepatocyte Trim31 is not limited to the liver, but affects the entire metabolic system, including reducing body weight gain and visceral fat, improving dyslipidemia, and downregulating serum inflammatory cytokines and chemokines.

**Fig. 6 TRIM31 interacts with RHBDF2 and mediates degradation of Rhbdf2 by K48-linked polyubiquitination. a** Representative western blotting for co-immunoprecipitation (CO-IP) detection in L02 cells transfected with HA- or Flag-tagged RHBDF2 or TRIM31 vectors. The anti-Flag or anti-HA antibodies were used as probes ($n = 4$ per experiment). **b** The glutathione S-transferase (GST) pull-down assays in which either GST-tagged RHBDF2 (RHBDF2-HA-GST) or blank HA-GST was used to pull down TRIM31-Flag or GST-tagged TRIM31 (TRIM31-HA-GST) or blank HA-GST was used to pull down RHBDF2-Flag. Purified GST was used as the control ($n = 4$ per experiment). **c** The binding domains of RHBDF2 and TRIM31 were detected using full-length and truncated Rhbdf2 or Trim31 expression vectors based on IP assays. Anti-Flag or anti-HA antibodies were used to confirm the binding sites of RHBDF2 and TRIM31, respectively ($n = 4$ per experiment). **d** The ubiquitination levels of Rhbdf2 in the liver samples of Trim31-Flox or THKO mice in the presence of HFD feeding for 16 weeks ($n = 3$ mice per group). **e** Representative western blotting assays of lysates from L02 cells transfected with Myc-tagged ubiquitin (Myc-Ub), RHBDF2-HA, full-length TRIM31-Flag, and TRIM31 with RING domain mutant (TRIM31-Flag RING$^\Delta$), followed by IP with anti-HA, probed with K48-Ub or anti-Myc. **f** Representative western blotting assays of lysates from 500 μM PA-treated WT L02 (WT-L02) cells or L02 with TRIM31 knockout (THKO-L02) cells for 4 h, followed by IP with anti-RHBDF2, probed with anti-Ub, K48-Ub, K63-Ub. **g** Ubiquitination levels of RHBDF2 after TRIM31-Flag overexpression and in response to PA administration in L02 cells co-transfected with RHBDF2-HA and the indicated Myc-tagged ubiquitin constructs (K48O, K63O, K33O, K6O, K29O, K27O) (upper). K48O means ubiquitin in which all lysines except K48 were mutated. The empty vector was used as a control. Representative western blotting indicating the ubiquitination levels of RHBDF2 in L02 cells transfected with the K48O vector in different combinations and the indicated downstream events cascades protein expression levels in WCL (lower). **h** Representative images of the intracellular TG levels in THKO-L02 cells transfected with adenovirus-packed full-length TRIM31 sequences (AdTRIM31) or different truncated TRIM31 sequences, which were then incubated with 500 μM PA for 4 h. The adenovirus-containing GFP vector (AdGFP) was used as controls (magnification, ×100; $n = 10$ images per group for each staining) (##$P < 0.01$ vs. AdTRIM31 groups; **$P < 0.01$ vs. AdGFP groups). The relevant experiments presented in this part were performed independently at least three times. Data are expressed as mean ± SEM. The relevant experiments presented in this part were performed independently at least three times. Significance determined by one-way analysis of variance (ANOVA) followed by Dunnett's multiple comparisons test.

Because the liver is the most important metabolism/immune organ, this may be a supporting role of Trim31 regulating the liver. Importantly, previous studies have also confirmed that dysfunctional Trim31 is tightly involved in inflammation-related diseases, followed by significant upregulation of Rhbdf2 levels[15,21]. Also, Rhbdf2 has been shown to be regulated by ubiquitination. Consistent with these studies, we determined that Trim31 levels were significantly decreased in the livers of HFD-treated mice and patients with hepatic steatosis.

Additionally, in the complicated regulatory network of liver metabolism, in the present study, the increased Rhbdf2–MAP3K7 axis activity and downstream pro-inflammatory events during HFD-triggered liver steatosis were markedly prevented by liver-specific Trim31 overexpression, indicating the inhibitory function of Trim31 on the Rhbdf2-MAP3K7-mediated inflammatory response, insulin resistance, and steatosis. Positive ex vivo gene therapy to restore the expression of Trim31 in mice also showed that it can effectively inhibit the activity of Rhbdf2 and alleviate the above three pathological phenotypes. Surprisingly, previous studies have found that the function of Trim31 may have a "Janus-faced" role in the regulation of immune-related diseases, catalyzing the targeted substrate degradation or signal transduction by ubiquitination[18–20]. Reports have shown that Trim31-deficient mice lost the ability to inhibit the inflammatory response in colonitis[21,22]; however, Trim31 was determined to be a positive promoter that could exacerbate cardiomyopathy triggered by inflammation-associated signaling[18]. These seemingly opposite conclusions may indicate that the function of Trim31 is closely related to organ types and cell types. In our present work, we found that Trim31 has the ability to suppress liver steatosis and improve insulin signaling and hepatic inflammation by promoting Rhbdf2 degradation via K48-linked ubiquitination, further restraining Rhbdf2-MAP3K7 activity and downstream events. Trim31 performs its biological functions at the molecular level based on the different motifs. It has been reported that the C-terminal CC domain of Trim31 is basically responsible for the binding and interaction with the target, while the N-terminal RING domain is responsible for ubiquitin ligase activity and catalytic function[22,23,33]. Indeed, here we found that K48-linked Rhbdf2 ubiquitination could be catalyzed in vitro, and Trim31 deletion greatly blocked this endogenous ubiquitination of Rhbdf2. More detailed in vitro studies of the molecular biological functions of Trim31 have shown that it directly binds to Rhbdf2

via the CC domain to target Rhbdf2 activity; the RING-finger of Trim31 ubiquitinates Rhbdf2, mediating proteasomal Rhbdf2 degradation, via the K48 linkage. However, the interaction between endogenous Rhbdf2 and Trim31 in liver cells still needs further investigation. Once the interaction between Rhbdf2 and Trim31 was blocked, the protective function of Trim31 on HFD-induced liver steatosis was strongly inhibited. In addition, the non-functional Trim31 RING domain mutant lost its E3 ligase activity to reduce the activation of the Rhbdf2-MAP3K7 axis and the downstream JNK1/2-NF-κB pathway and improve insulin signaling. Of note, the Trim31 RING domain mutant also has an inappreciable effect on lipid metabolism in PA-challenged hepatocytes. Therefore, Trim31-Rhbdf2 binding and the subsequent Rhbdf2 ubiquitination are required for and conduce to the mitigation of liver steatosis, insulin resistance, and liver inflammation triggered by hepatocyte Rhbdf2-MAP3K7 signaling. It should be noted that Rhbdf2, as an important inactive member of the rhomboid intramembrane proteinase family, has been confirmed to be associated with the regulation and trafficking of inflammatory signals[15–17]. However, previous studies have shown that the production of the pro-inflammatory cytokine TNF-α is independent of Rhbdf2 in cells other than immune cells[34]. Our previous finding also indicated that the absence of Rhbdf2 in non-immune cells did not affect the expression and secretion of inflammatory factors such as TNF-α, IL-6 and IL-1β[15]. Of note, although the release of TNF-α in non-immune cells, e.g., hepatocytes, does not completely depend on the trafficking of Rhbdf2, increased ubiquitination of Rhbdf2 is able to recruit MAP3K7 and elevate its phosphorylation levels, and then promote downstream cascades activation under stimulation conditions, significantly activate NF-κB inflammatory signals and accelerate the secretion of inflammatory factors[15]. In the present study, we have found that in PA-induced hepatocytes, Trim31 significantly promotes the degradation of Rhbdf2 by K48 ubiquitin linkage, followed by a strong decrease in the expression of TNF-α and other inflammatory factors. Previous studies have also found that Trim31 could promote the inactivation of the NLRP3 inflammasome, which also implied that, in addition to Rhbdf2, Trim31 has other target substrates and regulatory mechanisms.

In conclusion, and of functional relevance, we determined that liver Trim31 expression correlated with NAFLD/NASH and metabolic disorder e.g., insulin resistance and glycometabolic

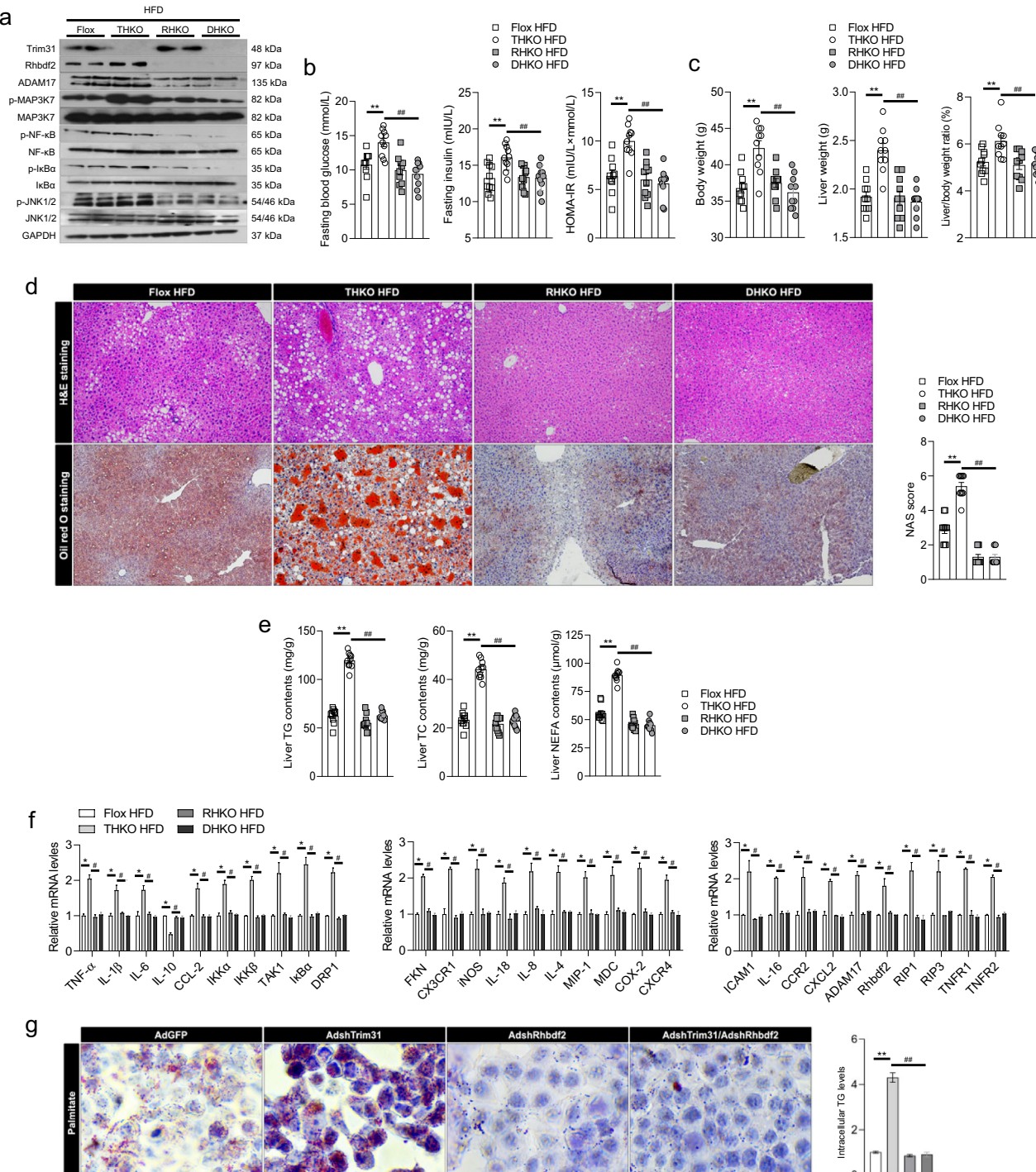

**Fig. 7 Rhbdf2 signaling is obbligato for the protective effect of Trim31 on suppression of liver steatosis and inflammation. a** Representative immunoblotting bands of Trim31, Rhbdf2, and corresponding downstream event indicator levels of total and phosphorylated ADAM17, MAP3K7, p-MAP3K7, NF-κB, p-NF-κB, IκBα, p-IκBα, JNK1/2, and p-JNK1/2 in liver samples isolated from the indicated mice fed a HFD for 16 weeks. **b**, **c** Records for body weight, liver weight, the ratio of liver weight/body weight (%) (**b**), and fasting blood glucose levels, fasting insulin levels, and HOMA-IR index (**c**) in a 16-week HFD-fed indicated mice ($n = 10$ mice per group) (**$P < 0.01$ vs. Flox HFD groups, ##$P < 0.01$ vs. THKO HFD groups). **d** Representative pictures for Oil red O staining and H&E staining-indicated liver histopathologic changes of a 16-week HFD-fed indicated mice (magnification, ×100; $n = 10$ images per group for each staining) (**$P < 0.01$ vs. Flox HFD groups, ##$P < 0.01$ vs. THKO HFD groups). **e** Liver lipid contents including TG, TC, and NEFA of the indicated mice after a 16-week HFD ingestion ($n = 10$ mice per group) (**$P < 0.01$ vs. Flox HFD groups, ##$P < 0.01$ vs. THKO HFD groups). **f** qPCR analysis of the relative mRNA expression of genes in the livers associated with inflammation signaling in 16-week HFD-fed mice ($n = 10$ liver samples per group) (*$P < 0.05$ vs. Flox HFD groups, #$P < 0.05$ vs. THKO HFD groups). **g** Representative images of Oil red O staining of primary hepatocytes that were transfected with AdshRhbdf2 or AdshTrim31 or co-treated with AdshRhbdf2/AdshTrim31 or PA for 10 h (magnification, ×200; $n = 10$ images per group for each staining) (**$P < 0.01$ vs. Flox HFD groups, ##$P < 0.01$ vs. THKO HFD groups). The relevant experiments presented in this part were performed independently at least three times. Data are expressed as mean ± SEM. The relevant experiments presented in this part were performed independently at least three times. Significance determined by one-way analysis of variance (ANOVA) followed by Dunnett's multiple comparisons test.

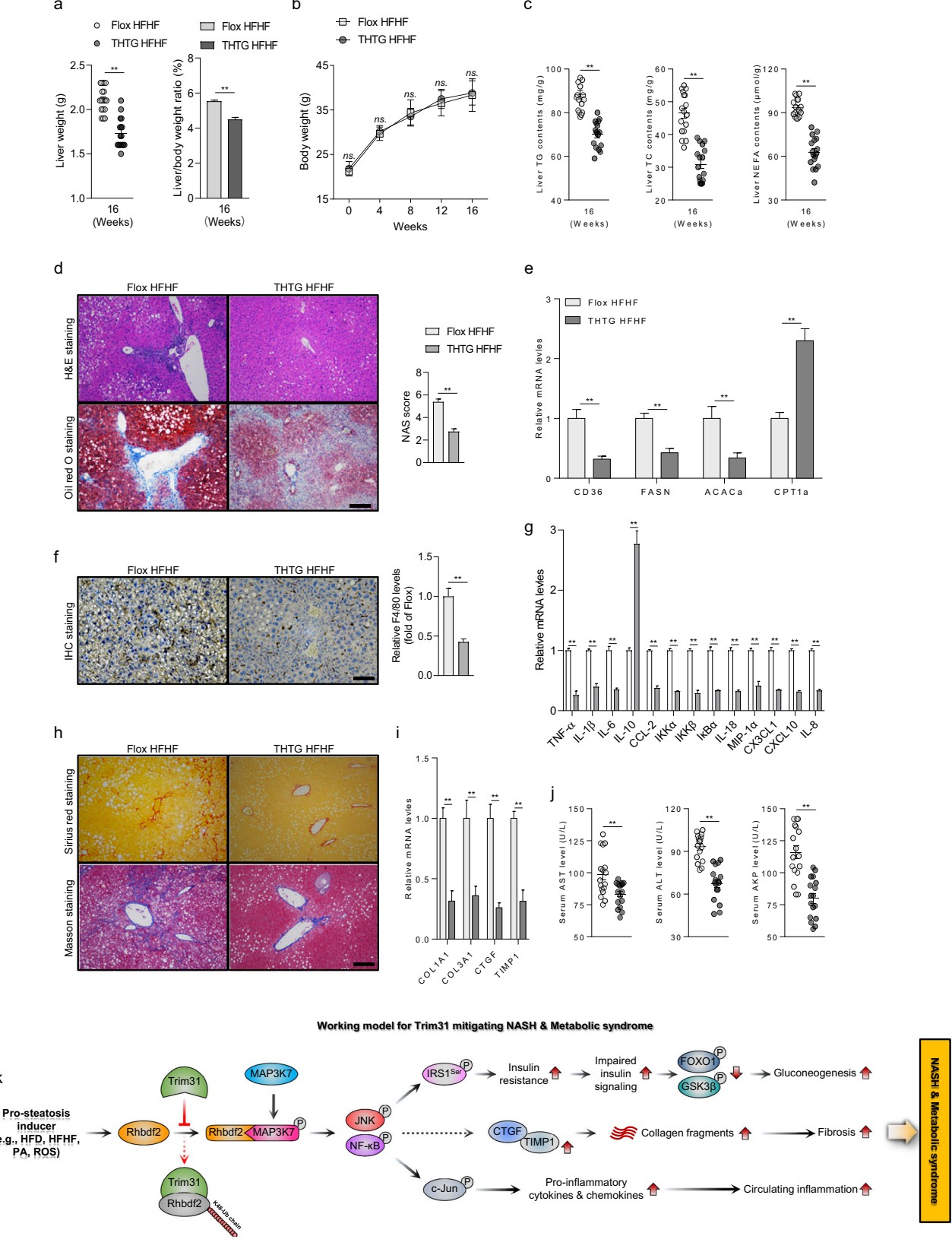

disorder both in humans and in murine. Furthermore, the hepatic Trim31 expression in mice is responsive to dietary interventions that significantly moderated insulin resistance, hepatic steatosis, and inflammation phenotype via mechanically, regulating Rhbdf2 proteasome degradation. Nevertheless, as an important member

of E3 ubiquitin ligases, the activity of Trim31 in the liver from mice or human patients also needs to be strictly and carefully monitored, evaluated, and controlled for its function. Finally, these obtained findings encourage Trim31 as a feasible therapeutical target for NAFLD/NASH and associated-metabolic

**Fig. 8 Therapeutic effects of Trim31 on HFHF-induced nonalcoholic steatohepatitis (NASH) in vivo. a, b** Records for liver weight, the ratio of liver weight/body weight (**a**) and time-course changes of body weight in THTG and NTG mice (**b**) after a 16-week HFHF administration ($n = 17$ mice per group) (**P < 0.01 vs. NTG HFHF groups, ns., no significant difference). **c** Liver lipid contents including TG, TC, and NEFA of the indicated mice after 16-week HFHF ingestion ($n = 17$ mice per group) (**P < 0.01 vs. NTG HFHF groups). **d** Representative pictures for Oil red O staining and H&E staining-indicated liver histopathologic changes of a 16-week HFHF-fed THTG and NTG mice (magnification, ×100; $n = 10$ images per group for each staining) (**P < 0.01 vs. NTG HFHF groups). **e** qPCR analysis of the relative mRNA expression of genes in the livers associated with fatty acid metabolism after a 16-week HFHF administration ($n = 10$ liver samples per group) (**P < 0.01 vs. NTG HFHF groups). **f** Immunohistochemistry staining of F4/80 analysis showing the histopathologic changes of the liver samples in indicated mice (magnification, ×200; $n = 10$ images per group for each staining) (**P < 0.01 vs. NTG HFHF groups). **g** qPCR analysis of the relative mRNA expression of genes in the livers associated with inflammation after a 16-weeks HFHF administration ($n = 10$ liver samples per group) (**P < 0.01 vs. NTG HFHF groups). **h** Representative pictures for Sirius red staining and Masson staining showing the collagen deposition levels of the liver samples in indicated mice (magnification, ×100; $n = 10$ images per group for each staining). **i, j** qPCR analysis of the relative mRNA expression of genes in the livers associated with collagen synthesis after a 16-weeks HFHF administration (**i**) ($n = 10$ liver samples per group); and hepatic function indicator AST, ALT, and AKP levels in the 16-week HFHF-fed THTG and NTG mice (**j**) ($n = 17$ mice per group) (**P < 0.01 vs. NTG HFHF groups). **k** Schematic diagram showing the molecular mechanism underlying Trim31-mediated Rhbdf2 degradation and protective effects in the development and progression of NASH. Data are expressed as mean ± SEM. The relevant experiments presented in this part were performed independently at least three times. Significance determined by Student's two-tailed $t$ test analysis.

disorders which might be accountable to diagnosis and therapeutic schemes.

## Methods

**Study approval and ethics statements.** The whole experimental procedures and protocols involving laboratory animals' operation were performed in accordance with *the Guide for the Care and Use of Laboratory Animals*, issued by the National Institutes of Health in 1996, and approved by *the Institutional Animal Care and Use Committee* in Chongqing Key Laboratory of Medicinal Resources in the Three Gorges Reservoir Region, School of Biological and Chemical Engineering, Chongqing University of Education. The protocols used in this study were also in accordance with the *Regulations of Experimental Animal Administration* issued by the Ministry of Science and Technology of the People's Republic of China (http://www.most.gov.cn). All studies associating with human subjects (commercial human cell lines and liver samples) were based on Declaration of Helsinki, and completely permitted by the *Academic Research Ethics Committee* in Chongqing Key Laboratory of Medicinal Resources in the Three Gorges Reservoir Region, the Key Laboratory of Biorheological Science and Technology (Chongqing University) and Shandong Cancer Hospital and Institute of Shandong First Medical University & Shandong Academy of Medical Sciences. According to the research approved by IRB/IEC, written informed consents were required and samples were then collected from the donors. Additionally, the corresponding written informed consents were obtained from the study participants themselves. Appropriate security measures were taken to protect the privacy and confidentiality of the subjects.

**Reagents.** The primary antibodies used in the work and against the following proteins were obtained from Cell Signaling Technology, Inc (CST): anti-GAPDH (#2118, dilution 1:10000), p-AKT (#4060, dilution 1:1000), AKT (#4691, dilution 1:1000), p-GSK3β (#9322, dilution 1:1000), GSK3β (#12456, dilution 1:1000), FOXO1 (#2880, dilution 1:1000), anti-p-JNK (#4668, dilution 1:1000), anti-JNK (#9258, dilution 1:1000), anti-MAP3K7 (#4505, dilution 1:1000), anti-p-MAP3K7 (#9339, dilution 1:1000) and p-IκBα (#2859, dilution 1:1000). Antibodies against anti-TNFR1 (#ab223352, dilution 1:1000), anti-TNFR2 (#ab109322, dilution 1:1000), anti-MKK7 (#ab52618, dilution 1:1000), anti-p-MKK7 (#ab192592, dilution 1:1000), anti-c-Jun (#ab40766, dilution 1:1000), anti-p-c-Jun (#ab32385, dilution 1:1000), anti-IRS-1 (#ab52167, dilution 1:1000), anti-NF-κB (#ab16502, dilution 1:1000), anti-p-NF-κB (#ab86299, dilution 1:1000), p-FOXO1 (#ab131339, dilution 1:1000), anti-IκBα (#ab32518, dilution 1:1000), anti-IKKβ (#ab124957, dilution 1:1000) and anti-Rhbdf2 (#ab116139, dilution 1:1000) were purchased from Abcam (Cambridge, MA, USA). In addition, the antibodies against anti-p-MAP3K7 (#PA5-99340, dilution 1:1000), anti-p-IKKβ (#PA5-36653, dilution 1:1000), anti-Trim31 (#PA5-40961, dilution 1:1000), anti-ADAM17 (#PA5-27395, dilution 1:1000) and anti-p-IRS-1(Ser[307]) (#PA1-1054, dilution 1:1000) were obtained from Thermo Fisher Scientific (Waltham, MA, USA). The antibody against anti-p-IRS-1(Tyr[608]) (#09-432, dilution 1:1000) was purchased from Millipore. The antibody against anti-PEPCK (#sc-271029, dilution 1:1000) was purchased from Santa Cruz Biotechnology. The antibodies including anti-Rhbdf2 (#orb386934, dilution 1:1000) and anti-G6Pase (#ARP44223-P050, dilution 1:1000) were obtained from Biobyt (St Louis, MO, USA) and Aviva Systems Biology Corporation (San Diego, CA, USA), respectively. Moreover, antibodies against anti-HA (Abcam, #ab9110, dilution 1:1000), anti-Flag (Thermo Fisher Scientific, #MA1-91878, dilution 1:1000), anti-Myc (Abcam, #ab9106, dilution 1:1000), Anti-Ub (Abcam, #ab134953, dilution 1:1000), anti-Ub (linkage-specific K48) (Abcam, #ab140601, dilution 1:1000) and anti-Ub (linkage-specific K63) (Abcam, #ab179434, dilution 1:1000) were also used in the current study. The anti-Trim31 antibody (C-term) (#AP13642B, dilution 1:1000) used in this

experiment was obtained from ABGENT, Inc., (San Diego, CA, USA). The Pierce™ Rapid Gold BCA Protein Assay Kit (#A53225, Thermo Fisher Scientific) was used to detect protein concentration of samples. The HRP-conjugated secondary antibodies (Abcam) with 1:10000 dilution was used in immunoblotting assay for visualization. The MG132, palmitate (PA), cycloheximide, chloroquine, N-Acetyl-L-cysteine (NAC), catalase (CAT), TNF-α were purchased from Sigma-Aldrich (St Louis, MO).

**Human liver tissue samples.** Human liver tissue samples were collected painlessly from adult patients with nonalcoholic fatty liver disease who underwent liver transplantation or liver biopsy. The corresponding control liver tissue was harvested from the donor who could not be used for liver transplantation due to non-hepatic reasons. Written informed consent was integrally obtained from the liver donors and their families. The patient characteristics and liver injury-associated serology are shown in Supplementary Table 1. All procedures associated with human subjects used in this study were based on the Declaration of Helsinki, and completely permitted by the Academic Research Ethics Committee in Chongqing Key Laboratory of Medicinal Resources in the Three Gorges Reservoir Region and other participating units.

**Animals.** Trim31[flox/flox] mice based on C57BL/6N background were generated using CRISPR/Cas9-mediated genome engineering system. Exons 4 and 5 of Trim31 were then selected as conditional knockout region (CKO). In brief, the chosen exons of Trim31 were flanked by loxP sites, and therefore two single guide RNAs (gRNA1 and gRNA2) targeting Trim31 introns were designed. The targeting vector containing Trim31 exon 4 and 5 flanked by two loxP sites and the two homology arms were used as the template. The targeting vector, guide RNA1, and guide RNA2 and Cas9 mRNAs were co-injected into fertilized eggs for CKO mouse production. The obtained mice, which had exon 4 and 5 flanked by two loxP sites on one allele, were used to construct Trim31[flox/flox] mice. Hepatocyte-specific Trim31 deletion (THKO) mice were created by mating Trim31[flox/flox] mice with albumin-Cre (Alb-Cre) mice (Jackson Laboratory, Bar Harbor, Maine, USA). A simple schematic diagram has been indicated in Supplementary Fig. S3a. Trim31[flox/flox] mice littermates were used in the study as controls for the obtained THKO mice.

The hepatocyte-specific Rhbdf2-knockout (RHKO) mice were also created using CRISPR/Cas9 system by specifically ablating the 4[th] exon of Rhbdf2 in hepatocytes. Detailed protocols and information regarding the establishment and genotype determination of these mice have been described previously[35]. In addition, the hepatocyte-specific Trim31 and Rhbdf2 double deletion (DHKO) mice were generated by crossing Rhbdf2[flox/flox] mice with THKO mice.

Conditional Trim31 transgenic (TG) mice were established by micro-injecting CAG-loxP-CAT-loxP-Trim31 into fertilized eggs isolated from C57BL/6 mice. The obtained pups were then genotyped by PCR followed by sequencing analysis. The obtained mice were identified by PCR analysis of tail genomic DNA. The offspring of these TG mice were mated with the Alb-Cre mice to establish hepatocyte-specific Trim31 transgenic (THTG) mice. The corresponding littermates without Trim31 overexpression in hepatocytes were used as controls (NTG). Additionally, all the other normal wild-type (WT) C57BL/6 N mice used in the current study were purchased from Beijing Vital River Laboratory Animal Technology Co., Ltd. (Beijing, China).

**Animal experiment design.** All animal procedures and protocols were approved by the Animal Care and Use Committee of all participating Units. Prior to all experiment's proper starts, the mice were subjected to adapt to the living environment for 7 days. The mice were housed in a constant temperature, humidity

(controlled by GREE central air-conditioner, #GMV-Pd250W/NaB-N1, China) and pathogen-free controlled environment (25 °C ± 2 °C, 50–60%) cage with a standard 12 h light/12 h dark cycle, plenty of water and food (pathogen-free) in their cages.

*Animal experiment design 1#.* The 6–8-week-old WT male mice (total 15) were fed with HFD fodder (20% kcal protein, 60 kcal% fat and 20% kcal carbohydrate, #D12492; Research Diets, New Brunswick, NJ, USA) for 16 weeks to induce fatty liver. The additional WT mice (total 15) were fed with a standard normal chow diet (20% kcal protein, 10 kcal% fat, and 70% kcal carbohydrate, #D12450H; Research Diets, New Brunswick, NJ, USA) for 16 weeks to be served as the control group (NCD). The age-matched (6–8-week-old) *ob/ob* mice (total 10) (#N000103, Nanjing Biomedical Research Institute of Nanjing University) were fed with NCD and then treated as another fatty liver model. In time-course experiments, a total of 15 WT mice for each time point were included. At the end of the experimental period, the liver tissue samples were collected from mice to detect corresponding signaling events.

*Animal experiment design 2#.* To investigate the protective function of Trim31 on HFD-induced insulin resistance, hepatic steatosis, and inflammation, the ex vivo gene therapy interventions by lentivirus-loading full-length Trim31 sequences (LV-Trim31) or mutant Trim31 with RING domain deletion (LV-Trim31 RING$^\Delta$) transduction and transplantation were performed in 8-weeks HFD-fed preconditioned THKO mice. The detailed protocols of the ex vivo therapy experiments were established in accordance with our previous reports[15]. A brief diagram of the experimental design has been indicated in corresponding figures. At the end of ex vivo therapy interventions, these mice were fasted for 8 h and then subjected to glucose and insulin tolerance tests. Accordingly, eye blood and liver samples were collected, weighed, and stored at −80 °C condition. The other part of liver tissue was subjected to histological analysis and biochemical analysis. A minimum of 3 independent experiments were conducted for all present data.

*Animal experiment design 3#.* To specifically over-expressed Trim31 in hepatocytes in vivo experiments, the adeno-associated virus serotype 8 (AAV8)-thyroxine-binding globulin (TBG) encoding full-length Trim31 sequences (AAV-Trim31) and AAV8 encoding mutant Trim31 with RING domain deletion (AAV-Trim31 RING$^\Delta$) delivery system were established according to a standard molecular procedure. Briefly, the whole opening reading frame (ORF) encoding Trim31 without intervening stop codon was cloned into AAV8 vector to generate AAV-Trim31. Accordingly, 6-weeks HFD-fed preconditioned WT mice were injected with AAV-Trim31 or AAV-Trim31 RING$^\Delta$ by tail vein with 100 µl of virus containing $2 \times 10^{11}$ vg of vectors and then fed with HFD for additional 10 weeks. The empty vector (AAV-GFP) was injected into mice as the corresponding control.

*Animal experiment design 4#.* To establish a fatty liver model, the male THKO, RHKO, DHKO, THTG mice and their corresponding littermates control mice at the age of 6–8 weeks were fed with HFD diet for 16 weeks to investigate the pathological changes. Also, the age-matched THKO, RHKO, DHKO, THTG mice and littermates were separately fed with NCD fodder for 16 weeks and treated as controls.

*Animal experiment design 5#.* To further investigate the protective effects of Trim31 on liver steatosis, a nonalcoholic steatohepatitis (NASH) mice model was established in accordance with previous reports[25]. The THTG mice were then fed with HFHF diet (14% protein, 42% fat, 44% carbohydrates, 0.2% cholesterol and with a total of 42 g/L of carbohydrate mixed in drinking water at a ratio of 55% fructose and 45% sucrose by weight) for 16 weeks to produce the phenotype of NASH. Also, the NTG mice used in this design as controls were synchronously allowed ad libitum access to this diet for 16 weeks.

**Cell culture and treatment.** The L02 cell line (Human normal hepatocyte cell line) was obtained from the Type Culture Collection of the Chinese Academy of Sciences, Shanghai, China. All resuscitated cell lines used in our laboratory were passaged no more than 30 times. Cell lines involved in experiments need to be tested for mycoplasma contamination by PCR analysis. The L02 cells were maintained in Dulbecco's Modified Eagle Medium (DMEM) (#A4192101, Thermo Fisher Scientific) containing 10% fetal bovine serum (#16140071, Gibco™) and 1% penicillin-streptomycin (#15140-122; Gibco™) and were incubated in a 5% CO$_2$, 37 °C water-jacket type cell incubator (Thermo Fisher Scientific). Primary hepatocytes used in the current experiments were isolated and collected from corresponding experimental mice by liver perfusion method as described previously[15,25]. Briefly, under painless anesthesia condition, mouse's abdominal cavity was opened. Therefore, the livers were carefully perfused with 1×liver perfusion medium (#17701-038, Gibco™) and 1× liver digest medium (#17703-034, Gibco™) via the portal vein. Then, 100 µm steel mesh was used to grind and filter the digested liver tissue. The primary hepatocytes were collected by centrifuging the filter liquor at 300 × *g*, 4 °C for 5 min, and further purified with 50% percoll solution (#17-0891-01, GE Healthcare Life Sciences). The obtained hepatocytes were cultured in DMEM medium containing 10% fetal bovine serum and 1% penicillin-

streptomycin and cultured in a 5% CO$_2$, 37 °C cell incubator. To construct a cell model of lipid deposition in vitro, the corresponding concentration of palmitic acid (PA) (dissolved in 0.5% fatty acid–free BSA) was prepared and obtained according to previous reports[15,25]. Then, the primary hepatocytes or L02 cells were treated with cell culture medium-containing PA for 10 or 4 h. Fatty acid–free BSA (0.5%) alone was used as vehicle control.

**Establishment of knockout cell lines.** The generation and protocol of Trim31-deficient cell lines used in this study were performed as described previously[25,36]. In brief, cell lines with Trim31 deletion were produced by CRISPR/Cas9 gene-editing system. The sgRNA targeting the human Trim31 genes were produced and packed into lentiCRISPR-V2 vectors to form the Cas9-sgRNA lentivirus. The oligo sequences used for the generation of sgRNA expression vector are as follows: sgRNA-F: CACCCAACTCGCTGTTGCGGAATC; and sgRNA-R: AAAC GATTCCGCAACAGCGAGTTG. The packaging vectors pSPAX2 and pMD2.G, together with sgRNA expression vector were then transfected separately into HEK293T cells using FuGENE® 6 Transfection Reagent for 42 h. Next, the L02 cells were transduced with the obtained supernatant containing lentivirus to construct the gene knockout cell lines. The cell clones with target gene deletion were selected by immunoblotting.

**Plasmids construction and transfection.** Human or mouse full-length Rhbdf2 and Trim31 expression plasmid were established by PCR-based amplification of cDNA, and then cloned into the 3×Flag-tagged pcDNA3.1 vector or 3×HA-tagged pcDNA3.1 vector (Invitrogen). Truncated Rhbdf2 and Trim31 fragments expression vector including TRIM31-Flag RING$^\Delta$, TRIM31-Flag Box$^\Delta$, TRIM31-Flag CC$^\Delta$, RHBDF2-HA TMD$^\Delta$, RHBDF2-HA IRHD$^\Delta$, and RHBDF2-HA Tail$^\Delta$ as indicated in the figure legends, were obtained using standard PCR methods and were then cloned into corresponding vectors. The Myc-ubiquitin WT expression vectors were constructed based on pcDNA3.1 vector. In addition, ubiquitin and corresponding derivatives, including ubiquitin in which the only complete amino acid residue was ubiquitin-K48O, ubiquitin-K63O, ubiquitin-K33O, ubiquitin-K6O, ubiquitin-K29O, ubiquitin-K27O, and control vector, were then packed into the Myc-tagged pcDNA3.1 plasmid (Thermo Fisher Scientific). Vectors were carefully transfected into L02 cells with Lipofectamine™ 3000 Transfection Reagent (Invitrogen™) according to the manufacturer's instructions. Moreover, to further investigate the effects of Trim31 on the lipid metabolism process in vitro experiments, here we have prepared an adenovirus-loaded Trim31 expression vector. Using a similar process to that of adeno-associated virus vector preparation, human full-length TRIM31 sequences and specific short hairpin RNA oligonucleotides sequences targeting human TRIM31 (shTRIM31) (shRNA sequences RNAi#1: TTCCCGTCAAAGGAAGTTTGG; RNAi#2: TATGATGGACT CATGCCTTGC) were respectively packed into adenovirus (AdTRIM31; AdshTRIM31) by Easy Adenoviral Vector System Kit (#240009, Agilent Technologies). The AdshRNA was used as a control for knockdown or overexpression, respectively. The recombinant adenovirus was purified and titrated to 5×10$^{10}$ plaque-forming units (PFU). The verification of the virus is based on DNA analysis of the virus, which is a plaque virus purified by restriction enzymes. After that, the hepatocytes were infected with adenovirus diluted in the culture medium, the number of infections was 50 times, and the infection was 24 h.

**Adeno-associated virus and lentivirus construction and production.** AAV8-TBG vector, a pre-packaged AAV in serotype 8 with overexpression of GFP, was used to produce recombinant AAV8-TBG-gene of interest-GFP expression vector. This vector contains transcriptional control elements from the thyroxine-binding globulin (TBG) promoter, cloning sites for the insertion of a complementary DNA, and the polyA signal. Terminal repeats from AAV serotype 2 flank the expression cassette. The murine full-length Trim31 sequences or Trim31 with RING domain deletion sequences were then cloned into AAV8-TBG-GFP, respectively. This newly created vector AAV-TBG-Trim31-GFP or AAV-TBG-Trim31 RING$^\Delta$-GFP was packaged into AAV8, purified by ViraBind™ AAV Purification Mega Kit (VPK-141/VPK-141-5, Cell Biolabs, VPK-141/VPK-141-5, San Diego, USA) and accordingly titered by QuickTiter™ AAV Quantitation Kit (Cell Biolabs, VPK-145). Viral particles were diluted to a total volume of 50 µl with saline immediately before injection.

To generate the lentiviral-Trim31 (LV-Trim31) or lentiviral-Trim31 with RING domain deletion (LV-Trim31 RING$^\Delta$) vectors, the full-length Trim31 cDNA sequences were packaged into pLenti-CMV-GFP-Puro (Addgene) to upregulate Trim31 expression (pLenti-CMV-Trim31-GFP-Puro or pLenti-CMV-Trim31 RING$^\Delta$-GFP-Puro) in vivo experiments. The commercial Lenti-Pac HIV Expression Packaging Kit (LT002, GeneCopoeia, MD, USA) and corresponding Lenti-Pac 293Ta Cell Line were used to produce LV particles. Next, according to the product instruction, the 293T cells culture supernatants containing virus particles were harvested. The newly created vector was concentrated and purified by ViraBind™ PLUS Lentivirus Concentration and Purification Kit (Cell Biolabs, VPK-095) and then titered by QuickTiter™ Lentivirus Quantitation Kit (Cell Biolabs, VPK-112). The functional LV titers in the 10$^6$ TU/ml range were achieved, and after concentration yields of up to 10$^9$ TU/ml were attained.

**Intracellular triglyceride levels assay**. Intracellular triglyceride (TG) levels were detected using commercial Triglyceride Assay Kit-Quantification (#ab65336, Abcam) according to the manufacturer's instructions.

**Metabolic indicators and serum cytokine parameters**. To perform intraperitoneal injection of glucose tolerance test (GTT) and insulin tolerance test (ITT), the mice involving this part of the experiment were fasted for 8 h to ensure the correction of physiological response. Mice were given an i.p. injection of glucose (2 g/kg BW) (#158968, Sigma-Aldrich, Shanghai, China). Then, the concentration of blood glucose of tail venous blood at 0, 15, 30, 60, and 120 min after glucose administration were detected using commercial blood glucose test strips (ACCU-CHEK®, Roche Diabetes Care GmbH, Shanghai, China). To examine the insulin tolerance test, mice were administrated with an intraperitoneal injection of insulin (1 U/kg BW, Sigma Aldrich). The GTT & ITT analysis procedures were performed in accordance with our previous reports[15]. Accordingly, blood samples were harvested from the tail vein at 0, 15, 30, 60, and 120 min post-injection for detection of glucose levels. Homeostasis model assessment (HOMA)-IR index was calculated from fasting levels of glucose and insulin in serum, respectively. The cytokines and chemokines of mice were evaluated using a corresponding commercial enzyme-linked immunosorbent assay (ELISA) kit. The TNF-α (#MTA00B), IL-1β (#MLB00C), IL-10 (#M1000B), IL-6 (#M6000B), and CCL-2 (#MJE00B) ELISA kits were purchased from R&D system (Shanghai, China) and used according to the product specification. All the corresponding serum were carefully stored at −80 °C refrigerator until used.

**Hepatic function parameters and liver lipid level detection**. The concentration of serum alanine transaminase (ALT) (#MAK052, Sigma-Aldrich), aspartate aminotransferase (AST) (#MAK055, Sigma-Aldrich), alkaline phosphatase (AKP) (#ab83369, Abcam), serum insulin (#ab277390, Abcam) and hepatic triglyceride (TG) (#MAK266, Sigma-Aldrich), total cholesterol (TC) (#ab65359, Abcam) and non-esterified free fatty acids (NEFA) (#E-BC-K014, Elabscience, Inc., Houston, USA) were detected using commercially-available detection kits in the indicated groups according to the product specification.

**Histopathologic examination**. To perform histologic and immunohistochemical analysis, the liver samples were accordingly fixed with 10% neutral formalin-histological tissue fixative (#HT501128, Sigma-Aldrich), embedded in paraffin (#YA0010, Solarbio Life Sciences, Beijing, China), and then sectioned transversely. The thin liver samples sections were stained with hematoxylin and eosin (H&E) (#abs9217, Hematoxylin and Eosin Staining Kit, Absin, Shanghai, China) to visualize the pattern of lipid deposition and injury of tissue. To show lipid accumulation in liver, the sections were frozen in Tissue-Tek optimum cutting temperature (O.C.T.) (#4583, Tissue-Tek, Sakura Finetek, USA) and then stained with Oil Red O Stain Kit (#ab150678, Abcam) for 10 min. After being rinsed with 60% isopropyl alcohol (#I9030, Sigma-Aldrich), the tissue sections were re-stained with hematoxylin. Furthermore, to visualize glycogen levels in liver samples, sections were stained with Periodic Acid-Schiff stain (PAS) (#G1281, Solarbio Life Sciences, Beijing, China). To perform immunohistochemistry assay, embedded sections were deparaffinized before treatment with primary antibodies including anti-Rhbdf2 (#orb386934, Biobyt, dilution 1:250), anti-Trim31 (#PA5-40961, dilution 1:200) or anti-p-MAP3K7 (#PA5-99340, dilution 1:200) at 4 °C overnight in the indicated groups. The corresponding goat anti-mouse or anti-rabbit IgG antibodies (Abcam) were used as the secondary antibody. All the histological procedure was performed in accordance with the standard procedures as indicated in reagent specifications and operating instructions and detected by 3 histologists without knowledge of the treatment procedure. The images were captured using an optical microscope (Olympus, Japan) for tissue slices observation and fluorescence microscope (Olympus, Japan) for immunofluorescence sections observation.

**Adipose tissue sampling and adipocyte size detection**. The white adipose tissue samples from NCD-fed Flox, THKO, NTG, and THTG mice, and HFD-fed Flox, THKO, NTG, and THTG mice were fixed with 4% paraformaldehyde, embedded in paraffin, and sectioned transversely. Adipose tissues were then subjected to hematoxylin and eosin (H&E) staining analysis. All of the sections were detected by 3 histologists without knowledge of the treatment procedure. The images were captured using an optical microscope (Olympus, Japan). The average cell diameter was examined using Image J software from the National Institutes of Health (NIH). The total adipose cell numbers were determined by counting the cells on the H&E sections from at least 3 fields per mouse. The size of the cells was approximated assuming cubic packing as previously indicated[13,37].

**Immunoprecipitation assay**. The immunoprecipitation assay was performed in the current study as previously reported[15,22,25]. In brief, the L02 cells were transiently transfected with corresponding vectors in the indicated groups using Lipofectamine™ 3000 Transfection Reagent (Invitrogen™) according to the manufacturer's instructions and were cultured for an additional 36 h. Subsequently, the cells were harvested and homogenized into immunoprecipitation (IP)-specific lysis solution (#87787, Pierce™ IP Lysis Buffer, Thermo Scientific Pierce) at 4 °C, followed by centrifugation at 12,000 × g for 20 min in a refrigerated centrifuge.

The collected cell lysates were incubated with Protein A/G Magnetic Agarose Beads (#78609, Thermo Scientific Pierce) at room temperature with mixing for 2 h, and then mixed with the indicated antibodies at 4 °C overnight. The immune compound was harvested after washing with immunoprecipitation buffer and subjected to western blotting assay by incubating with indicated primary antibodies and the corresponding secondary antibodies.

**Glutathione S-transferase (GST) pull-down assays**. Direct protein interaction binding between Trim31 and Rhbdf2 was performed using the GST pull-down assays as described previously[15,25]. The Pierce™ GST Protein Interaction Pull-Down Kit (#21516, Thermo Fisher Scientific) was used to help with this part of the experiment. In brief, the Rosetta (DE3) E.coli cells were transformed with the plasmid pGEX-4T-1-GST-Trim31 or pGEX-4T-1-GST-Rhbdf2 and then induced expression by incubating with 0.5 mM isopropyl β-D-thiogalactopyranoside (IPTG) (#I5502, Sigma-Aldrich). The extraction from lytic E.coli was mixed with GST beads at 4 °C for 1 h. The GST beads were then incubated with Flag-tagged Trim31 or Flag-tagged Rhbdf2, which were prepared by immunoprecipitation for the additional 4 h. Proteins that had interacted were eluted in elution buffer and were subjected to western blotting analysis using anti-Flag antibodies. The E. coli expressing only a GST-tag were used as the negative control.

**In vivo ubiquitination assays**. The in vivo ubiquitination analysis process was performed using a previously reported protocol[15,22,25]. In brief, the cells were transfected, cultured in the indicated groups, and then homogenized into immunoprecipitation (IP)-specific lysis solution (#87787, Pierce™ IP Lysis Buffer, Thermo Scientific Pierce) at 4 °C for sample preparation. The lysates containing 1% SDS solution were denatured by heating for 5 min at boiling water. After that, the supernatants were diluted tenfold with IP solution buffer. The samples were subjected to immunoprecipitation with the indicated antibodies.

**In vitro binding and ubiquitination assay**. Trim31, Rhbdf2, and Trim31 with RING domain deletion proteins were expressed with a TNT® Quick Coupled Transcription/Translation System (Promega) in accordance with the previous reports[15,22,25] and manufacturer's instructions. Protein interaction binding assays were performed by mixing corresponding Flag-tagged Rhbdf2 and Trim31 together, followed by immunoprecipitation with Flag antibody and immunoblotting with Trim31 antibody. Ubiquitination levels were analyzed with a ubiquitination kit (Boston Biochem) following protocols of the manufacturer's instructions.

**RNA extraction and real-time quantitative PCR**. Total RNA from cells or liver tissues was extracted using TRIzol reagent (#15596026, Thermo Fisher Scientific) according to procedures recommended by the manufacturer. Briefly, 1 µg of total RNA extraction was reverse transcribed using the M-MLV-RT system (Invitrogen). The program was performed at 42 °C for 1 h and terminated by deactivation of the enzyme at 70 °C for 10 min. PCR was conducted using SYBR Green (Bio-Rad) in ABI PRISM 7900HT detection systems (Applied Biosystems). The specific primer sequences for RT-PCR were produced by Sangon Biotech (Shanghai) Co., Ltd., and indicated in Supplementary Table 2. Fold induction values were calculated according to the $2^{(-\Delta\Delta Ct)}$ expression, where $\Delta Ct$ represents the differences in cycle threshold number between the target gene and GAPDH, and $\Delta\Delta Ct$ represents the relative change in the differences between control and treatment groups.

**Immunoblotting analysis**. To perform immunoblotting analysis, cells or liver tissues were homogenized into RIPA Lysis and Extraction Buffer (#9806, CST) to yield a homogenate. Next, the final liquid supernatants were concentrated by centrifugation at $13,000 \times g$, 4 °C for 30 min. Protein concentration was determined by Pierce™ Rapid Gold BCA Protein Assay Kit (Thermo Fisher Scientific) with bovine serum albumin as a standard. The total protein extraction samples were then subjected to western blotting analysis. Equal amounts of total protein of cells or tissues were subjected to 10 or 12% sodium dodecyl sulfate-polyacrylamide gel electrophoresis (SDS-PAGE) system and then transferred to a 0.45 µM PVDF membrane (Millipore Company, USA) followed by immunoblotting using the indicated primary antibodies. Next, the PVDF membranes were incubated with 5% skim milk (Difco™ Skim Milk, BD, USA) in 1×TBS buffer (#T1080, Solarbio, Beijing, China) containing 0.1% Tween-20 (#1247ML100, BioFROXX, Germany) (TBST) for 1 h and mixed with the primary antibodies at 4 °C overnight. Membranes were rinsed in 1×TBST 3 times and subsequently incubated with HRP-conjugated anti-rabbit or anti-mouse secondary antibodies (Abcam) for 1–2 h at room temperature (25 °C). Immunoblotting bands were visualized by Pierce™ ECL Plus Western Blotting Substrate (#32134, Thermo Fisher Scientific) and exposed to Kodak (Eastman Kodak Company, USA) X-ray film. Corresponding protein expression was then determined as grey value (Version 1.52 g, Mac OS X Snow Leopard, Image J, National Institutes of Health, USA) and standardized to housekeeping genes (GAPDH) and expressed as a fold of control.

**Statistical analysis**. The relevant experiments presented in the current study were performed independently at least three times. All data associated with this study were analyzed using an appropriate statistical approach, as shown in the figure

legends. Quantitative values of data were expressed as mean ± standard error of the mean unless indicated otherwise. Comparisons were analyzed by one-way analysis of variance (ANOVA) followed by Dunnett's multiple comparisons test for multiple groups or two-tailed Student's $t$ test for two groups. GraphPad Prism Software (Version 8.2.0 for Mac OS X Snow Leopard; Graph Pad Software, Inc., San Diego, CA) and SPSS Statistics Software (Version 26.0.0.2 for Mac OS X Snow Leopard; IBM, Inc., New York, USA) were used for the data analysis. A $P$ value <0.05 ($P < 0.05$) was considered to be statistically significant. Specifically, the animal experiment data were harvested blindly. No data were excluded in the final statistical analysis. A randomization process was performed in grouping mice with the same phenotypes.

**Reporting summary**. Further information on research design is available in the Nature Research Reporting Summary linked to this article.

## Data availability

The data that support the findings of this work are available from the corresponding author on request. There are no restrictions on data availability in the current work. Source data are provided with this paper.

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

## Acknowledgements

This work was supported by (1) National Natural Science Foundation of China (NSFC Grant No.: 81703527); (2) Chongqing Research Program of Basic Research and Frontier Technology (Grant No. cstc2018jcyjAX0811, cstc2018jcyjA3533); (3) Science and Technology Research Program of Chongqing Education Commission of China (Grant No.: KJQN201901608, KJQN201901615, KJZD-M201801601, KJZD-K202001603); (4) Chongqing Professional Talents Plan for Innovation and Entrepreneurship Demonstration Team (CQCY201903258, cstc2021ycjh-bgzxm0202); (5) School-level Research Program of Chongqing University of Education (Grant No.: 2019BSRC001); (6) Advanced Programs of Post-doctor of Chongqing (Grant No.: 2017LY39); (7) Supported by Youth Project of Science and Technology Research Program of Chongqing Education Commission of China (Grant No.: KJQN201901606). We are also very grateful to Mr. H. Chen and Mrs. Y. F. Li for warm technical help in the ex vivo gene therapy experiment preparation, equipment supply, and patients' sample processing and analysis.

## Author contributions

Conceptualization: M.-X.X., C.-F.Z., B.-C.W., J.T. Methodology: M.-X.X., W.D., B.-K.Z., X.-P.T., L.-C.Z., C.-X.G., C.-F.Z., B.-C.W., J.T. Investigation: D.-X.L., Q.K, S.-Y.Z., L.-L.L., C.Y., T.-T.T., J.-J.Z., L.-Y.W., J.L., H.W., Y.S., Q.-F.Y., Q.L., D.-S.L. Data acquisition & analysis: M.-X.X., L.-F.H., X.L., G.K., J.L., M.-X.X., J.F. Funding acquisition & Project administration: C.-F.Z., B.-C.W., J.T., M.-X.X. Supervision: C.-F.Z., B.-C.W., J.T. Writing—original draft: M.-X.X., W.D., B.-K.Z., X.-P.T., L.-C.Z., C.-X.G. Writing—review and editing: C.-F.Z., B.-C.W., J.T., M.-X.X.

## Competing interests

The authors declare no competing interests.
