## [Peer Review File · Nature Communications]

Reviewers' Comments:

Reviewer #1:

Remarks to the Author:

The authors of this article have studied the important roles of Trim31/Rhbdf2 signaling pathway in nonalcoholic fatty liver disease (NAFLD) and nonalcoholic steatohepatitis (NASH). They used hepatocyte-specific deletion of Trim31 or adenovirus mediated overexpression of Trim31 to demonstrate the function of Trim31 in NAFLD/NASH models via ubiquitination-dependent degradation of Rhbdf2 protein, which regulates insulin signaling pathway and other metabolic function. In addition, they used primary hepatocytes treated with palmitate (PA) or TNF-alpha to show the changes in the levels of Trim31 and its target Rhbdf2 protein in a reciprocal manner.

The authors showed expert skills in conducting many experiments shown in this study. However, the current manuscript contains many weak and/or inconsistent data that need to be corrected and/or revised.

General comments:

- 1) First of all, this manuscript requires extensive revision in English. For instance, many sentences contain redundant or duplicated words (e.g., in contrast, however, --- in two different places) in the same sentences.
- 2) There are many mistakes in describing the actual results in the text (see the specific points).
- 3) Additionally, some background information, such as the insulin signaling pathways and rationale to use palmitic acid as a model of fat accumulation in primary hepatocytes, are needed to help the general readers who may not be familiar with the specific field of liver disease or terminologies.
- 4) The results of the proper control experiments were not shown. For instance, the levels of Rhbdf2 after transfection with AdRhbdf2 or AdshRhbdf2 vector in Fig. 5D were not provided. The same comments can be made for the data shown in Fig. 6C and Fig. 7G after transfection with many expression vectors for different sizes of the truncated target proteins.
- 5) Throughout this study, the molecular weights of many target proteins were not shown. This needs to be corrected.

Specific comments:

- 6) Please describe the method of how the primary hepatocytes were treated with PA and what was the vehicle control (solvent) for this treatment.
- 7) Fig. 1: The levels of Trim31 and Rhbdf2 should be fluctuated in an opposite manner since Trim31 is a ubiquitin-E3-ligase for Rhbdf2 protein degradation. In fact, Trim31 levels were rapidly decreased at 1, 5, and 10-h in PA-treated hepatocytes (Fig. 1E). Thus, it is strange to see the decreased levels of Rhbdf2 protein at 10-h in PA-treated hepatocytes (instead of its elevated levels as expected by the reciprocal regulation between Trim31 and Rhbdf2) (Fig. 1F). Thus, the levels of Rhbdf2 at 1, 5, and 10-h in PA-treated hepatocytes need to be shown. Additionally, it is unclear why cycloheximide (CHX) treatment prevented the loss of Rhbdf2 at 5 h and then decreased at 10 h in PA-treated hepatocytes (Fig. 1G, top panel). This result needs clarification. Furthermore, the human data shown in the same row with Fig. 1E (the results with primary hepatocytes) should be relocated or rearranged into the human data. Finally, the concentrations of 10 nM NAC and 10 nM MG132 seem too low and need to be double-checked.
- 8) Fig. 2H: The levels of p-IRS1 (Ser307) and p-IRS1-(Tyr608) seem to be regulated in an opposite direction for the insulin signaling pathway. It would be highly desirable if the authors should briefly describe the opposite regulation of these proteins in insulin resistance with a proper reference(s).
- 9) Fig. 3K: The authors should have shown the proper control experiments to show the results of transfected AdshRNA control versus AdshTrim31 as well as AdGFP versus AdTrim31 in hepatocytes.
- 10) Fig. 5B: Immunoreactive bands of ADAM17 in THKO+HFD appear indifferent from those of Flox+HFD, if one compared the ADAM17/GAPDH levels.
- 11) Fig. 6: Please rearrange the data shown in Fig. 6C (shown in a vertical arrangement) to prevent any confusion with the results in Fig. 6G. In addition, it is unclear about the reason why similar levels of Rhbdf2 protein in the Trim31-floxed mice treated with HFD versus normal control diet were observed in Fig. 6D. This reviewer expected that higher levels of Ub-conjugated Rhbdf2 protein in HFD-fed Trim31-floxed mice might result in its proteasomal degradation, leading to

decreased amounts of its protein. Similar comments can be made for the Fig. 6F results.

12) Supplementary Fig. 1B: The immunoblot results clearly show that the level of Ub-conjugated Rbdf2 protein was markedly increased in PA-exposed hepatocytes compared to other lanes (top panel of the top figure). This result of increased levels of Ub-conjugation with an expectation of greater degradation of its protein is therefore in contrast with the similar levels of Rbdf2 protein (bottom lane and lysate in the bottom panel) as well as the elevated amounts of this protein in the patients of NAFLD and NASH (Supplementary Fig. 1A). These inconsistent results need a clear explanation.

13) Supplementary Fig. 3: It could be better if the Trim31 protein levels in other organs were shown (like Fig. S2).

14) Supplementary Fig. 4H: The immunoblot data and densitometric protein levels and mRNA amounts of G6Pase in THTG+HFD seem inconsistent.

15) Supplementary Fig. 6A: The densitometric protein levels of p-IKK β /IKK β in THKO+HFD mice seem incorrect since the immunoreactive IKK β levels in THKO+HFD were greater (~2-fold) than those of Flox+HFD. In addition, the immunoblot data and densitometric protein levels of p-NF-kB/NF-kB in THKO+Palmitate seem inconsistent since the levels of p-NF-kB in THKO+Palmitate were much lower (~50%) than those of Flox+Palmitate in Supplementary Fig. 6D. These inconsistent results need corrections.

Minor comments:

16) Line 97: Ref #1 published in 2015 does not seem to be "the latest studies." The authors should have included a more recent review article.

17) Lines 107 and 415: "More seriously" may be an improper word and need to be replaced.

18) Line 110: "perfectly valid pharmaceutical therapeutics" may be redundant and needs to be shortened.

19) Line 111: This sentence seems unclear.

20) Lines 117-118: The content of this part seems redundant with the previous sentence.

21) Lines 119-122: This is not a full sentence and needs correction.

22) Lines 122-124: "Our previous studies..." cited only 1 study.

23) Lines 127-130: "Studies..." cited only 1 study and needs correction.

24) Lines 128-129: From Ref #17, it should be "a proteasomal inhibitor MG-132" instead of "protease inhibitors".

25) Lines 132-134: The sentence is unclear.

26) Lines 142-143: This sentence seems unclear and needs to state the involvement of Trim31-mediated invasion and metastasis.

27) Line 144: "just" may be an improper word.

28) Line 182: "remarkably" may be an incorrect word and could be changed to "markedly" or "remarkably".

29) Lines 186-188: The results with Cycloheximide should have been described.

30) Lines 209-211: The HOMA-IR results shown in Fig. 2D were not described.

31) Lines 221-226: The results of some other proteins, such as p-AKT, p-GSK3beta and p-FOX1, shown in Fig. 2H and S4H should have been mentioned.

32) Lines 244-246: The context of corresponding controls seems unclear.

33) Line 253: the context of "normal levels", as "compared to its controls" seems unclear.

34) Lines 293-294: ---- downregulation of contents of hepatic TG, TC, and NEFA were greatly promoted in the THKO (LV+) mice (Fig. 4G). Correction: --- high contents of hepatic TG, TC and NEFA were greatly decreased in THKO (LV+) mice (Fig. 4G).

35) Line 299: ---- were markedly suppressed ---. Correction: ---- were markedly altered --- because some (e.g., p-IRS1-Tyr608, p-Akt, and IL-10) were increased.

36) Line 317-318: The results of some other proteins, such as Adam17 and TNFR1/2 proteins, shown in Fig. 5B and 5C should have been mentioned with respects to the status of NAFLD/NASH.

37) Line 327-329: The results of some other proteins, such as p-AKT, p-GSK3beta and p-FOX1, shown in Fig. 5D should have been described.

38) Lines 367-368: If HFD decreases Trim31, should it also reduce ubiquitination of Rbdf2?

39) Lines 373-375: A reference is needed when the previous study was mentioned.

40) Lines 389-390: Some other proteins shown in Fig. 6G should be described.

41) Line 434: misspelling of --- Trim31-regulated hepatic ---.

42) Line 452: redundancy: In addition, these data we obtained were also further determined ---.

43) Lines 462-465: A reference is needed for the HFHF diet when this diet was mentioned, instead

of being mentioned in 'line 745'.

- 44) Lines 471-473: It may be better to describe some more results shown in Fig. 8D to 8I.
- 45) Line 484: "reasonable diet" could be replaced with "healthy diets".
- 46) Line 495: "threat" may be an improper word and may be replaced with "intake" or "exposure".
- 47) Line 516: "three pathological phenotypes" seems unclear.
- 48) Line 524: The liver does not seem to be the largest immune organ in the body.
- 49) Line 533: "retrained" may be an improper word and could be replaced with "prevented" or "managed".
- 50) Line 571: There should be a discussion related to MAP3K7, e.g., what could happen to MAP3K7 when Rhd2 is ubiquitinated?
- 51) Line 578: it should be "IL-1 β " instead of "IL- β ".
- 52) Line 672: ---- two homology arms was ---. Correction: ---- two homology arms were ---.
- 53) Line 736: misspelling of --- 2x1011 vg of vector --- needs to be double-checked.

Reviewer #2:

Remarks to the Author:

The authors showed that liver-specific Trim31 ablation facilitated NAFLD and NASH-associated phenotypes in mice, whereas transgenic or ex vivo gene therapy-mediated Trim31 gain-of-function in mice liver with NAFLD or NASH phenotypes alleviated their phenotypes. They also showed that Rhd2 signaling was required for the protective effect of Trim31 in mice.

Identifying the regulators of NAFLD or NASH is of significant interest and if substantiated would have impact in this field and beyond. The experimental data presented partially provides a solid framework for how the hypothesis was derived. However, additional key direct data is required to support the primary claim of the paper.

Major concern:

1. Human TRIM31 and mouse Trim31 are not highly homologous. The lengths of TRIM31 and Trim31 are 425 and 507, respectively, and carboxy-terminal region is shorter in human TRIM31. When full-length two sequences are analyzed by Needleman-Wunsch alignment, the homology and identity are 54%, 40% respectively. Moreover, when amino-terminal RING and B-box deleted sequences (TRIM31(132-425) and Trim31(132-507)) are analyzed, the homology and identity are 46%, 32% respectively. So the results of mouse analysis should be carefully extrapolated to human pathophysiology. That is, the phenomena observed in mice may occur only in mice and may not be applicable to humans. Experiments validated with human materials appear to be only Figure 1D, 1K, 1L, Figure 6, and Supplementary Figure 1A. It seems to be necessary to perform the experiments using TRIM31 humanized mice, human TRIM31 tg mice, or adenovirus encoding human TRIM31.

2. Interaction between TRIM31 and RHD2 or ubiquitination of RHD2 by TRIM31 were only showed using human construct or human cell lines. Because of the low homology between human TRIM31 and mouse Trim31 genes, the authors should verify its interaction and ubiquitination using mouse system. Endogenous interaction between TRIM31 and RHD2 in the mice liver should be determined. Then, in vivo ubiquitination assays of RHD2 should be performed using the liver samples isolated from Flox- and THKO-mice. Moreover, the binding domains of TRIM31 and RHD2 were determined using both human and mouse constructs.

Figure 6C: It looks like there is no band of TRIM31-Flag CCdelta and RHD2-HA Taildelta. So we cannot evaluate whether these domains are required for binding.

3. Why were protein levels in the HFD feeding mice liver analyzed as fold of NCD? Moreover, why were protein levels in the primary hepatocytes treated with PA analyzed as fold of control? To verify the authors' conclusions, THKO-mice with HFD feeding should be compared with Flox-mice with HFD feeding, and THKO-mice with PA treatment should be compared with Flox-mice with PA treatment as well.

Figure 2H and supplementary figure 4H: Analyzing the data of NCD and HFD separately, G6Pase and PERCK levels in the liver of between Flox- and THKO-mice with HFD feeding seems to be largely unchanged. When analyzing similarly the data of NTG and THTG, whereas PERCK levels are

decreased, G6Pase levels are increased in the THKG-mice with HFD feeding. How does the authors explain these discrepancies? I also wonder how mRNA levels are analyzed. Like protein levels, were mRNA levels in the HFD feeding mice liver analyzed as fold of NCD? Supplementary figure 6A and 6B: Again, protein levels in the HFD feeding mice liver were analyzed as fold of NCD. Analyzing the data of NCD and HFD separately, p-IKKbeta/IKKbeta levels are decreased in THKO-mice with HFD feeding, whereas p-IkappaBalpha/IkappaBalpha seems to be largely unchanged, which are different from the author's claim. Similarly, about the analysis of mRNA, were mRNA levels in the HFD feeding mice liver analyzed as fold of NCD? Supplementary figure 6D: Protein levels in the primary hepatocytes treated with PA were analyzed as fold of control. Analyzing the data of control and PA separately, p-NFkappaB/NFkappaB levels are decreased in the THKO-mice with HFD feeding, which are different from the author's claim.

4. Figure 5D: Compared AdGFP with AdshRhbdf2, it looks like that there are not clear differences except p-IRS1(Ser307)/IRS1 in the PA/THKO background. So the conclusion in the page 12, lines 324-326 is not based on sufficient evidences.

Minor comments:

1. All immunoblot data: Indicating molecular weight markers for blots are informative.
2. Please use conventional gene/protein symbols, that is, human gene symbols are all in upper-case (e.g., TRIM31), whereas mice gene symbols are only the first letter in upper-case (e.g., Trim31).
3. Figure 4I right panel: The labels of p-IRS1(Tyr608)/IRS1 and p-IRS1(Ser307)/IRS1 may be misplaced with each other.
4. Page 15 line 400 and Page 16 lines 433-434: The authors used TRIM31-RING delta construction in Supplementary figure 8-10, which can still bind to RHBDF2. So the authors cannot verify whether TRIM31- RHBDF2 interaction is required for the protective function of TRIM31 from these data.
5. Supplementary figure 8: The title of the figure does not represent its contents.
6. Page 17 line460: not "MAP37K" but "MAP3K7"

Reviewer #3:

Remarks to the Author:

In this work, the authors aim at studying the role of the E3 ubiquitin ligase-tripartite motif containing protein 31 (Trim31) in non-alcoholic fatty liver disease (NAFLD) and nonalcoholic steatohepatitis (NASH) development in mice. They found that Trim31 can protect against diet-induced insulin resistance, liver steatosis, inflammation, and hepatic fibrosis by promoting degradation of Rhbdf2 by K48-linked polyubiquitination, which resulted in suppression of Rhbdf2-MAP3K7 signaling and downstream events cascades. Their results clearly demonstrate that Trim31 is a key suppressor of NAFLD/NASH and metabolic disorder and may represent a molecular target for the treatment of these diseases.

The study is novel, well-conducted and the conclusions are supported by in vivo data from several animal models and in vitro experiments.

The manuscript is generally well written, but I would recommend a careful edition by a native English speaker. The results are convincing, and the conclusions are appropriated

However, I have some concerns regarding specific points:

1- The title of Figure 1 and the corresponding text in the Results section "Trim31 activity is restrained in livers with hepatic steatosis" does not seem appropriated. The term "Trim31 activity" in the subheading may be confusing since no activity has been measured in the liver from mice or human patients. Indeed, in this figure, the measured parameter has been Trim31 protein expression, which has been correlated with increased levels of Rhbdf2 levels. Although, the authors further demonstrate a direct interaction of Trim31 with Rhbdf2, the authors should be more cautious in their conclusion at this point of the paper.

2- In the results section, line 181, "Also, Rhbdf2 has been shown to be regulated by ubiquitination modification 16. Consistent with these studies, we confirmed that Rhbdf2 levels were remarkably

increased in liver, as compared to that levels in non-steatosis samples". In the second part of the sentence the authors should clarify in which livers the Rhbsf2 levels were increased, the sentence must be better explained (steatosis and NASH livers?)

3- In Figure 1, the authors show results of Trim31 levels measured by immunoblotting from 4 mice. No details are given about the animals used to measure Trim31 levels in the liver after HFD. Are these results representative from a larger cohort of mice? How many animals have been fed with HFD?

In Animal experiment design 1, it is stated that "At the end of experimental period, the liver tissue samples were collected from mice to detect corresponding signaling events." What else have been measured to confirm that these mice are obese and with fatty liver? Similarly, it is stated that "Moreover, the dynamic expression levels from 0 to 16 weeks after HFD treatment indicated that Trim31 expression was gradually suppressed in the fatty liver (Fig. 1C)". However, expression levels should be correlated to a quantification of lipids in the liver to allow this affirmation. The authors should give more details about the obese and fatty liver phenotype of HFD-fed mice, indicate the number of mice used in this experiment, and quantify the degree of fat accumulation in the liver in these mice in order to correlate this parameter to Trim31 expression levels.

4- The authors observed that significantly lower expression levels of Trim31 were detected in livers from NASH patients than those in the patients only with simple steatosis (Fig. 1D). To what could be due these differences? A better description of the differences between patients with simple steatosis and NASH could help to understand these results. One can speculate that liver inflammation could be responsible for these differences. Do the authors have some information about inflammation in these patients? Moreover, the authors need to specify in Supplementary Table 1 the clinical parameters (and corresponding values for each patient) used to classify the patients as patients suffering simple steatosis or NASH.

5- While Rhbdf2 levels have been measured by immunoblotting in the liver from patients with simple steatosis and NASH (Supplementary Fig. S1A), the authors did not report Rhbdf2 levels in mice fed a HFD or in obese ob/ob mice. Have them been measured? It would also be of interest to confirm this result in different mouse models.

6- Figure 3J: In the main text, lines 251-253, it is stated that "Likewise, expression of fatty acid uptake and synthesis-related genes and fatty acid β -oxidation-related gene in THTG mice also tended to normal levels, as compared to its controls (Fig. 3J)". However, in the Figure it is not compared with normal levels, it is compared with NTG HFD. Therefore, the authors should add the values of chow diet fed mice or, otherwise, in the main text the comparison should be referred to NTG HFD, indicating that the expression of fatty acid uptake and synthesis-related genes are decreased and fatty acid β -oxidation-related gene are increased in THTG mice compared to NTG HFD.

7- In Supplementary Fig. S5A, the authors measure visceral fat weight and show that visceral fat weight was elevated in the HFD-fed THKO mice, but this elevation was significantly restrained in THTG mice with a marked change in the ratio of visceral fat weight to body weight (Supplementary Fig. S5A). However, no details are given in the Methods section about white adipose tissue sampling or the methodology used to measure adipose cell size. It is crucial to clearly specify the adipose tissue depots studied. It would also be of certain interest to compare visceral vs. subcutaneous fat weights.

The authors stated: "Meanwhile, adipose cells size was greatly upregulated between HFD-fed mice and NCD control, but no significant difference between HFD-fed THKO and THTG group (Supplementary Fig. S5B). This result further suggested that adipocyte proliferation might links to the higher visceral fat weight in HFD-fed mice than those in the controls." What do the authors mean with "controls"? This paragraph should be better explained.

8- In "Animal experiment design 3#" corresponding to section "Trim31-Rhbdf2 interaction is required for Trim31-regulated hepatic steatosis and inflammation", the authors indicate that to specifically overexpress Trim31 in hepatocytes in vivo, AAV8 vectors were used. The Trim31-encoding AAV8 vectors were delivered via tail vein. What was the promoter used in the expression cassette to drive expression of Trim31 and Trim31 RING Δ ? In case the promoter used is not

hepatic-specific, authors need to report mRNA/protein expression levels of Trim31 and Trim31 RING Δ in liver and in off-target tissues (including tissues for which AAV8 shows tropism when administered systemically such as adipose tissue, pancreas, skeletal muscle...).

In line 733, the authors state that "the opening reading frame (ORF) encoding Trim31 without a stop codon was cloned into AAV8 vector to generate AAV-Trim31". Please, clarify why a stop codon was not used. Did the expression cassette bear a polyA sequence?

Regarding AAV8 vectors used as control, please clarify if GFP-encoding vectors or empty vectors were used. In line 737 and Supplementary Figure 10 legend the authors specify that "The empty vector was injected into mice as the corresponding control". However, in line 423 and Supplementary Figure 10, AAV-GFP were used. In addition, please, clearly indicate components of the expression cassette (i.e. promoter used, transgene).

The authors also need to specify methods used to produce, purify and titer AAV vectors used in the study.

Please, also indicate expression cassettes and methods used to produce, purify and titer lentiviral vectors used in the manuscript.

In line 420, the authors indicate "adeno-associated virus subtype 8". The term "subtype" should be substituted by the term "serotype", already used by the authors in the Methods section.

9- In Figure 8, the authors aim to investigate the therapeutic feasibility and effect of targeting Trim31-Rhbdf2-MAP37K pathway in the development and progression of NASH. However, in this part of the work, the authors studied the effect of Trim31 overexpression in THTG mice. Thus, they do not properly evaluate the therapeutic feasibility and effect of targeting Trim31-Rhbdf2 administered signaling as suggested by the subheading. They rather studied the effect of Trim31 overexpression in a model of NASH. In a translational point of view, this strategy certainly goes further than the use of a NAFLD model induced by HFD, but the conclusions of this experiment are limited because of the use of a transgenic model. A gene therapy approach using AAV vectors would be more appropriated to properly evaluate the therapeutic potential of Trim31 targeting. Importantly, in the last sentence of the results part, line 480, the authors indicate that "Collectively, our data revealed the efficacy of Trim31 in the treatment of steatohepatitis and associated metabolic syndrome in murine." In the major part of their experiments they demonstrate prevention of steatohepatitis but not reversion. Only, in the ex vivo gene therapy intervention approach by lentivirus-loading full-length Trim31 sequences (Figure 4) the overexpression of Trim31 is induced 8 weeks after the HFD-feeding. However, in this experiment they should show that in the moment of the injection of the transduced hepatocytes to the recipient mice, the steatosis is already established, and that after the injection of the vectors there is an amelioration of the phenotype. Only in this case they can postulate that Trim31 revealed efficacy in the treatment of steatohepatitis.

10- In the Discussion section (line 493), the authors stated that "we identified liver Trim31 as a markedly negative regulator of high-energy diet (HFD/HFHF)-stimulated or genetically induced chronic inflammation, insulin resistance, liver steatosis and NASH."

This sentence is confusing, and I would suggest to rather state: we identified liver Trim31 as a markedly protective regulator against high-energy diet (HFD/HFHF)-stimulated or genetically induced chronic inflammation, insulin resistance, liver steatosis and NASH.

Response to the Reviewers' Comments

To Reviewer #1:

The authors of this article have studied the important roles of Trim31/Rhbd2 signaling pathway in nonalcoholic fatty liver disease (NAFLD) and nonalcoholic steatohepatitis (NASH). They used hepatocyte-specific deletion of Trim31 or adenovirus mediated overexpression of Trim31 to demonstrate the function of Trim31 in NAFLD/NASH models via ubiquitination-dependent degradation of Rhbd2 protein, which regulates insulin signaling pathway and other metabolic function. In addition, they used primary hepatocytes treated with palmitate (PA) or TNF-alpha to show the changes in the levels of Trim31 and its target Rhbd2 protein in a reciprocal manner.

The authors showed expert skills in conducting many experiments shown in this study. However, the current manuscript contains many weak and/or inconsistent data that need to be corrected and/or revised.

We appreciate the reviewer's compliment on our manuscript's importance that "showed expert skills in conducting many experiments shown in this study".

General comments:

Question 1:

First of all, this manuscript requires extensive revision in English. For instance, many sentences contain redundant or duplicated words (e.g., in contrast, however, --- in two different places) in the same sentences.

Response 1:

Thank you so much for your efforts to review and improve our work. According to your comments, we have re-organized and corrected the statement in this manuscript with extensive revision in English to ensure the accuracy and stringency of our conclusions.

Also, the whole manuscript has been subjected to professional language editing for extensive revision (LetPub, London, Co., Ltd).

Based on the following revision, we hope the present revision meets your requirements.

Question 2:

There are many mistakes in describing the actual results in the text (see the specific points).

Response 2:

We thank the reviewer for pointing out the issue. We have corrected and revised results section in the text (Marked with blue color) according to your specific points.

Question 3:

Additionally, some background information, such as the insulin signaling pathways and rationale to use palmitic acid as a model of fat accumulation in primary hepatocytes, are needed to help the general readers who may not be familiar with the specific field of liver disease or terminologies.

Response 3:

Thank you very much for your comments. The necessary background knowledge and information regarding insulin signaling pathways and rationale to use palmitic acid as a model of fat accumulation in primary hepatocytes have been included and added in discussion section (Marked with blue color in text) to make our paper easier to understand for general readers.

Description in revised manuscript:

Previous studies indicated that increased release of free fatty acids from adipose cells results in triacylglycerol (TG) accumulation in hepatocytes, which may progress into steatosis and steatohepatitis^{13, 15}. It was also have shown that saturated fats, including palmitic acid (PA) and stearic acid, are more toxic than other fatty acids and promote hepatocyte toxicity in steatohepatitis models. Significant deposition of TG in the liver was caused by a disturbance of the lipid metabolism equilibrium. The metabolic disturbance not only promoted steatosis and hepatocyte injury, but it also markedly suppressed insulin signaling and facilitated insulin resistance.

- *Ref 1: Xu, Minxuan, et al., Functional loss of inactive rhomboid-like protein 2 mitigates obesity by suppressing pro-inflammatory macrophage activation-triggered adipose inflammation. Mol. Metab. 34, 112-123 (2020).*
- *Ref 2: Xu, Minxuan, et al., iRhom2 Promotes Hepatic Steatosis by Activating MAP3K7-Dependent Pathway. Hepatology 73, 1346-1364 (2021).*

Question 4:

The results of the proper control experiments were not shown. For instance, the levels of Rhbdf2 after transfection with AdRhbdf2 or AdshRhbdf2 vector in Fig. 5D were not provided.

The same comments can be made for the data shown in Fig. 6C and Fig. 7G after transfection with many expression vectors for different sizes of the truncated target proteins.

Response 4:

Thank you so much for your comments here. The levels of Rhbdf2 after transfection with AdRhbdf2 or AdshRhbdf2 vector were showed below. This result was also included in revised Supplementary Fig. 9a.

In addition, the results of the proper control experiments data for Fig. 6C and Fig. 7G after transfection for target proteins have been indicated below:

revised Supplementary Fig. 9b

revised Supplementary Fig. 15

Question 5:

Throughout this study, the molecular weights of many target proteins were not shown. This needs to be corrected.

Response 5:

The molecular weights of corresponding target proteins in the current work have been included in all figures and supplementary figures.

Specific comments:

Question 6:

Please describe the method of how the primary hepatocytes were treated with PA and what was the vehicle control (solvent) for this treatment.

Response 6:

According to your suggestions, description regarding “the method of how the primary hepatocytes were treated with PA and what was the vehicle control (solvent) for this treatment” has been included in Method-Cell Culture and Treatment section.

The corresponding references were displayed below.

Description in revised manuscript:

To construct a cell model of lipid deposition *in vitro*, corresponding concentration of palmitic acid (PA) (dissolved in 0.5% fatty acid-free BSA) was prepared and obtained according to previous reports. Then, the primary hepatocytes or L02 cells were treated with cell culture medium-containing PA for 10 h or 4 h. Fatty acid-free BSA (0.5%) alone was used as a control.

- *Ref 1: Xu, Minxuan, et al., iRhom2 Promotes Hepatic Steatosis by Activating MAP3K7-Dependent Pathway. Hepatology 73, 1346-1364 (2021).*
- *Ref 2: Liu, Dan, et al., TNFAIP3 interacting protein 3 overexpression suppresses nonalcoholic steatohepatitis by blocking TAK1 activation. Cell. metab. 31, 726-740 (2020).*

Question 7:

Fig. 1: The levels of Trim31 and Rhbdf2 should be fluctuated in an opposite manner since Trim31 is a ubiquitin-E3-ligase for Rhbdf2 protein degradation. In fact, Trim31 levels were rapidly decreased at 1, 5, and 10-h in PA-treated hepatocytes (Fig. 1E). Thus, it is strange to see the decreased levels of Rhbdf2 protein at 10-h in PA-treated hepatocytes (instead of its elevated levels as expected by the reciprocal regulation between Trim31 and Rhbdf2) (Fig. 1F). Thus, the levels of Rhbdf2 at 1, 5, and 10-h in PA-treated hepatocytes need to be shown. Additionally, it is unclear why cycloheximide (CHX) treatment prevented the loss of Rhbdf2 at

5 h and then decreased at 10 h in PA-treated hepatocytes (Fig. 1G, top panel). This result needs clarification. Furthermore, the human data shown in the same row with Fig. 1E (the results with primary hepatocytes) should be relocated or rearranged into the human data. Finally, the concentrations of 10 nM NAC and 10 nM MG132 seem too low and need to be double-checked.

Response 7:

We thank the reviewer for pointing out this issue.

“Reviewer comment: Thus, it is strange to see the decreased levels of Rhbdf2 protein at 10-h in PA-treated hepatocytes (instead of its elevated levels as expected by the reciprocal regulation between Trim31 and Rhbdf2) (Fig. 1F). Thus, the levels of Rhbdf2 at 1, 5, and 10-h in PA-treated hepatocytes need to be shown”.

Actually, in Fig. 1f, western blotting bands showed the change of Trim31 expression, not the change of Rhbdf2 expression. Additionally, we have also detected the levels of Rhbdf2 protein at 1, 5, and 10-h in PA-treated hepatocytes (see below). This result has been included in revised Supplementary Fig. 1e.

Indeed, the increased levels of Rhbdf2 by PA treatment suggested that its elevated levels as expected by the reciprocal regulation between Trim31 and Rhbdf2.

“Reviewer comment: Additionally, it is unclear why cycloheximide (CHX) treatment prevented the loss of Rhbdf2 at 5 h and then decreased at 10 h in PA-treated hepatocytes (Fig. 1G, top panel). This result needs clarification.

Cycloheximide (CHX) as a protein synthesis inhibitor in eukaryotes was used in our study to determine the inhibition of Rhbdf2 protein expression upon PA treatment. Indeed, compared

to the control group (0 h), the increase of Rhbdf2 at 5 h was inhibited, and then reduced at 10 h. The dose of CHX used in this cell experiment is 50 µg/ml.

In fact, prolonged PA treatment is able to markedly promote Rhbdf2 protein expression and induce hepatocytes injury. Increased Rhbdf2 levels significantly induce shedding of TNF- α and inflammatory response. Also, although the inhibition of protein synthesis by CHX was dose-dependent, previous studies have confirmed that CHX and increased TNF- α possessed a synergistic cytotoxicity, and consequently they are routinely used together to induce cell injury and death. Therefore, decreased of Rhbdf2 at 10 h upon PA+CHX treatment was attributable to CHX+TNF- α -triggered cytotoxicity and cell injury.

Also, as an inhibitor of protein translation, CHX is able to suppress PA-induced translation of Rhbdf2 protein and its production. Importantly, the stability of Rhbdf2 protein is dependent on the ubiquitination pathway. Consistent with this, Fig 1h and 1i further indicated that overexpression of Rhbdf2 by PA treatment was subjected to proteasomal degradation. Proteasomal degradation of Rhbdf2 was significantly suppressed by MG132 treatment. These phenomenon further revealed that partly, decrease of Rhbdf2 in Fig 1g (10 h), is associated with greater degradation of Rhbdf2.

“Reviewer comment: Furthermore, the human data shown in the same row with Fig. 1E (the results with primary hepatocytes) should be relocated or rearranged into the human data”. Finally, the concentrations of 10 nM NAC and 10 nM MG132 seem too low and need to be double-checked.

According to your comment, the human data shown in the same row with Fig. 1e (the results with primary hepatocytes) have been rearranged into the human data.

- Ref 1: Schneider-Poetsch, T. et al. (2010) Nat Chem Biol 6, 209-217.
- Ref 2: Klinge, S. et al. (2011) Science 334, 941-8.
- Ref 3: Nolop, K.B. and Ryan, U.S. (1990) Am J Physiol 259, L123-9.
- Ref 4: Reid, T.R. et al. (1989) J Biol Chem 264, 4583-9.

In addition, according to your concerns regarding the concentrations of NAC and MG132 used in our work, we have examined the original experimental records carefully and are very sorry for our carelessness for the dose unit. Actually, the dose of NAC and MG132 used in

the corresponding experiments are 10 μ M NAC and 10 μ M MG132. This issue has been double-checked and corrected in revised work.

Therefore, we also performed additionally *in vitro* experiments using 100 μ M NAC and 20 μ M MG132 to further determine our data.

Question 8:

Fig. 2H: The levels of p-IRS1 (Ser307) and p-IRS1-(Tyr608) seem to be regulated in an opposite direction for the insulin signaling pathway. It would be highly desirable if the authors should briefly describe the opposite regulation of these proteins in insulin resistance with a proper reference(s).

Response 8:

We thank the reviewer for pointing out this issue.

In response to a continuous challenge with high fat diet, Rhbdf2 expression is elevated in hepatocytes and phosphorylate MAP3K7, leading to the activation of downstream NF- κ B and MKK-JNK signaling, thereby inducing inflammatory response. The activated NF- κ B and MKK-JNK signaling further facilitated insulin resistance by promoting phosphorylation of Ser307 in IRS. The increased p-IRS (Ser307) significantly blocked interactions with the insulin receptor and inhibits insulin action, and disrupted its downstream AKT-GSK3 β phosphorylation, which contribute to glucose metabolic disorder.

Consistent with previous studies, in liver, prolonged HFD elicited a significant decrease of IRS-1 phosphorylation on Tyr608 and increase of Ser307 phosphorylation, indicative of IRS-1 inactivation. Also, IRS-1 phosphorylation at Tyr608 is associated with its activation, whereas phosphorylation at Ser307 is associated with inactivation of IRS-1; phosphorylation of IRS-1 at Ser307 is likely accomplished by JNK signaling. These changes were accompanied by inflammatory responses in terms of increases in the expression of NF- κ B and activation of the MKK and JNK pathway.

- Ref 1: Wang, Pi-Xiao, et al. "Targeting CASP8 and FADD-like apoptosis regulator ameliorates nonalcoholic steatohepatitis in mice and nonhuman primates." *Nature medicine* 23.4 (2017): 439-449.
- Ref 2: Liu Zhigang, Patil Ishan Y, Jiang Tianyi et al. High-fat diet induces hepatic insulin resistance and impairment of synaptic plasticity. *PLoS One*, 2015, 10: e0128274.
- Ref 3: Wang Pi-Xiao, Zhang Xiao-Jing, Luo Pengcheng et al. Hepatocyte TRAF3 promotes liver steatosis and systemic insulin resistance through targeting TAK1-dependent signalling. *Nat Commun*, 2016, 7: 10592.

Question 9:

Fig. 3K: The authors should have shown the proper control experiments to show the results of transfected AdshRNA control versus AdshTrim31 as well as AdGFP versus AdTrim31 in hepatocytes.

Response 9:

According to you comments here, the proper control experiments for the results of transfected AdshRNA control versus AdshTrim31 as well as AdGFP versus AdTrim31 in hepatocytes were performed and showed below. This result has been included in revised Supplementary Fig. S3d.

Question 10:

Fig. 5B: Immunoreactive bands of ADAM17 in THKO+HFD appear indifferent from those of Flox+HFD, if one compared the ADAM17/GAPDH levels.

Response 10:

The western blotting bands of ADAM17 in THKO+HFD has been additionally re-performed. The new ADAM17 bands has been updated in revised Fig. 5b.

Question 11:

Fig. 6: Please rearrange the data shown in Fig. 6C (shown in a vertical arrangement) to prevent any confusion with the results in Fig. 6G. In addition, it is unclear about the reason why similar levels of Rhbdf2 protein in the Trim31-floxed mice treated with HFD versus normal control diet were observed in Fig. 6D. This reviewer expected that higher levels of Ub-conjugated Rhbdf2 protein in HFD-fed Trim31-floxed mice might result in its proteasomal degradation, leading to decreased amounts of its protein. Similar comments can be made for the Fig. 6F results.

Response 11:

According to your comments here, the Fig. 6c has been rearranged and updated in revised manuscript.

In addition, thank you very much for your concerns here. Our current data and previous study have indicated that under physiological conditions, the endogenous Rhbdf2 was observed to be markedly ubiquitinated with both K48 and K63 linkage upon stimulus administration. Indeed, the higher levels of Ub-conjugated Rhbdf2 protein in HFD-fed Trim31-floxed mice might partly result in its proteasomal degradation, leading to decreased amounts of its protein. However, prolonged HFD treatment dramatically reduced the Trim31 protein expression and its activity. Accordingly, the K63-linked ubiquitination of Rhbdf2 and its signaling might be activated, and K48-linked ubiquitination of Rhbdf2 might be significantly decreased.

In Fig. 6d-f, we performed this ubiquitination test again to further confirm the ubiquitination levels change of Rhbdf2 in HFD-fed Trim31-floxed mice. The HFD treatment significantly increased total ubiquitination of Rhbdf2 and Rhbdf2 expression in WCL. However, only K48-linked ubiquitination of Rhbdf2 was almost completely abolished in Trim31-deficient cells, and K63-linked ubiquitination of Rhbdf2 was not affected. Thus, we also performed additionally experiments to re-confirm the Rhbdf2 expression of WCL in Fig. 6f. The Rhbdf2 protein expression was increased in PA-treated WT and THKO L02 cells. These data are consistent with our other results.

➤ Zhang, Yiran, et al., Uev1A-Ubc13 catalyzes K63-linked ubiquitination of RHBDF2 to promote TACE maturation. Cell. Signal. 42, 155-164 (2018).

Question 12:

Supplementary Fig. 1B: The immunoblot results clearly show that the level of Ub-conjugated Rhbdf2 protein was markedly increased in PA-exposed hepatocytes compared to other lanes (top panel of the top figure). This result of increased levels of Ub-conjugation with an

expectation of greater degradation of its protein is therefore in contrast with the similar levels of Rhbdf2 protein (bottom lane and lysate in the bottom panel) as well as the elevated amounts of this protein in the patients of NAFLD and NASH (Supplementary Fig. 1A). These inconsistent results need a clear explanation.

Response 12:

According to your comments here, we have performed this experiment again to confirm this result. As we mentioned in Response 11, the HFD treatment *in vivo* or PA treatment *in vitro* upregulate Rhbdf2 protein levels, but markedly downregulate Trim31 protein levels. After PA treatment, increased Rhbdf2 protein in our additional experiment was observed. Also, the high levels of the PA-induced Rhbdf2 expression was significantly decreased by NAC or CAT administration. This data is consistent with the Ubiquitination of Rhbdf2 levels and elevated amounts of this protein in the patients of NAFLD and NASH.

Question 13:

Supplementary Fig. 3: It could be better if the Trim31 protein levels in other organs were shown (like Fig. S2).

Response 13:

Thank you so much for your concerns here. The Trim31 protein levels in other organs for THTG mice have been showed in revised Supplementary Fig. 3b.

Question 14:

Supplementary Fig. 4H: The immunoblot data and densitometric protein levels and mRNA amounts of G6Pase in THTG+HFD seem inconsistent.

Response 14:

Thank you very much for reviewer's concerns here. According to your comments, we have re-performed the western blotting analysis and qPCR assay for additional 3 times to determine the amounts of G6Pase in original liver tissue of the HFD-fed THTG mice.

Undoubtedly, the densitometric protein levels and mRNA amounts of G6Pase in liver displayed a consistent trend. The results for immunoblot data and mRNA expression data have been updated in revised Supplementary Fig. 4h.

Question 15:

Supplementary Fig. 6A: The densitometric protein levels of p-IKK β /IKK β in THKO+HFD mice seem incorrect since the immunoreactive IKK β levels in THKO+HFD were greater (~2-fold) than those of Flox+HFD. In addition, the immunoblot data and densitometric protein levels of p-NF-kB/NF-kB in THKO+Palmitate seem inconsistent since the levels of p-NF-kB in THKO+Palmitate were much lower (~50%) than those of Flox+Palmitate in Supplementary Fig. 6D. These inconsistent results need corrections.

Response 15:

According to your suggestions, here we have performed additionally experiment to re-evaluate the densitometric protein levels of p-IKK β /IKK β . The corrected IKK β band has been submitted and updated. Besides, the western blotting data and densitometric protein levels of p-NF-kB/NF-kB in THKO+Palmitate were also repeated for 3 times to confirm the consistent results. All the corrected bands have been updated in revised Supplementary Fig. 6a and Supplementary Fig. 6d.

Minor comments:

Question 16:

Line 97: Ref #1 published in 2015 does not seem to be “the latest studies.” The authors should have included a more recent review article.

Response 16:

Thank you very much for your significant comments. We have included more recent review articles for this section. The new references have been updated in revised manuscript.

- Ref 1: Huang Daniel Q., El-Serag Hashem B., Loomba Rohit., Global epidemiology of NAFLD-related HCC: trends, predictions, risk factors and prevention. Nat. Rev. Gastroenterol. Hepatol. 18, 223-238 (2021).
- Ref 2: Samuel Varman T., Shulman Gerald I., Nonalcoholic Fatty Liver Disease, Insulin Resistance, and Ceramides. N. Engl. J. Med. 381, 1866-1869 (2019).

Question 17:

Lines 107 and 415: "More seriously" may be an improper word and need to be replaced.

Response 17:

The improper word "More seriously" has been replaced with a more proper word in our manuscript.

Question 18:

Line 110: "perfectly valid pharmaceutical therapeutics" may be redundant and needs to be shortened.

Response 18:

The sentence "perfectly valid pharmaceutical therapeutics" has been corrected as "...therapeutics approach for NASH...".

Question 19-24:

19) Line 111: This sentence seems unclear.

20) Lines 117-118: The content of this part seems redundant with the previous sentence.

21) Lines 119-122: This is not a full sentence and needs correction.

22) Lines 122-124: "Our previous studies..." cited only 1 study.

23) Lines 127-130: "Studies..." cited only 1 study and needs correction.

24) Lines 128-129: From Ref #17, it should be "a proteasomal inhibitor MG-132" instead of "protease inhibitors".

Response 19-24:

19) This sentence has been corrected and marked with blue color in revised manuscript.

20) The content of this part has been reorganized and rearranged to make them more reasonable.

21) This sentence has been carefully corrected.

22) "Our previous studies..." has been corrected as "Our previous study..."

23) This part has been corrected.

24) According to the Ref #17, this sentence has been corrected as "a proteasomal inhibitor MG-132". The revised sentence in manuscript were marked with blue color.

Question 25-28:

25) Lines 132-134: The sentence is unclear.

26) Lines 142-143: This sentence seems unclear and needs to state the involvement of

Trim31-mediated invasion and metastasis.

27) Line 144: “just” may be an improper word.

28) Line 182: “remarkedly” may be an incorrect word and could be changed to “markedly” or “remarkably”.

Response 25-28:

25) This sentence has been corrected.

26) “Trim31-mediated invasion and metastasis” has been included in this part.

27) The word “just” has been deleted.

28) The word “remarkedly” in corresponding sentence has been changed as “markedly”.

Question 29-33:

29) Lines 186-188: *The results with Cycloheximide should have been described.*

30) Lines 209-211: *The HOMA-IR results shown in Fig. 2D were not described.*

31) Lines 221-226: *The results of some other proteins, such as p-AKT, p-GSK3beta and p-FOX1, shown in Fig. 2H and S4H should have been mentioned.*

32) Lines 244-246: *The context of corresponding controls seems unclear.*

33) Line 253: *the context of “normal levels”, as “compared to its controls” seems unclear.*

Response 29-33:

29) The results with Cycloheximide (CHX) have been described in our paper.

30) The HOMA-IR results shown in Fig. 2d have been described in our revised manuscript.

31) The results of some other proteins, such as phosphorylated AKT (p-AKT), glycogen synthase kinase 3 β (p-GSK3 β) and forkhead box O1 (p-FOXO1), shown in Fig. 2h and Supplementary 4h have been described.

32) The context of corresponding controls have been corrected.

33) The context of “normal levels”, as “compared to its controls” has been corrected.

Question 34-44:

34) Lines 293-294: *---- downregulation of contents of hepatic TG, TC, and NEFA were greatly promoted in the THKO (LV+) mice (Fig. 4G). Correction: --- high contents of hepatic TG, TC and NEFA were greatly decreased in THKO (LV+) mice (Fig. 4G).*

35) Line 299: *---- were markedly suppressed ---. Correction: ---- were markedly altered --- because some (e.g., p-IRS1-Tyr608, p-Akt, and IL-10) were increased.*

36) Line 317-318: *The results of some other proteins, such as Adam17 and TNFR1/2 proteins, shown in Fig. 5B and 5C should have been mentioned with respects to the status of NAFLD/NASH.*

- 37) Line 327-329: The results of some other proteins, such as p-AKT, p-GSK3beta and p-FOX1, shown in Fig. 5D should have been described.
- 38) Lines 367-368: If HFD decreases Trim31, should it also reduce ubiquitination of Rhbdf2?
- 39) Lines 373-375: A reference is needed when the previous study was mentioned.
- 40) Lines 389-390: Some other proteins shown in Fig. 6G should be described.
- 41) Lines 434: misspelling of --- Trim31-regulated hepatic ---.
- 42) Lines 452: redundancy: In addition, these data we obtained were also further determined ---.
- 43) Lines 462-465: A reference is needed for the HFHF diet when this diet was mentioned, instead of being mentioned in 'line 745'.
- 44) Lines 471-473: It may be better to describe some more results shown in Fig. 8D to 8I.

Response 34-44:

- 34) According to your comment, this sentence has been corrected in revised manuscript.
- 35) The correction: "...were markedly altered..." has been included.
- 36) The results of some other proteins, such as ADAM17 and TNFR1/2 proteins, shown in Fig. 5b and 5c have been described.
- 37) The insulin signaling indicators including p-AKT, p-GSK3 β and p-FOXO1 proteins have been mentioned in this part.
- 38) Prolonged HFD intake significantly increases Rhbdf2 levels in the liver tissue, and total ubiquitination levels of Rhbdf2. Meanwhile, previous study has indicated that Rhbdf2 could be modified via K63-linked ubiquitination upon TNF- α or LPS treatment. Consistent with these, in our current work, the endogenous Rhbdf2 was showed to be virtually ubiquitinated with K48 and K63 linkage in PA-challenged L02 cells. As expected, only K48-linked polyubiquitination of Rhbdf2 was dramatically decreased in Trim31-deficient L02 cells (THKO-L02), but K63 and other ubiquitylation types of Rhbdf2 were not influenced. These results also showed that HFD treatment was able to decrease Trim31 expression. The down-regulation of Trim31 only reduces K48-linked polyubiquitination of Rhbdf2 *in vivo* and *in vitro*.
- 39) The corresponding reference for this part has been included.
- 40) The other proteins shown in Fig. 6g have been described.
- 41) The misspelling has been corrected.
- 42) This sentence has been reorganized and corrected.
- 43) The corresponding reference for HFHF diet-induced NASH model has been included.
- 44) This part has been reorganized and revised to show more results.

Question 45-53:

- 45) Line 484: “reasonable diet” could be replaced with “healthy diets”.
- 46) Line 495: “threat” may be an improper word and may be replaced with “intake” or “exposure”.
- 47) Line 516: “three pathological phenotypes” seems unclear.
- 48) Line 524: The liver does not seem to be the largest immune organ in the body.
- 49) Line 533: “retrained” may be an improper word and could be replaced with “prevented” or “managed”.
- 50) Line 571: There should be a discussion related to MAP3K7, e.g., what could happen to MAP3K7 when Rhbdf2 is ubiquitinated?
- 51) Line 578: it should be “IL-1 β ” instead of “IL- β ”.
- 52) Line 672: ---- two homology arms was ---. Correction: ---- two homology arms were ---.
- 53) Line 736: misspelling of --- 2x10¹¹ vg of vector --- needs to be double-checked.

Response 45-53:

- 45) The “reasonable diet” has been replaced with “healthy diets” in this part.
- 46) The word “threat” has been replaced with “exposure” in this sentence.
- 47) This sentence has been reorganized and corrected to make it easier to read and understand.
- 48) This sentence has been corrected as “the liver is the most important metabolism/immune organ...”.
- 49) This word has been corrected as “prevented” in this sentence.
- 50) According to your comment, more discussion associated with the relation between ubiquitination of Rhbdf2 and MAP3K7 phosphorylation levels has been included in this part.
- 51) This misspelling has been corrected.
- 52) The “two homology arms was...” has been corrected as “two homology arms were...”
- 53) The misspelling of “2x10¹¹ vg of vector...” has been corrected as “2×10¹¹ vg of vectors”.

We gratefully thank for the precious time the reviewer spent making constructive remarks. We hope our responses and revision could meet your requirement.

To Reviewer #2:

The authors showed that liver-specific Trim31 ablation facilitated NAFLD and NASH-associated phenotypes in mice, whereas transgenic or ex vivo gene therapy-mediated Trim31 gain-of-function in mice liver with NAFLD or NASH phenotypes alleviated their phenotypes. They also showed that Rhd2 signaling was required for the protective effect of Trim31 in mice.

Identifying the regulators of NAFLD or NASH is of significant interest and if substantiated would have impact in this field and beyond. The experimental data presented partially provides a solid framework for how the hypothesis was derived. However, additional key direct data is required to support the primary claim of the paper.

We appreciate the reviewer's compliment on our manuscript's importance that "The experimental data presented partially provides a solid framework for how the hypothesis was derived".

Major concern:**Question 1:**

Human TRIM31 and mouse Trim31 are not highly homologous. The lengths of TRIM31 and Trim31 are 425 and 507, respectively, and carboxy-terminal region is shorter in human TRIM31. When full-length two sequences are analyzed by Needleman-Wunsch alignment, the homology and identity are 54%, 40% respectively. Moreover, when amino-terminal RING and B-box deleted sequences (TRIM31(132-425) and Trim31(132-507)) are analyzed, the homology and identity are 46%, 32% respectively. So the results of mouse analysis should be carefully extrapolated to human pathophysiology. That is, the phenomena observed in mice may occur only in mice and may not be applicable to humans. Experiments validated with human materials appear to be only Figure 1D, 1K, 1L, Figure 6, and Supplementary Figure 1A. It seems to be necessary to perform the experiments using TRIM31 humanized mice, human TRIM31 tg mice, or adenovirus encoding human TRIM31.

Response 1:

We are very grateful to the reviewers for providing us with very meaningful and important suggestions here. Indeed, the human TRIM31 and mouse Trim31 are not highly homologous. Our results have confirmed the protective effects of Trim31 on steatosis in mice model. Thus, according to your comments, the ex vivo experiment was also used to determine the protective function of human TRIM31 on HFD-induced liver steatosis.

The outline of ex vivo experiment was indicated below:

The 8 weeks HFD-fed-preconditioned Trim31-deficient mice (THKO) were injected with lentivirus-loading human full-length TRIM31 sequences (LV-hTRIM31) via the portal vein. After additional 8 weeks HFD treatment, the blood and liver tissue from transplanted mice were harvested for the further experiments.

Consistent with the same methods used in mice model and results of mice, in *ex vivo* cultured and transduced hepatocytes, the mice with hepatocyte-specific hTRIM31 gain-of-function (THKO)(LV+) did result in marked reduce in HFD-induced liver weight, body weight, hepatic function indicators AST and ALT, and hepatic lipid deposition, as compared to those of controls (THKO)(LV-). Furthermore, the THKO (LV+) mice also had lower blood glucose levels than those of THKO (LV-) controls, as confirmed by results from the GTT and ITT test. We further indicated that the high contents of hepatic TG, TC and NEFA were greatly decreased in THKO (LV+) mice. Also, hepatocytes with hTRIM31 restoration reduced genes expression regarding fatty acid uptake and synthesis process and promoted genes expression regarding fatty acid β -oxidation process in HFD-fed THKO (LV+) mice. The activated Rbbdf2-MAP3K7 signaling and inflammation were markedly altered in THKO (LV+) mice compared to controls, as determined by immunoblotting and decreased concentrations of pro-inflammatory mediators.

These results might be partly and possibly extrapolated to human pathophysiology, or provide some evidence for pathological research. The obtained data have been included in revised Supplementary Fig.8.

In addition, the THKO mice were further injected with LV-hTRIM31-RING^Δ and subjected to next experiments. Consistent with the data from mice model, human TRIM31 vector with RING domain deletion did not mitigate HFD-fed liver steatosis, inflammation and metabolism disorder. These data also indicated that the RING domain of Trim31 in mice and TRIM31 in human is essential for the protective function of E3 ubiquitin-protein ligase. The corresponding results have been included in revised Supplementary Fig.12.

Question 2-1:

Interaction between TRIM31 and RHBDF2 or ubiquitination of RHBDF2 by TRIM31 were only showed using human construct or human cell lines. Because of the low homology between human TRIM31 and mouse Trim31 genes, the authors should verify its interaction and ubiquitination using mouse system. Endogenous interaction between TRIM31 and RHBDF2 in the mice liver should be determined.

Response 2-1:

Thank you so much for your concern here. Because of the low homology between human TRIM31 and mouse Trim31 genes, according to your suggestion, the endogenous interaction between Trim31 and Rhbdf2 in the mice liver after HFD administration has been determined. This experiment was performed using Abcam's Immunoprecipitation Kit (ab206996, Abcam, Cambridge, UK) and included in the revised Supplementary Fig.9c.

**Question 2-2:**

Then, in vivo ubiquitination assays of RHBDF2 should be performed using the liver samples isolated from Flox- and THKO-mice.

Response 2-2:

According to your comment for this issue, the K48-linked ubiquitination of Rhbdf2 in the liver samples isolated from HFD-fed Flox- and THKO-mice was performed. The corresponding results are consistent with the data in ubiquitination assays of RHBDF2 in human L02. This data was also included in the revised Fig.6d.

Question 2-3:

Figure 6C: It looks like there is no band of TRIM31-Flag CCdelta and RHBDF2-HA Taildelta. So we cannot evaluate whether these domains are required for binding. Moreover, the binding domains of TRIM31 and RHBDF2 were determined using both human and mouse constructs.

Response 2-3:

Thank you so much for your concern here. In our current work, we have confirmed Trim31 as a Rhbdf2-related protein. Also, as we concluded in results section: TRIM31 directly interacts with RHBDF2, as a key member of the E3 ubiquitin ligase, human TRIM31 is mainly composed of three parts: N-terminal RING-finger domain, B-Box domain, and C-terminal coiled-coil (CC) domain (Fig. 6c). To confirm which domain of TRIM31 is mainly responsible for the interaction with RHBDF2, a series of truncated mutants of human TRIM31 with Flag-tagged vectors including wild-type (TRIM31-Flag WT), RING-finger domain ablation mutant (TRIM31-Flag RING^Δ), B-Box domain deletion mutant (TRIM31-Flag Box^Δ) and coiled-coil domain deletion mutant (TRIM31-Flag CC^Δ) vectors were then produced for the following binding experiments. The co-immunoprecipitation assays demonstrated that except for TRIM31-Flag CC^Δ, RHBDF2 was coprecipitated with TRIM31 WT, TRIM31 RING^Δ and TRIM31 Box^Δ. These results indicated that the coiled-coil domain (CC) contributes its capacity to bind to RHBDF2.

In addition, because the CC domain of TRIM31 and N-terminal (Tail) domain of RHBDF2 in human are essential for their interaction, we wondered whether there was a similar biological function in mouse. Thus, according to your suggestion mentioned above, the binding domains of mice Trim31 and Rhbdf2 were also determined using TRIM31-Flag CC^Δ, Rhbdf2-HA Tail^Δ, and their corresponding wild type vectors. Consistent with protein binding result, in transfected mice hepatocytes, Trim31 with coiled-coil domain (CC) deletion did not bind to Rhbdf2. This result has been included in revised Supplementary Fig.9c.

Question 3-1:

Why were protein levels in the HFD feeding mice liver analyzed as fold of NCD? Moreover, why were protein levels in the primary hepatocytes treated with PA analyzed as fold of control? To verify the authors' conclusions, THKO-mice with HFD feeding should be compared with Flox-mice with HFD feeding, and THKO-mice with PA treatment should be compared with Flox-mice with PA treatment as well.

Response 3-1:

Thank you very much for your questions here. We are so sorry for the misunderstanding and carelessness in corresponding figures. Actually, all the protein expression were determined as grey value (Image J) and then standardized to GAPDH and expressed as a fold of corresponding "control". The detailed statistical method regarding which one is the "control" for each figure and the meaning of the notation have been included at the end of the figures' legend. We have changed the markers on the Y-axis of all the histograms to prevent readers from this misunderstanding.

Unalterably, HFD-fed THKO-mice was compared with HFD-fed Flox-mice, and *in vitro*, THKO cells with PA treatment was compared with Flox cells with PA treatment. The corresponding symbol (* or #) was marked in groups with significant differences.

For instance, in revised Fig. 2h, The GAPDH was used as loading control. * $P < 0.05$ and ** $P < 0.01$ vs. Flox HFD groups.

In revised Fig.5d, $**P < 0.01$ vs. Palmitate/THKO-AdRhbd2 groups or Palmitate/THTG-AdRhbd2 groups.

All the marks of figures used in our current work have been carefully double-checked and corrected to follow its statistical comparison.

Question 3-2:

Figure 2H and supplementary figure 4H: Analyzing the data of NCD and HFD separately, G6Pase and PERCK levels in the liver of between Flox- and THKO-mice with HFD feeding seems to be largely unchanged. When analyzing similarly the data of NTG and THTG, whearas PEPCCK levels are decreased, G6Pase levels are increased in the THTG-mice with HFD feeding. How does the authors explain these discrepancies? I also wonder how mRNA levels are analyzed. Like protein levels, were mRNA levels in the HFD feeding mice liver analyzed as fold of NCD?

Response 3-2:

Thank you so much for your concerns for this part. According to your comments, we have re-performed the western blotting assay and re-analyzed the protein expression levels of

G6Pase and PEPCK in the liver tissue of between HFD-fed Flox and THKO mice. The additional western blotting bands have been included in revised Fig. 2 and Supplementary Fig. 4.

revised Fig. 2h

revised Supplementary figure 4h

Also, our current study have revealed that hepatocyte Trim31 exhibited protective function on against HFD feeding-induced insulin resistance and steatosis. In THKO mice, the impaired insulin signaling caused by HFD administration dramatically down-regulated p-FOXO1, but up-regulated its downstream pathway activity (i.e., PEPCK and G6Pase). The activated PEPCK and G6Pase significantly promoted gluconeogenesis and occurrence of glucose metabolic disorder. In contrast, after HFD challenge, the mice with overexpression of Trim31 (THTG) exhibited reduced PEPCK and G6Pase levels, as compared to HFD-fed NTG mice. The Trim31 overexpression was able to markedly decrease gluconeogenesis progression, and further protect against HFD treatment-triggered metabolic disorder.

In addition, the mRNA expression levels of target genes were normalized to GAPDH expression, and $\Delta\Delta Ct$ represents the relative change in the differences between “control” and treatment groups. The detailed statistical method regarding which one is the “control” for each figure and the meaning of the notation have been included at the end of the figures’ legend.

For example, the mRNA levels in THKO-mice with HFD feeding was compared with Flox-mice with HFD feeding. $**P < 0.01$ vs. Flox HFD groups.

Question 3-3:

Supplementary figure 6A: Again, protein levels in the HFD feeding mice liver were analyzed as fold of NCD. Analyzing the data of NCD and HFD separately, p-IKKbeta/IKKbeta levels are decreased in THKO-mice with HFD feeding, whereas p-IkappaBalpha/IkappaBalpha seems to be largely unchanged, which are different from the author's claim. Similarly, about the analysis of mRNA, were mRNA levels in the HFD feeding mice liver analyzed as fold of NCD?

Supplementary figure 6D: Protein levels in the primary hepatocytes treated with PA were analyzed as fold of control. Analyzing the data of control and PA separately, p-NFkappaB/NFkappaB levels are decreased in the THKO-mice with HFD feeding, which are different from the author's claim.

Response 3-3:

Thank you very much for your questions here. The protein levels (western blotting bands) in HFD-fed THKO mice and Flox mice were carefully double-checked.

The original band for p-IKK β /IKK β levels in HFD-fed THKO mice was increased, as compared to HFD-fed Flox mice.

Next, according to your kindly comments, we have re-performed this experiments and re-submitted our additional and more representative bands for p-IKK β /IKK β levels and p-I κ B α /I κ B α levels. The new bands have been updated in revised Supplementary Fig. 6a.

Supplementary figure 6a

Moreover, the mRNA expression levels of target genes were normalized to GAPDH expression, and $\Delta\Delta C_t$ represents the relative change in the differences between “control” and treatment groups.

For instance, the mRNA levels in THKO-mice with HFD feeding was compared with Flox-mice with HFD feeding. * $P < 0.05$ and ** $P < 0.01$ vs. Flox HFD groups.

In addition, in the revised Supplementary Fig. 6d, we also re-performed this immunoblotting experiments for this regard. We have confirmed that p-NF- κ B/NF- κ B levels was significantly increased in PA-treated THKO groups, as compared to PA-treated Flox groups. Meanwhile, there is no significant difference between Control-Flox and Control-THKO groups in this part. The new representative bands have been updated and submitted for your consideration.

p-NF- κ B/NF- κ B of revised Supplementary figure 6d

figure 6d

Question 4:

Figure 5D: Compared AdGFP with AdshRhbdf2, it looks like that there are not clear differences except p-IRS1(Ser307)/IRS1 in the PA/THKO background. So the conclusion in the page 12, lines 324-326 is not based on sufficient evidences.

Response 4:

According to your suggestions, the corresponding protein levels i.e., p-AKT, p-GSK3 β and p-FOXO1 in the PA/THKO background were retested and re-performed using western

blotting assay, and then were subjected to additional evaluation. The new immunoblotting bands were updated in revised Fig. 5d.

revised Fig. 5d

Minor comments:

Question 1:

All immunoblot data: Indicating molecular weight markers for blots are informative.

Response to minor comment 1:

The molecular weights of corresponding target proteins in the current work have been included in all figures and supplementary figures.

Question 2:

Please use conventional gene/protein symbols, that is, human gene symbols are all in upper-case (e.g., TRIM31), whereas mice gene symbols are only the first letter in upper-case (e.g., Trim31).

Response to minor comment 2:

The conventional gene/protein symbols regarding Trim31 or TRIM31 used in our current work have been corrected and revised.

Question 3:

Figure 4I right panel: The labels of p-IRS1(Tyr608)/IRS1 and p-IRS1(Ser307)/IRS1 may be misplaced with each other.

Response to minor comment 3:

The labels of p-IRS1(Tyr608)/IRS1 and p-IRS1(Ser307)/IRS1 in Fig. 4i have been corrected. We are so sorry for this mistake.

Question 4:

Page 15 line 400 and Page 16 lines 433-434: The authors used TRIM31-RING delta construction in Supplementary figure 8-10, which can still bind to RHBDF2. So the authors cannot verify whether TRIM31- RHBDF2 interaction is required for the protective function of TRIM31 from these data.

Response to minor comment 4:

Thank you so much for your question here. As we mentioned in Response 2-3 to your Question 2-3, E3 ubiquitin ligase Trim31 has three key domains for its biological function: the RING domain, B-box domain and coiled-coil domain (CC). Our experimental results have indicated that except for TRIM31-Flag CC^Δ, RHBDF2 was coprecipitated with TRIM31 WT, TRIM31 RING^Δ and TRIM31 Box^Δ. These results confirmed that the coiled-coil domain (CC) of Trim31 contributes its capacity to bind to RHBDF2, and also means Trim31-Rhbdf2 interaction is required for the protective function of Trim31 in HFD-induced liver steatosis. The similar results were further observed in mice hepatocytes in vitro experiments.

Question 5:

Supplementary figure 8: The title of the figure does not represent its contents.

Response to minor comment 5:

The title of the this supplementary figure has been corrected as: “Allogeneic hepatocyte transplantation using lentivirus-mediated RING domain deletion of Trim31 did not alleviate HFD-triggered impaired insulin signaling and hepatic steatosis”.

Question 6:

Page 17 line460: not “MAP37K” but “MAP3K7”

Response to minor comment 6:

The misspelling in corresponding section has been corrected.

We gratefully appreciate for your valuable suggestions in improving our work. We hope our corrections in this current revised manuscript could meet your requirement.

To Reviewer #3:

In this work, the authors aim at studying the role of the E3 ubiquitin ligase-tripartite motif containing protein 31 (Trim31) in non-alcoholic fatty liver disease (NAFLD) and nonalcoholic steatohepatitis (NASH) development in mice. They found that Trim31 can protect against diet-induced insulin resistance, liver steatosis, inflammation, and hepatic fibrosis by promoting degradation of Rhbdf2 by K48-linked polyubiquitination, which resulted in suppression of Rhbdf2-MAP3K7 signaling and downstream events cascades. Their results clearly demonstrate that Trim31 is a key suppressor of NAFLD/NASH and metabolic disorder and may represent a molecular target for the treatment of these diseases.

The study is novel, well-conducted and the conclusions are supported by in vivo data from several animal models and in vitro experiments.

The manuscript is generally well written, but I would recommend a careful edition by a native English speaker. The results are convincing, and the conclusions are appropriated.

However, I have some concerns regarding specific points:

We appreciate the reviewer's compliment on our manuscript's importance that "The study is novel, well-conducted and the conclusions are supported by in vivo data from several animal models and in vitro experiments".

Question 1:

The title of Figure 1 and the corresponding text in the Results section "Trim31 activity is restrained in livers with hepatic steatosis" does not seem appropriated. The term "Trim31 activity" in the subheading may be confusing since no activity has been measured in the liver from mice or human patients. Indeed, in this figure, the measured parameter has been Trim31 protein expression, which has been correlated with increased levels of Rhbdf2 levels. Although, the authors further demonstrate a direct interaction of Trim31 with Rhbdf2, the authors should be more cautious in their conclusion at this point of the paper.

Response 1:

Thank you so much for your kindly comments here. Accordingly, we have carefully re-checked and corrected the corresponding text as "Trim31 expression is downregulated in livers with hepatic steatosis" and "Trim31 expression is restrained in livers with hepatic steatosis". In addition, in Discussion section, the concerns you mentioned above have been included in our text as a possible study limitation.

Question 2:

In the results section, line 181, “Also, Rhbdf2 has been shown to be regulated by ubiquitination modification 16. Consistent with these studies, we confirmed that Rhbdf2 levels were remarkably increased in liver, as compared to that levels in non-steatosis samples”. In the second part of the sentence the authors should clarify in which livers the Rhbdf2 levels were increased, the sentence must be better explained (steatosis and NASH livers?)

Response 2:

We are sorry for the statement of this confusing sentence. This sentence has been double-checked and corrected as “Consistent with these studies, we confirmed that Rhbdf2 levels were remarkably increased in livers of human patients with NASH and simple steatosis phenotypes, as compared to that levels in non-steatosis samples.”

Question 3-1:

In Figure 1, the authors show results of Trim31 levels measured by immunoblotting from 4 mice. No details are given about the animals used to measure Trim31 levels in the liver after HFD. Are these results representative from a larger cohort of mice? How many animals have been fed with HFD?

Response 3-1:

Thank you so much for your question here. In Figure 1 legend, “*n*=4 per experiment” was included in statistics and used as sample size in the corresponding subfigures. In western blotting bands, each lane represents a random liver sample of mice.

For instance, in Fig.1a, a total of 15 wild-type C57BL/6N mice were used and fed with HFD fodder. Also, a total of 15 wild-type C57BL/6N mice were fed with NCD diet as control. At the end of experiment period, four random liver samples were selected from all sacrificial mice, and subjected to western blotting assay. The other liver samples were harvested and stored for addition experiments.

In Fig. 1b, a total of 10 *ob/ob* mice were included in this part. In Fig. 1c, respectively, 10 mice for each time point were included in this part.

Total number of experimental animals used in this part has been included in Method section.

Question 3-2:

In Animal experiment design 1, it is stated that “At the end of experimental period, the liver tissue samples were collected from mice to detect corresponding signaling events.” What else have been measured to confirm that these mice are obese and with fatty liver? Similarly, it is stated that “Moreover, the dynamic expression levels from 0 to 16 weeks after HFD treatment

indicated that *Trim31* expression was gradually suppressed in the fatty liver (Fig. 1C).”. However, expression levels should be correlated to a quantification of lipids in the liver to allow this affirmation. The authors should give more details about the obese and fatty liver phenotype of HFD-fed mice, indicate the number of mice used in this experiment, and quantify the degree of fat accumulation in the liver in these mice in order to correlate this parameter to *Trim31* expression levels.

Response 3-2:

Thank you so much for your comments here. To confirm that the WT mice we used in our study were obese and with fatty livers, histological analysis of the livers and quantification of lipids in the liver were included in this part.

As shown in this figure, prolonged HFD treatment (16 weeks) significantly increased liver lipid accumulation and promoted progression of hepatic steatosis, as compared to NCD-fed mice. This figure was also included in revised Supplementary Fig. 1b ($n=15$ for HFD-fed mice; $n=15$ for NCD-fed mice)

Question 4:

The authors observed that significantly lower expression levels of *Trim31* were detected in livers from NASH patients than those in the patients only with simple steatosis (Fig. 1D). To what could be due these differences? A better description of the differences between patients with simple steatosis and NASH could help to understand these results. One can speculate that liver inflammation could be responsible for these differences. Do the authors have some

information about inflammation in these patients? Moreover, the authors need to specify in Supplementary Table 1 the clinical parameters (and corresponding values for each patient) used to classify the patients as patients suffering simple steatosis or NASH.

Response 4:

Thank you very much for your concerns. Significantly reduced expression levels of Trim31 were observed in livers from NASH patients than those in the patients only with simple steatosis. Consistent with our further results, in mice model, liver inflammation induced by long-term HFD diet administration could help to determine these differences. Also, NASH-associated liver inflammation could be responsible for these differences. To further and better understand the role of inflammation in progression of NASH pathology, according to the Guidelines of NAFLD by AASLD in 2017, more information regarding inflammation (circulating TNF- α and IL-6 levels) in these patients and corresponding clinical parameters of diagnostic grading (NAFLD activity score, NAS) for simple steatosis or NASH were included in revised Supplementary Table 1.

- Ref: Chalasani, Naga, et al. "The diagnosis and management of nonalcoholic fatty liver disease: practice guidance from the American Association for the Study of Liver Diseases." *Hepatology* 67.1 (2018): 328-357.

Question 5:

While *Rhbdf2* levels have been measured by immunoblotting in the liver from patients with simple steatosis and NASH (Supplementary Fig. S1A), the authors did not report *Rhbdf2* levels in mice fed a HFD or in obese *ob/ob* mice. Have them been measured? It would also be of interest to confirm this result in different mouse models.

Response 5:

According to your suggestion, here we have additionally performed the western blotting assay to determine the protein expression of *Rhbdf2* in the livers of HFD-fed mice and *ob/ob* mice. Consistent with our other experiment results, protein expression of *Rhbdf2* was significantly increased in the livers of HFD-fed mice and *ob/ob* mice. This corresponding immunoblotting results have been included in revised Supplementary Fig. 1a.

Question 6:

Figure 3J: In the main text, lines 251-253, it is stated that “Likewise, expression of fatty acid uptake and synthesis-related genes and fatty acid β -oxidation-related gene in THTG mice also tended to normal levels, as compared to its controls (Fig. 3J)”. However, in the Figure it is not compared with normal levels, it is compared with NTG HFD. Therefore, the authors should add the values of chow diet fed mice or, otherwise, in the main text the comparison should be referred to NTG HFD, indicating that the expression of fatty acid uptake and synthesis-related genes are decreased and fatty acid β -oxidation-related gene are increased in THTG mice compared to NTG HFD.

Response 6:

Thank you very much for your kindly comments here. We are so sorry for this ambiguous and confusing text. Actually, in the main text, the comparison was consistently referred to NTG HFD. According to your suggestion, we have carefully revised and corrected this corresponding sentence as “Likewise, the expression of fatty acid uptake and synthesis-related genes are decreased and fatty acid β -oxidation-related gene are increased in THTG mice compared to NTG HFD groups”.

Question 7-1:

In Supplementary Fig. S5A, the authors measure visceral fat weight and show that visceral fat weight was elevated in the HFD-fed THKO mice, but this elevation was significantly restrained in THTG mice with a marked change in the ratio of visceral fat weight to body weight (Supplementary Fig. S5A). However, no details are given in the Methods section about white adipose tissue sampling or the methodology used to measure adipose cell size. It is crucial to clearly specify the adipose tissue depots studied. It would also be of certain interest to compare visceral vs. subcutaneous fat weights.

Response 7-1:

Thank you very much for your suggestions, the corresponding method and Refs for “Adipose Tissue Sampling and Adipocyte Size Detection” has been included in Method section.

Adipose Tissue Sampling and Adipocyte Size Detection

The white adipose tissue samples from NCD-fed Flox, THKO, NTG and THTG mice, and HFD-fed Flox, THKO, NTG and THTG mice were fixed with 4% paraformaldehyde, embedded in paraffin, and sectioned transversely. Adipose tissues were then subjected to hematoxylin and eosin (H&E) staining analysis. All of the sections were detected by 3 histologists without knowledge of the treatment procedure. The images were captured using

an optical microscope (Olympus, Japan). The average cell diameter was examined using Image J software from the National Institutes of Health (NIH). The total adipose cell numbers were determined by counting the cells on the H&E sections from at least 3 fields per mouse. The size of the cells was approximated assuming cubic packing as previously indicated^{13,37}.

- Ref: Xu, Minxuan, et al., Functional loss of inactive rhomboid-like protein 2 mitigates obesity by suppressing pro-inflammatory macrophage activation-triggered adipose inflammation. Mol. Metab. 34, 112-123 (2020).
- Ref: Wang Pi-Xiao, Zhang Xiao-Jing, Luo Pengcheng et al. Hepatocyte TRAF3 promotes liver steatosis and systemic insulin resistance through targeting TAK1-dependent signalling. Nat. Commun. 7, 10592 (2016).

In addition, the subcutaneous fat weight of corresponding groups and visceral vs. subcutaneous fat weight ratio were additional performed and included in revised Supplementary Fig. S5a. Consistent with our other results, subcutaneous fat weight was also increased in HFD-fed THKO mice compared to Flox mice, but was decreased by THTG mice compared to HFD-fed NTG mice.

Question 7-2:

The authors stated: “Meanwhile, adipose cells size was greatly upregulated between HFD-fed mice and NCD control, but no significant difference between HFD-fed THKO and THTG group (Supplementary Fig. S5B). This result further suggested that adipocyte proliferation might links to the higher visceral fat weight in HFD-fed mice than those in the controls.” What do the authors mean with “controls”? This paragraph should be better explained.

Response 7-2:

Thank you so much for your concerns here. We have revised and corrected this confusing text as “Meanwhile, adipose cells size was greatly upregulated between HFD-fed mice and Flox mice, but no significant difference between HFD-fed THKO and THTG group. This result further suggested that adipocyte proliferation might links to the higher visceral fat weight in HFD-fed THKO mice than those in the HFD-fed Flox mice”.

Question 8-1:

In liver and in off-target tissues (including tissues for which AAV8 shows tropism when administered systemically such as adipose tissue, pancreas, skeletal muscle...).^Δ In case the promoter used is not hepatic-specific, authors need to report mRNA/protein expression levels of Trim31 and Trim31 RING^Δ- In “Animal experiment design 3#” corresponding to section “Trim31-Rhbd2 interaction is required for Trim31-regulated hepatic steatosis and inflammation”, the authors indicate that to specifically overexpress Trim31 in hepatocytes in vivo, AAV8 vectors were used. The Trim31-encoding AAV8 vectors were delivered via tail vein. What was the promoter used in the expression cassette to drive expression of Trim31 and Trim31 RING.

Response 8-1:

Thank you so much for reviewer’s question here. We are very sorry for ambiguity of the text in this section. Actually, the vector used in this experiments is adeno-associated virus serotype 8 -thyroxine-binding globulin (AAV8-TBG). The promoter is hepatocyte-specific promoter TBG.

A simple outline of AAV8-TBG-Trim31 and AAV8-TBG-Trim31 RING^Δ vector were showed below:

We also corrected AAV8 as AAV8-TBG vector in Method section.

Question 8-2:

In line 733, the authors state that “the opening reading frame (ORF) encoding Trim31 without a stop codon was cloned into AAV8 vector to generate AAV-Trim31”. Please, clarify why a stop codon was not used. Did the expression cassette bear a polyA sequence?

Response 8-2:

As we mentioned in Response 8-1, expression cassette with poly-A tail was included in our vector. Also, this sentence should be corrected and revised as “the whole opening reading frame (ORF) encoding Trim31 without intervening stop codon was cloned into AAV8 vector to generate AAV-Trim31”. This unprecise statement has been carefully corrected in our this section.

Question 8-3:

Regarding AAV8 vectors used as control, please clarify if GFP-encoding vectors or empty vectors were used. In line 737 and Supplementary Figure 10 legend the authors specify that “The empty vector was injected into mice as the corresponding control”. However, in line 423 and Supplementary Figure 10, AAV-GFP were used. In addition, please, clearly indicate components of the expression cassette (i.e. promoter used, transgene).

Response 8-3:

The empty vector used as corresponding control in our study is AAV8-loading GFP vector. We have corrected this “empty vector” in corresponding text as “AAV-GFP”.

The outline of AAV-GFP vector has been showed below:

Question 8-4:

The authors also need to specify methods used to produce, purify and titer AAV vectors used in the study.

Response 8-4:

According to your comments, the protocol regarding AAV construction, production and purification has been included in Methods section.

Adeno-associated virus construction and production

AAV8-TBG vector, a pre-packaged AAV in serotype 8 with overexpression of GFP, was used to produce recombinant AAV8-TBG-gene of interest-GFP expression vector. This vector contains transcriptional control elements from the thyroxine-binding globulin (TBG) promoter, cloning sites for the insertion of a complementary DNA and the polyA signal. Terminal repeats from AAV serotype 2 flank the expression cassette. The murine full-length Trim31 sequences or Trim31 with RING domain deletion sequences was then cloned into AAV8-TBG-GFP, respectively. This newly created vector AAV-TBG-Trim31-GFP or AAV-TBG-Trim31 RING^Δ-GFP was packaged into AAV8, purified by ViraBind™ AAV Purification Mega Kit (VPK-141/VPK-141-5, Cell Biolabs, VPK-141/VPK-141-5, San Diego, USA) and accordingly titered by QuickTiter™ AAV Quantitation Kit (Cell Biolabs, VPK-145). Viral particles were diluted to a total volume of 50 µl with saline immediately before injection.

Question 8-5:

Please, also indicate expression cassettes and methods used to produce, purify and titer lentiviral vectors used in the manuscript. In line 420, the authors indicate “adeno-associated virus subtype 8”. The term “subtype” should be substituted by the term “serotype”, already used by the authors in the Methods section.

Response 8-5:

According to your comments, the protocol regarding lentiviral vectors construction, production and purification has been included in Methods section.

Lentivirus construction and production

To generate the lentiviral-Trim31 (LV-Trim31) or lentiviral-Trim31 with RING domain deletion (LV-Trim31 RING^Δ) vectors, the full-length Trim31 cDNA sequences were packaged into pLenti-CMV-GFP-Puro (Addgene) to upregulate Trim31 expression (pLenti-CMV-Trim31-GFP-Puro or pLenti-CMV-Trim31 RING^Δ-GFP-Puro) *in vivo* experiments. The commercial Lenti-Pac HIV Expression Packaging Kit (LT002, GeneCopoeia, MD, USA) and corresponding Lenti-Pac 293Ta Cell Line were used to

produce LV particles. Next, according to the product instruction, the 293T cells culture supernatants containing virus particles were harvested. The newly created vector was concentrated and purified by ViraBind™ PLUS Lentivirus Concentration and Purification Kit (Cell Biolabs, VPK-095) and then titered by QuickTiter™ Lentivirus Quantitation Kit (Cell Biolabs, VPK-112). The functional LV titers in the 10⁶ TU/ml range were achieved, and after concentration yields of up to 10⁹ TU/ml were attained.

In addition, according to your comment, the term “subtype” has been substituted by the term “serotype” in our manuscript.

Question 9-1:

In Figure 8, the authors aim to investigate the therapeutic feasibility and effect of targeting Trim31-Rhbd2-MAP37K pathway in the development and progression of NASH. However, in this part of the work, the authors studied the effect of Trim31 overexpression in THTG mice. Thus, they do not properly evaluate the therapeutic feasibility and effect of targeting Trim31-Rhbd2 administered signaling as suggested by the subheading. They rather studied the effect of Trim31 overexpression in a model of NASH. In a translational point of view, this strategy certainly goes further than the use of a NAFLD model induced by HFD, but the conclusions of this experiment are limited because of the use of a transgenic model. A gene therapy approach using AAV vectors would be more appropriated to properly evaluate the therapeutic potential of Trim31 targeting.

Response 9-1:

Thank you so much for your comments here. To reduce this limitation, the pro-conditioned 8-weeks-HFHF fed mice were then injected with the AAV8-TBG-Trim31 vectors to further evaluate the preventive effects of Trim31 on HFHF-induced NASH. Consistent with the results of Fig. 8, indeed, lipid accumulation, inflammation and fibrosis were significantly decreased by AAV8-delivered Trim31. This data was also included in our revised Supplementary figure.17 for your consideration.

Question 9-2:

Importantly, in the last sentence of the results part, line 480, the authors indicate that “Collectively, our data revealed the efficacy of Trim31 in the treatment of steatohepatitis and associated metabolic syndrome in murine.” In the major part of their experiments they demonstrate prevention of steatohepatitis but not reversion. Only, in the *ex vivo* gene therapy intervention approach by lentivirus-loading full-length Trim31 sequences (Figure 4) the overexpression of Trim31 is induced 8 weeks after the HFD-feeding. However, in this experiment they should show that in the moment of the injection of the transduced hepatocytes to the recipient mice, the steatosis is already established, and that after the injection of the

vectors there is an amelioration of the phenotype. Only in this case they can postulate that Trim31 revealed efficacy in the treatment of steatohepatitis.

Response 9-2:

According to your meaningful comments, the sentence regarding brief summary of this result has been carefully revised and corrected to properly follow the obtained data. This sentence has been corrected as “Collectively, our data revealed the positive effects of Trim31 on mitigation of steatohepatitis and associated metabolic syndrome in murine.” Also, as you mentioned above, *ex vivo* gene therapy is a very common method and widely used in animal model. We are very grateful and surprised for your advice on using existing steatosis cells for transplantation. This method you proposed is very instructive for our future study, and we will use this method in our future experiments.

Question 10:

In the Discussion section (line 493), the authors stated that “we identified liver Trim31 as a markedly negative regulator of high-energy diet (HFD/HFHF)-stimulated or genetically induced chronic inflammation, insulin resistance, liver steatosis and NASH.”. This sentence is confusing, and I would suggest to rather state: we identified liver Trim31 as a markedly protective regulator against high-energy diet (HFD/HFHF)-stimulated or genetically induced chronic inflammation, insulin resistance, liver steatosis and NASH.

Response 10:

Thank you so much for your kindly suggestion, here we have corrected this sentence in text according to your comment.

We gratefully thank for the precious time the reviewer spent making constructive remarks. We hope our responses and revision could meet your requirement.

Reviewers' Comments:

Reviewer #2:

Remarks to the Author:

The authors have responded to questions raised by the reviewers and made great changes to the manuscript. However, the reviewer is not fully convinced by the newly added data with regard to the following points:

Response 1:

The authors have made big improvements to this point. Only one to be pointed out in this question: Please indicate the immunoblot analysis of TRIM31 and GAPDH in liver tissue as shown in the revised supplementary Figure 8a, which shows that hTRIM31-RING Δ is certainly introduced.

Response 2-1:

Revised supplementary Figure 9c upper panels: Why GAPDH is co-immunoprecipitated with TRIM31 or RHBDF2? Moreover, this method cannot rule out the possibility of non-specific precipitation of prey proteins. Immunoprecipitation with anti-RHBDF2 antibody may cause nonspecific immunoprecipitation of mouse TRIM31 protein and vice versa, which is not mediated by RHBDF2-TRIM31 binding. To avoid this, IP experiments are usually performed using isotype-matched control antibodies as control experiments. Please perform experiments with these controls as well and also show the immunoblot analysis of input fraction.

Response 2-3:

Revised Figure 6C and revised supplementary Figure 9c lower panels: Again, it looks like there is no band of TRIM31-Flag CCdelta and RHBDF2-HA Taildelta in input fraction, which means that TRIM31-Flag CCdelta and RHBDF2-HA Taildelta are not expressed. To assess the relative interaction of multiple proteins with a given protein, the expression levels of the multiple proteins need to be comparable. So the authors should increase the expression levels of TRIM31-Flag CCdelta and RHBDF2-HA Taildelta. For example, increasing the amount of plasmid to be introduced or using a proteasome inhibitor are possible. If the authors cannot increase the expression level by all means, in vitro pulldown assay using recombinant proteins would be a good way. Moreover, immunoprecipitated bait proteins should be assessed. If samples are immunoprecipitated with an anti-FLAG antibody, IB:FLAG/IP:FLAG is required, and the same is true for an anti-HA antibody.

Response 4:

Revised Figure 5D: Again, compared AdGFP with AdshRhbdf2, it looks like that there are not clear differences in p-AKT/AKT, pGSK3 β /GSK3 β , and pFOXO1/FOXO1 in the PA/THKO background. So the conclusion in the page 13, lines 345-347 is not based on sufficient evidences.

Response to minor comment 4:

Page 15 line 427, Page 16 line 430, and Page 17 lines 466-468: Revised supplementary figure 10-13 can support the requirement of the RING-finger domain of TRIM31 for the effects of TRIM31 on regulation of the hepatic steatosis and inflammation. However, these data cannot support whether TRIM31-RHBDF2 interaction is required for the protective function of TRIM31, because TRIM31-RING delta can still bind to RHBDF2. To support the authors' claims, the authors should use TRIM31 mutant construction, which cannot bind to RHBDF2. It is uncertain whether TRIM31-Flag CCdelta can be used in these experiments, as this mutant construct must be expressed as much as that of the wild-type. I wonder if the statement of "Trim31-Rhbdf2 interaction is essential" is essential in this author's fine work...

Reviewer #3:

Remarks to the Author:

In this new version of the manuscript, the Authors carefully addressed all the critical issues raised in the first review. The authors made changes to the manuscript that significantly improve its quality. However, I have a minor concern that still needs to be addressed:

Question 3-2:

In Animal experiment design 1, it is stated that "At the end of experimental period, the liver tissue samples were collected from mice to detect corresponding signaling events." What else have been measured to confirm that these mice are obese and with fatty liver? Similarly, it is stated that "Moreover, the dynamic expression levels from 0 to 16 weeks after HFD treatment indicated that Trim31 expression was gradually suppressed in the fatty liver (Fig. 1C)." However, expression levels should be correlated to a quantification of lipids in the liver to allow this affirmation. The authors should give more details about the obese and fatty liver phenotype of HFD-fed mice, indicate the number of mice used in this experiment, and quantify the degree of fat accumulation in the liver in these mice in order to correlate this parameter to Trim31 expression levels.

Response 3-2:

Thank you so much for your comments here. To confirm that the WT mice we used in our study were obese and with fatty livers, histological analysis of the livers and quantification of lipids in the liver were included in this part.

As shown in this figure, prolonged HFD treatment (16 weeks) significantly increased liver lipid accumulation and promoted progression of hepatic steatosis, as compared to NCD-fed mice. This figure was also included in revised Supplementary Fig. 1b (n=15 for HFD-fed mice; n=15 for NCD-fed mice).

New Question 3-2:

In the revised Supplementary Fig. 1b, the second panel (middle) shows decreased levels of liver TG in HFD-fed mice vs. NCD-fed mice, in contrast to what is described in the main text and would be expected. The authors should carefully revise this figure.

Reviewer #4:

Remarks to the Author:

The Authors have addressed all the comments (including experimental) raised by the Reviewer and the manuscript has much improved in quality and clarity.

Response to the Reviewers' Comments R2

To Reviewer #2:

The authors have responded to questions raised by the reviewers and made great changes to the manuscript. However, the reviewer is not fully convinced by the newly added data with regard to the following points:

We appreciate the reviewer's compliment on our manuscript's importance that "The authors have responded to questions raised by the reviewers and made great changes to the manuscript".

General comments:

New Question 1:

The authors have made big improvements to this point. Only one to be pointed out in this question: Please indicate the immunoblot analysis of TRIM31 and GAPDH in liver tissue as shown in the revised supplementary Figure 8a, which shows that hTRIM31-RING Δ is certainly introduced.

New Response 1:

Thank you so much for your efforts to review and improve our work. According to your comments, here we have indicated the western blotting assay of TRIM31 and GAPDH levels in liver tissue as shown in the revised supplementary Figure 8a, which further confirmed that hTRIM31-RING Δ is certainly introduced and expressed in liver tissue of transplanted mice. The corresponding western blotting band for expression of hTRIM31(RING mutant) has been included in revised supplementary Figure 13a. The TRIM31 antibody (C-term) (Cat: #AP13642B) used in this experiment was obtained from ABGENT, Inc., (San Diego, CA, USA).

Revised supplementary Figure 13a

New Question 2:

Revised supplementary Figure 9c upper panels: Why GAPDH is co-immunoprecipitated with TRIM31 or RHBDF2? Moreover, this method cannot rule out the possibility of non-specific precipitation of prey proteins. Immunoprecipitation with anti-RHBDF2 antibody may cause nonspecific immunoprecipitation of mouse TRIM31 protein and vice versa, which is not mediated by RHBDF2-TRIM31 binding. To avoid this, IP experiments are usually performed using isotype-matched control antibodies as control experiments. Please perform experiments with these controls as well and also show the immunoblot analysis of input fraction.

New Response 2:

We thank the reviewer for pointing out the issue. According to your suggestion, we have re-performed this IP assay experiment using isotype-matched control antibodies as control experiments. The corresponding result has been included in revised supplementary Figure 10c.

Also, we are sorry for the ambiguous mark of co-IP assay in Revised supplementary Figure 9c upper panels. Actually, the GAPDH expression was detected using Input samples in this part. This issue has been carefully corrected.

Revised supplementary Figure 10c

New Question 3:

Revised Figure 6C and revised supplementary Figure 9c lower panels: Again, it looks like there is no band of TRIM31-Flag CCdelta and RHBDF2-HA Taildelta in input fraction, which means that TRIM31-Flag CCdelta and RHBDF2-HA Taildelta are not expressed. To assess

the relative interaction of multiple proteins with a given protein, the expression levels of the multiple proteins need to be comparable. So the authors should increase the expression levels of TRIM31-Flag CCdelta and RHBDF2-HA Taildelta. For example, increasing the amount of plasmid to be introduced or using a proteasome inhibitor are possible. If the authors cannot increase the expression level by all means, *in vitro* pulldown assay using recombinant proteins would be a good way. Moreover, immunoprecipitated bait proteins should be assessed. If samples are immunoprecipitated with an anti-FLAG antibody, IB:FLAG/IP:FLAG is required, and the same is true for an anti-HA antibody.

New Response 3:

We are very grateful to the reviewers for providing us with very meaningful and important suggestions here. According to your comments, increasing the amount of vectors were used to elevate the expression levels of TRIM31-Flag CCA and RHBDF2-HA TailΔ. Thus, this protein binding experiments were next re-performed according to our original approach, as described in methods section. The results we obtained in the re-tested experiment are consistent with our following data and conclusion. The re-performed result was updated in revised Figure 6c.

Revised Figure 6c

Also, we further re-performed the immunoprecipitation to determine the possible protein interaction between Trim31-Flag CCA and Rhbdf2-HA TailΔ *in vitro*. Consistent with results obtained in L02 cells, indeed, protein interaction cannot be significantly observed in cells with Trim31-Flag CCA and Rhbdf2-HA TailΔ transfection. The re-performed result was

updated in revised supplementary Figure 10c lower panels.

Revised supplementary Figure 10c

New Question 4:

Revised Figure 5D: Again, compared AdGFP with AdshRhbdf2, it looks like that there are not clear differences in p-AKT/AKT, pGSK3 β /GSK3 β , and pFOXO1/FOXO1 in the PA/THKO background. So the conclusion in the page 13, lines 345-347 is not based on sufficient evidences.

New Response 4:

Thank you very much for your comments. Here, compared AdGFP with AdshRhbdf2, we have re-performed this experiments to highlight the differences in p-AKT/AKT, pGSK3 β /GSK3 β , and pFOXO1/FOXO1 in the PA/THKO groups, and make them comply with our following results and conclusion. The corresponding WB bands have been updated and revised in Figure 5d.

Revised Figure 5d

Minor comment:

New Question:

Page 15 line 427, Page 16 line 430, and Page 17 lines 466-468: Revised supplementary figure 10-13 can support the requirement of the RING-finger domain of TRIM31 for the effects of TRIM31 on regulation of the hepatic steatosis and inflammation. However, these data cannot support whether TRIM31-RHBDF2 interaction is required for the protective function of TRIM31, because TRIM31-RING delta can still bind to RHBDF2. To support the authors' claims, the authors should use TRIM31 mutant construction, which cannot bind to RHBDF2. It is uncertain whether TRIM31-Flag CCdelta can be used in these experiments, as this mutant construct must be expressed as much as that of the wild-type. I wonder if the statement of "Trim31-Rhbd2 interaction is essential" is essential in this author's fine work...

New Response:

Thank you so much for your concerns here. In our study, we have confirmed Trim31 as a Rhbdf2-related protein.

Also, as we concluded in results section:

TRIM31 directly interacts with RHBDF2. As a key member of the E3 ubiquitin ligase, human TRIM31 is mainly composed of three parts: N-terminal RING-finger domain, B-Box domain, and C-terminal coiled-coil (CC) domain (Fig. 6c). To confirm which domain of TRIM31 is mainly responsible for the interaction with RHBDF2, a series of truncated mutants of human TRIM31 with Flag-tagged vectors were then produced for the following binding experiments. The co-immunoprecipitation assays demonstrated that the coiled-coil domain (CC) contributes its capacity to bind to RHBDF2.

On the other hand, the following supplementary figures 11-14 indicate the requirement of the RING-finger domain of TRIM31 for the protective effects of TRIM31 on regulation of the hepatic steatosis and inflammation. Therefore, we have concluded that C-terminal CC domain of TRIM31 is primarily responsible for the binding and interaction with the targeted protein, while its N-terminal RING domain is responsible for ubiquitin ligase activity and catalytic function. The results obtained in our current work are consistent with previous reports (Refs below).

Importantly, according to your significant suggestions, to make our conclusion more rigorous

and accurate, the statement of “Trim31-Rhbf2 interaction is essential.....” in our revised manuscript has been changed as “**Trim31-Rhbf2 interaction significantly and positively contributes to.....**”. Indeed, we believe that this statement is more convincing for readers. Of note, the influences of Trim31-Rhbf2 interaction on NAFLD progression and its more precise molecular mechanisms need to be further studied in the following experiments. And we will continue to study it in the future.

Again, thank you very much for your constructive suggestions to improve our current work.

Relevant Refs:

1. Song, Hui, et al., *The E3 ubiquitin ligase TRIM31 attenuates NLRP3 inflammasome activation by promoting proteasomal degradation of NLRP3. Nat. Commun.* **7**, 1-11 (2016).
2. Wang, Haiyu, et al., *TRIM31 regulates chronic inflammation via NF- κ B signal pathway to promote invasion and metastasis in colorectal cancer. Am. J. Transl. Res.* **10**, 1247 (2018).
3. Zanchetta, Melania E., and Germana Meroni, *TRIM proteins legitimately enter the MAGEic RING. Cell. Cycle.* **14**, 1134 (2015).

We gratefully appreciate for your valuable suggestions in improving our work.

We hope our corrections in this current revised manuscript could meet your requirements.

To Reviewer #3:

In this new version of the manuscript, the Authors carefully addressed all the critical issues raised in the first review. The authors made changes to the manuscript that significantly improve its quality. However, I have a minor concern that still needs to be addressed:

We appreciate the reviewer's compliment on our revised manuscript's importance that "The authors made changes to the manuscript that significantly improve its quality".

New Question 1:

In Animal experiment design 1, it is stated that "At the end of experimental period, the liver tissue samples were collected from mice to detect corresponding signaling events." What else have been measured to confirm that these mice are obese and with fatty liver? Similarly, it is stated that "Moreover, the dynamic expression levels from 0 to 16 weeks after HFD treatment indicated that Trim31 expression was gradually suppressed in the fatty liver (Fig. 1C)." However, expression levels should be correlated to a quantification of lipids in the liver to allow this affirmation. The authors should give more details about the obese and fatty liver phenotype of HFD-fed mice, indicate the number of mice used in this experiment, and quantify the degree of fat accumulation in the liver in these mice in order to correlate this parameter to Trim31 expression levels.

New Response 1:

Thank you very much for your significant comments. To determine the fatty liver phenotype, a set of indicators including body weight (BW), liver weight (LW), LW/BW ratio, liver TG, TC, NEFA levels, serum AST and ALT contents, NAS score of liver sections, and Trim31, Rbhd2 mRNA expression profiles of liver samples were collected at each time point (0, 4, 8, 16 weeks) in NCD- and HFD-fed mice. A total of 15 WT mice for each time point were used in this part. The corresponding results collected from these experiments were further included in new reorganized supplementary Figure 2a-d.

Revised supplementary Figure 2a-d

Meanwhile, to correlate these parameters to Trim31 expression in obese and fatty liver phenotype of HFD-fed mice, multiple Pearson correlation analysis were further performed to confirm that Trim31 expression was negatively correlated with fatty liver pathologies in HFD-fed mouse model. The corresponding results were further included in new reorganized supplementary Figure 2e-l.

Revised supplementary Figure 2e-l

New Question 2:

In the revised Supplementary Fig. 1b, the second panel (middle) shows decreased levels of liver TG in HFD-fed mice vs. NCD-fed mice, in contrast to what is described in the main text and would be expected. The authors should carefully revise this figure.

New Response 2:

Thank you very much for reviewer’s concerns here. We are so sorry for our careless mistake regarding the ambiguous mark of TG contents analysis. Accordingly, we have re-examined

the raw data and re-tested the relevant indicator using the original samples. The previous raw data and the following retested raw data for this issue were concurrently submitted. The corresponding retested data for hepatic TG levels has been included in revised Supplementary Fig. 1c.

Revised Supplementary Fig. 1c

Finally, we gratefully appreciate for your valuable suggestions in improving our work. We hope our revised work could meet your requirements.

To Reviewer #4:

The Authors have addressed all the comments (including experimental) raised by the Reviewer and the manuscript has much improved in quality and clarity.

Response:

Thank you so much for your efforts to review and improve our current study.

Reviewers' Comments:

Reviewer #2:

Remarks to the Author:

The authors have addressed all the questions raised by the reviewer and the manuscript has been greatly improved in quality.

Reviewer #3:

Remarks to the Author:

The Authors have addressed all the comments raised by this Reviewer and the manuscript has significantly improved its quality.

Response to the Reviewers' Comments

Reviewer #2 (Remarks to the Author):

The authors have addressed all the questions raised by the reviewer and the manuscript has been greatly improved in quality.

Response:

Thank you so much for your efforts to review and improve our current study.

Reviewer #3 (Remarks to the Author):

The Authors have addressed all the comments raised by this Reviewer and the manuscript has significantly improved its quality.

Response:

Thank you so much for your efforts to review and improve our current study.